# Cyclophilin A supports translation of intrinsically disordered proteins and affects haematopoietic stem cell ageing

Laure Maneix [1,2,3,4,15], Polina Iakova[1,2,3,4,15], Charles G. Lee[5], Shannon E. Moree [1,2,3,4], Xuan Lu[3], Gandhar K. Datar[2,3], Cedric T. Hill[6], Eric Spooner[7], Jordon C. K. King[1,2,4], David B. Sykes[6], Borja Saez [8], Bruno Di Stefano [2,3,4], Xi Chen [3], Daniela S. Krause [9], Ergun Sahin[1,10], Francis T. F. Tsai[3,11,12], Margaret A. Goodell [2,3,4], Bradford C. Berk[13], David T. Scadden[6] & André Catic [1,2,3,4,14] ✉

Loss of protein function is a driving force of ageing. We have identified peptidyl-prolyl isomerase A (PPIA or cyclophilin A) as a dominant chaperone in haematopoietic stem and progenitor cells. Depletion of PPIA accelerates stem cell ageing. We found that proteins with intrinsically disordered regions (IDRs) are frequent PPIA substrates. IDRs facilitate interactions with other proteins or nucleic acids and can trigger liquid–liquid phase separation. Over 20% of PPIA substrates are involved in the formation of supramolecular membrane-less organelles. PPIA affects regulators of stress granules (PABPC1), P-bodies (DDX6) and nucleoli (NPM1) to promote phase separation and increase cellular stress resistance. Haematopoietic stem cell ageing is associated with a post-transcriptional decrease in PPIA expression and reduced translation of IDR-rich proteins. Here we link the chaperone PPIA to the synthesis of intrinsically disordered proteins, which indicates that impaired protein interaction networks and macromolecular condensation may be potential determinants of haematopoietic stem cell ageing.

Haematopoiesis is a dynamic regenerative process. Haematopoietic stem cells (HSCs) give rise to rapidly dividing progenitor cells that spawn hundreds of billions of cells daily[1]. In contrast to progenitor cells, stem cells are long-lived and highly durable, with a low mitotic index. Given their longevity and the absence of frequent cell division to dispose of protein aggregates, maintaining protein homeostasis (proteostasis) is critical to HSC biology. As the proteostasis capacity of cells declines with age[2], protein aggregates accumulate. Yet, maintenance of a balanced proteome

[1]Huffington Center on Aging, Baylor College of Medicine, Houston, TX, USA. [2]Stem Cells and Regenerative Medicine Center, Baylor College of Medicine, Houston, TX, USA. [3]Department of Molecular and Cellular Biology, Baylor College of Medicine, Houston, TX, USA. [4]Cell and Gene Therapy Program at the Dan L. Duncan Comprehensive Cancer Center, Houston, TX, USA. [5]Department of BioSciences, Rice University, Houston, TX, USA. [6]Center for Regenerative Medicine, Massachusetts General Hospital, Harvard Medical School, Boston, MA, USA. [7]Whitehead Institute for Biomedical Research, Cambridge, MA, USA. [8]Center for Applied Medical Research, Hematology-Oncology Unit, Pamplona, Navarra, Spain. [9]Georg-Speyer-Haus, Institute for Tumor Biology and Experimental Therapy, Frankfurt am Main, Germany. [10]Department of Molecular Physiology and Biophysics, Baylor College of Medicine, Houston, TX, USA. [11]Department of Biochemistry and Molecular Pharmacology, Baylor College of Medicine, Houston, TX, USA. [12]Department of Molecular Virology and Microbiology, Baylor College of Medicine, Houston, TX, USA. [13]Department of Medicine, University of Rochester School of Medicine and Dentistry, Rochester, NY, USA. [14]Michael E. DeBakey Veterans Affairs Medical Center, Houston, TX, USA. [15]These authors contributed equally: Laure Maneix, Polina Iakova. ✉e-mail: catic@bcm.edu

is essential for stem cell function[3–5]. Proteostasis requires precise control of protein translation, folding, transport and degradation. Molecular chaperones are key actors in this network, facilitating the folding of newly translated polypeptides and preserving the functional conformation of pre-existing proteins. Chaperone function to maintain proteome integrity is of particular importance in long-lived stem cells, allowing these cells to retain their regenerative potential and avert the effects of ageing[6].

Protein misfolding is a driving force of ageing[7]. Mammalian cells express several hundred chaperones and co-chaperones to reduce protein misfolding[8]. Among them are four different families of prolyl isomerases, which facilitate conversion between *trans*- and *cis*-isomers of proline. Cyclophilins, the most abundant type of prolyl isomerases, have been implicated in the ageing process. For example, a recent heterochronic parabiosis study demonstrated that cyclophilin A (peptidyl-prolyl isomerase A, PPIA) is the single most upregulated gene across tissues in young animals exposed to old blood, indicating that this chaperone is sensitive to ageing[9]. Although young tissues increased PPIA levels in response to blood from older mice, proteomic evidence indicated that PPIA levels decline with age[10]. In addition, despite the high expression of PPIA, protein aggregates in neurodegenerative diseases can functionally sequester the available pool of this chaperone[11]. Despite growing evidence of its role in ageing, how PPIA affects the cellular proteome has not previously been studied.

The recent realization that many functional proteins lack a defined structure has revolutionized our understanding of the protein structure–function relationship[12]. Intrinsically disordered proteins interact with proteins, nucleic acids or other molecules through conformational selection or induced fit[13]. Hence, proteins rich in intrinsically disordered regions (IDRs) regulate many cellular processes by promoting specific activities between molecules or allowing the formation of network hubs through scaffolding. The lack of structural order does not imply that these proteins are transient intermediates during evolution. Quite the opposite, genes encoding intrinsically disordered proteins are targets of positive selection[14,15]. However, little is known about the translation of IDR-rich proteins. Specifically, whether they require dedicated chaperone support has not yet been examined. Our research on chaperones within the haematopoietic system indicates that PPIA is a master regulator governing the synthesis of disordered proteins. Furthermore, genetic depletion of PPIA—or natural ageing—results in a stem and progenitor cell proteome distinctively lacking IDR-rich proteins, which is not reflected at the messenger RNA (mRNA) level. These results demonstrate that reduced structural proteome diversity is both a consequence of and a driver of the ageing process.

## Results

### Loss of PPIA causes an ageing-like haematopoietic phenotype

To identify the most prevalent chaperones in the haematopoietic compartment, we analysed the proteome of mouse haematopoietic stem and progenitor cells (HSPCs) through semi-quantitative two-dimensional (2D) gel electrophoresis and mass spectrometry (MS) (Fig. 1a and Extended Data Fig. 1a). Strikingly, PPIA accounted for up to 14% of discernible protein peaks within the cytosolic proteome, making it one of the most abundant chaperones in HSPCs (Fig. 1a,b and Supplementary Data 1). In addition, PPIA was the most highly expressed chaperone at the transcript level, accounting for over 0.5% of all mRNAs (Fig. 1b and Supplementary Data 2). These findings are supported by a recent study that examined the proteomic composition across diverse haematopoietic compartments and also detected a pronounced expression of PPIA protein in HSPCs (Extended Data Fig. 1b)[16]. Prolyl isomerases are conserved enzymes that are grouped into four classes. Cyclophilins comprise one class, with 17 members in humans[17] (Extended Data Fig. 1c). Cyclophilins catalyse the isomerization of proline, the only proteinogenic amino acid that exists in abundance in both *trans* and *cis* configurations[18]. It has been shown previously in PPIA knockout (*Ppia*[−/−]) mice[19] that this gene is non-essential, and the animals showed no apparent phenotype under homeostatic conditions in the C57BL/6 background[20].

To assess the role of PPIA in haematopoiesis, we compared knockout and heterozygous animals in a series of functional assays following bone marrow (BM) transplants. In these assays, heterozygous animals were indistinguishable from wild types. We competitively transplanted CD45.2[+] total nucleated BM cells from *Ppia*[−/−] or *Ppia*[+/−] mice together with equal numbers of CD45.1[+] wild-type support BM cells into lethally irradiated CD45.1[+] recipient animals (Fig. 1c). Six months after transplantation, when the BM was fully repopulated by long-lived donor HSCs, we observed a statistically significant decrease of *Ppia*[−/−] B lymphocytes in the blood of recipient animals. In contrast, myeloid cells were increased in recipients of knockout cells (Fig. 1d). Changes in BM progenitor cells drove this myeloid skewing in the peripheral blood (PB), as we observed an increase in common myeloid *Ppia*[−/−] progenitor cells at the expense of lymphoid progenitor cells (Fig. 1e). We also found higher relative and absolute numbers of HSPCs and myeloid-biased CD150[high] HSCs[21] in recipients of *Ppia*[−/−] BM (Fig. 1f).

To functionally define stem cell activity, we performed limiting dilution transplantation experiments with *Ppia*[−/−] and *Ppia*[+/−] BM cells. The results were comparable between these groups, indicating that higher numbers of immunophenotypic stem cells in the knockout BM did not correlate with increased stem cell activity (Extended Data Fig. 1d). Next, we tested the self-renewing ability of HSCs by measuring the repopulation of BM following serial transplantations of *Ppia*[−/−] or *Ppia*[+/−] donor cells (Fig. 1g). Unlike wild-type or heterozygous BM cells, *Ppia*[−/−] cells showed declining engraftment after the first round of transplantation and displayed exhaustion in long-term repopulation assays (Fig. 1h). Taken together, these functional transplant assays revealed cell-intrinsic defects leading to myeloid skewing, an immunophenotypic but not functional increase in stem cells, and impaired self-renewal with accelerated exhaustion in *Ppia*[−/−] HSCs. These three characteristics are

**Fig. 1 | PPIA deficiency induces an ageing-like haematopoietic phenotype.**
**a**, Left: HSPC lysate labelled with amine-reactive dye and separated on a 2D electrophoresis gel. Right: quantitative representation. Outlines indicate acetylated and non-acetylated PPIA. Dominant ontologies within each peak are depicted (representative of two independent experiments). **b**, Left: RNA-seq reads in the mouse HSPC transcriptome. PPIA is the sixth most highly expressed gene and the most highly transcribed chaperone in young HSPCs. Right: MS-based protein levels in the mouse HSPC proteome. PPIA is the second most highly expressed chaperone protein in the total proteome of young HSPCs. The results are representative of two independent biological replicates. **c**, Experimental workflow of competitive BM transplantation. **d**, Six months after BM transplantation, flow cytometry reveals that PPIA knockout (*Ppia*[−/−]) BM donor cells undergo a myeloid shift in the PB compared to animals receiving *Ppia*[+/−] BM. Total blood reconstitution was measured as a ratio of CD45.2[+] to CD45.1[+] cells (*n* = 10 mice per group at transplant initiation; data are representative of two independent experiments). **e**, Seven months after transplantation, BM flow cytometry shows that mice transplanted with *Ppia*[−/−] donor cells have increased common myeloid progenitors (CMPs) and decreased common lymphoid progenitors (CLPs) compared to recipients of *Ppia*[+/−] BM (*n* = 10 per group at initiation). **f**, Flow cytometry analysis comparing the ratios of HSPCs (LKS; lineage[−]/c-Kit[+]/Sca1[+] cells) and CD150[high] (lineage[−]/c-Kit[+]/Sca1[+]/CD34[−]/CD135[−]/CD150[high]) HSCs following transplantation of *Ppia*[−/−] or *Ppia*[+/−] donor cells (*n* = 10 per group at initiation). **g**, Experimental workflow of competitive serial BM transplantation. **h**, Flow cytometry shows that *Ppia*[−/−] donor-derived progenitor cells exhaust in serial transplantations. Depicted is the proportion of donor-derived (CD45.2[+]) cells among PB cells two, four and twelve months after the first transplantation (*n* = 5 mice per group in the first round, *n* = 8 mice per group in the second and third rounds; data are representative of two independent experiments). For **d**–**f** and **h**, data are means ± s.d.; *\*P* < 0.05, *\*\*P* < 0.01 and *\*\*\*P* < 0.001; two-sided Wilcoxon rank-sum test. FPKM, fragments per kilobase of transcript per million mapped reads; iFOT, intensity-based fraction of total; NS, not significant.

hallmarks of haematopoietic ageing[22–25], suggesting that the absence of PPIA resembles premature ageing at the stem-cell level.

To substantiate the causal relationship between PPIA and the ageing phenotype, we subsequently performed rescue experiments. Haematopoietic stem and progenitor cells (lin⁻/Sca1⁺/c-Kit⁺), 18 months in age, were transduced with either a *Ppia* vector or a control mock vector (Fig. 2a). The overexpression of *Ppia* induced a significant enhancement in haematopoietic reconstitution (Fig. 2b,c). The observed improvement in haematopoiesis was not associated with a myeloid bias (Fig. 2d). The enhancement was sustained over a duration of six months, suggesting an improvement in the long-term function of haematopoietic stem cells.

## PPIA substrates are enriched for IDRs

PPIA is a highly and ubiquitously expressed prolyl isomerase that interacts with a wide range of proteins[26,27]. PPIA isomerizes proline residues of the nascent polypeptide chain. Several in vitro refolding studies have demonstrated that *cis/trans* isomerization of prolyl bonds can be rate-limiting during translation[28–31]. To gain insight into the molecular changes caused by its depletion, we aimed to identify the substrate proteins of PPIA. We accounted for non-specific interactions and distinguished PPIA-selective substrate proteins using a previously identified PPIA G104A mutant[32], which is functionally impaired. We tested several mutations and found that inactivation of the catalytic core yielded

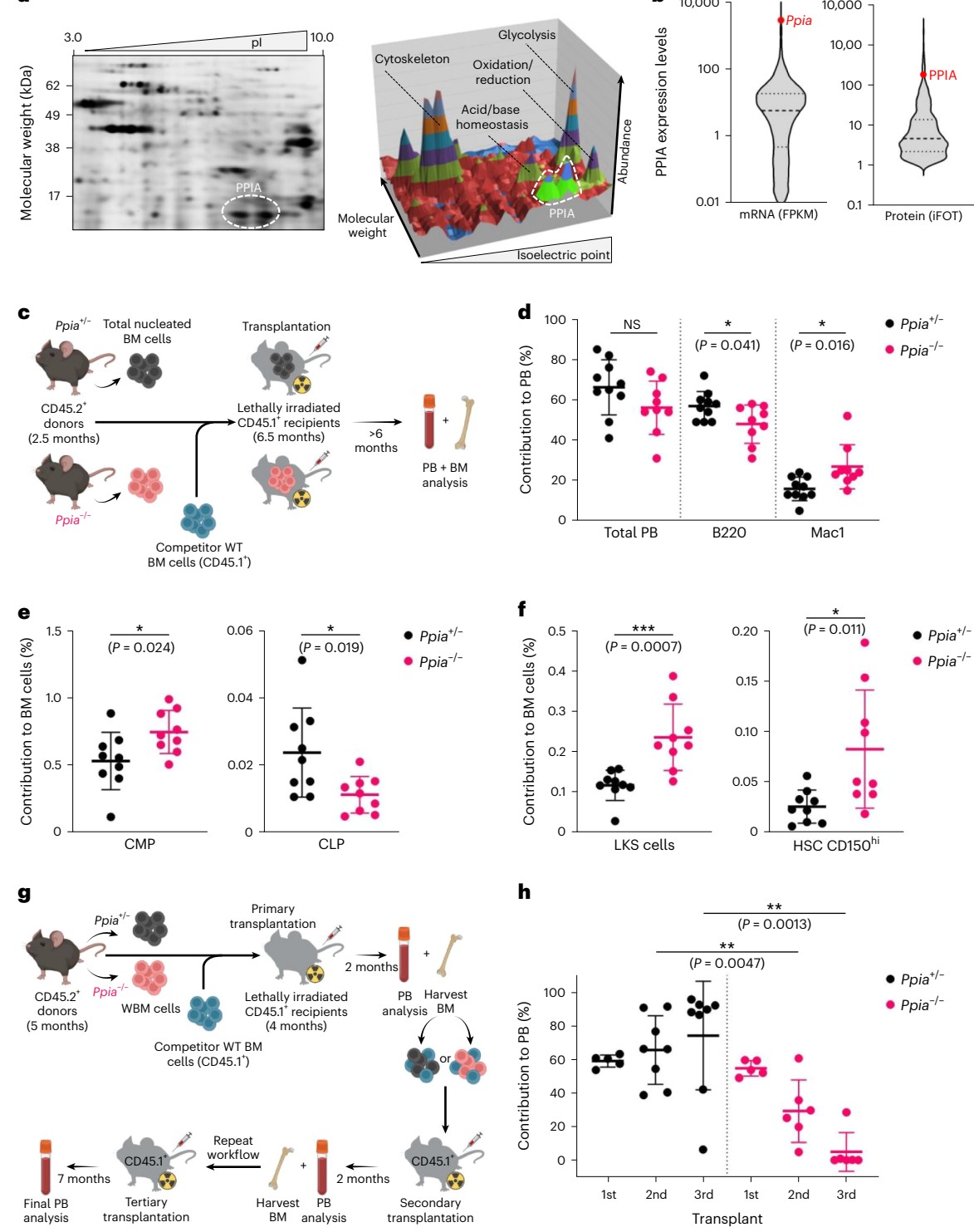

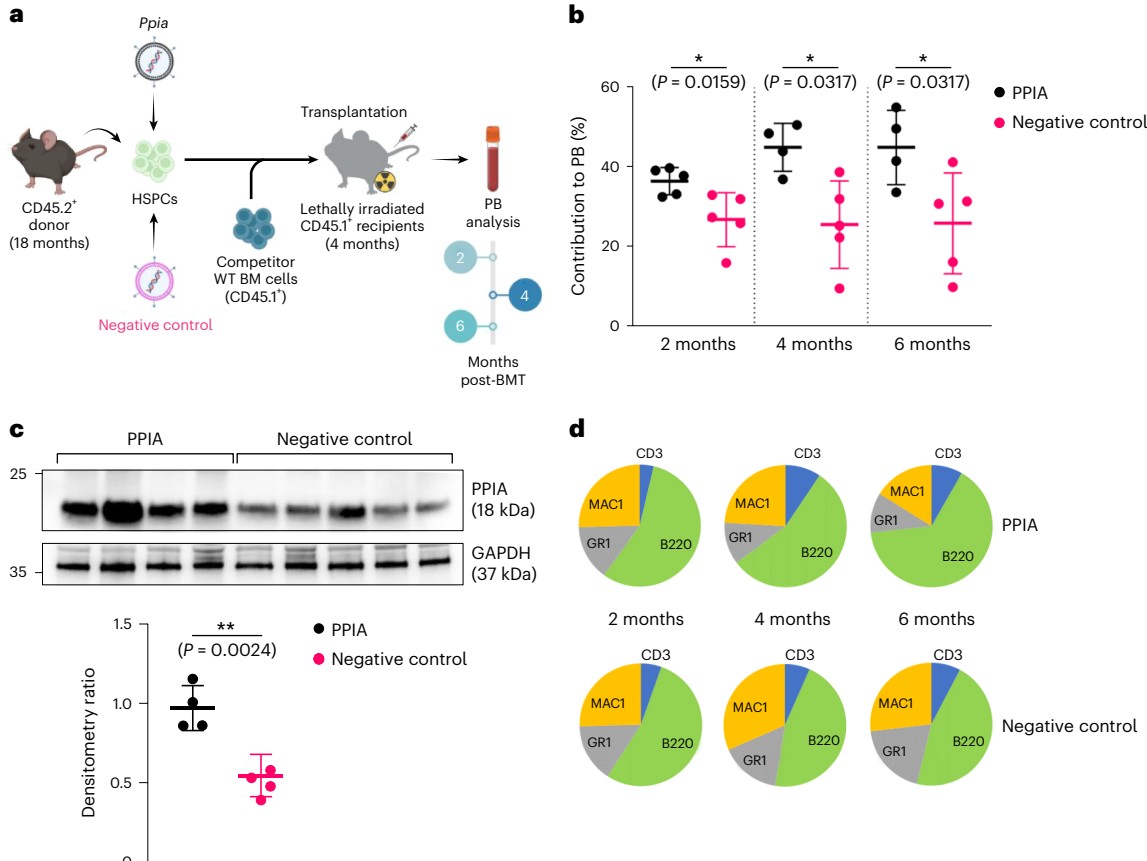

**Fig. 2 | PPIA overexpression improves transplantation outcomes of aged BM. a**, Experimental workflow. PB chimerism after up to six months of observation following transplantation of aged (18-month-old) lineage⁻/c-Kit⁺/Sca1⁺ BM cells transduced with *Ppia* expressing lentivirus or negative control (reverse complement). **b**, Overexpression of PPIA improves haematopoietic reconstitution of aged CD45.2⁺ BM two, four and six months after transplantation (data are means ± s.d.; transplant initiated with *n* = 5 mice per group). **c**, PPIA levels in the PB are elevated in animals receiving *Ppia*-transduced BM (data are means ± s.d.). **d**, PPIA overexpressing blood shows no signs of myeloid bias in the CD45.2⁺ lineage (*P < 0.05, **P < 0.01; two-sided Wilcoxon rank-sum test).

insoluble PPIA, while the G104A mutation, which moderately reduces substrate access to the catalytic core through an obstructing methyl group (Fig. 3a), allowed for normal expression levels and intracellular distribution (Fig. 3b). Therefore, proteins interacting preferably with the wild-type PPIA over the PPIA G104A mutant are probably direct substrates of this chaperone. Differential co-immunoprecipitation between the wild-type PPIA and the mutant PPIA revealed ~400 substrates of the wild-type enzyme (Fig. 3a,c, Extended Data Fig. 2a,b and Supplementary Data 3), including 49 transcriptional regulators (Extended Data Fig. 2c). Because we performed the co-immunoprecipitation in the cytosolic cell fraction, these results suggest that PPIA interacts with its DNA-binding substrates during translation and before nuclear translocation (Fig. 3c).

In addition to DNA-binding proteins, the most prevalent group of PPIA substrates included RNA-associated proteins involved in ribosome assembly, translation and splicing (Fig. 3d). When compared to the global proteome, immunoprecipitated PPIA substrates feature higher levels of IDRs, which represent unstructured protein regions displaying a sequence-driven preference for conformational heterogeneity[33] (Fig. 3e). Prolyl isomerases such as PPIA catalyse the reversible *trans*- and *cis*-conversions of peptide bonds in proline residues, which can be over-represented within IDRs[34].

### PPIA promotes expression of its substrates

In line with PPIA's proposed activity as a co-translational chaperone[8] and given that proline isomerization is slow and often rate-limiting during translation, we expected PPIA expression to directly affect de novo protein translation of its substrates. We determined the synthesis of proteins using pulsed stable-isotope labelling in cell culture (SILAC) in HeLa and 293T cells with either normal or reduced levels of PPIA (Fig. 4a and Extended Data Fig. 2d). We found that loss of PPIA reduced expression, specifically of PPIA substrates, in both cell types (Fig. 4b and Supplementary Data 4). Overall, the synthesis levels of PPIA-targeted proteins were lower than for other proteins, and these reduced further when PPIA was depleted (Fig. 4c). These results demonstrate that PPIA supports de novo translation of its target proteins, consistent with previous reports that IDR-rich proteins have a lower translation rate[35,36] (Extended Data Fig. 2e).

To confirm that PPIA also promotes protein translation within the haematopoietic system, and to delineate a more acute kinetic timeline of this process, we employed staining techniques using a fluorescent aminoacyl-tRNA analogue on highly purified BM stem cells. Notably, over a span of 2 h, we discerned a significantly decreased rate of de novo translation in stem cells that had been treated with the potent PPIA inhibitor TMN355 (Extended Data Fig. 3a)[37].

Within PPIA substrate proteins, the dominant ontologies we found were translation and mRNA splicing, and we discovered that more than 20% of PPIA substrates are known to engage in protein phase separation[38–42] (Fig. 4d). For example, liquid–liquid phase separation allows the intracellular compartmentalization of ribonucleoprotein (RNP) assemblies through changes in solubility and subsequent formation of membrane-less organelles[34,43]. Prominent examples of phase-separating proteins that we found to bind more robustly to wild-type than to mutant PPIA include the mRNA regulator poly(A) binding protein cytoplasmic 1 (PABPC1), found in stress granules and

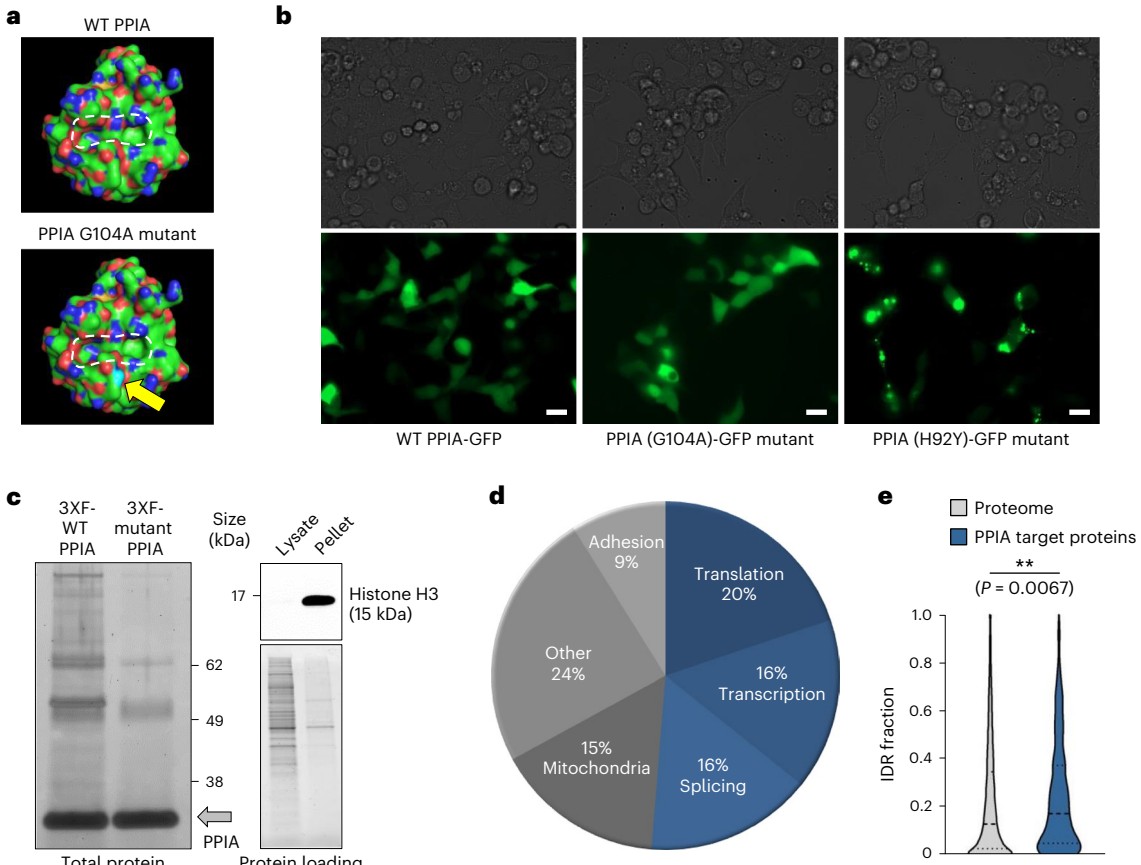

**Fig. 3 | PPIA interacts with intrinsically disordered proteins. a**, 3D models of the wild-type (WT) PPIA structure and the G104A mutant PPIA, which has restricted access to the catalytic core[17,32]. Dashed white lines outline the PPIA catalytic core. The arrow indicates the additional A104 methyl group in the mutant (light blue spheres). The PPIA structure was sourced from X-ray crystallography data deposited at the Protein Data Bank (7ABT)[11]. **b**, Expression pattern of WT and mutant PPIA proteins. 293T cells were transiently transfected with WT PPIA-GFP, PPIA (G104A)-GFP mutant or PPIA (H92Y)-GFP catalytic core mutant. At 24 h post transfection, the expression pattern of the WT and PPIA mutant proteins was assessed with a Zeiss Celldiscoverer 7 imaging system, using a GFP filter (bottom) or bright field (top). Scale bars, 20 μm. Data are representative of three independent experiments. **c**, Left: Immunoprecipitation and SYPRO Ruby gel stain of triple FLAG-tagged wild-type (3XF-WT PPIA) and

G104A point-mutant PPIA (3XF-Mutant PPIA) were performed in 293T cells to identify PPIA-interacting proteins. The grey arrow indicates positive enrichment for PPIA protein in the pull-down fractions. Right: purity of the cytosolic cell lysates was verified by a lack of histone H3 protein expression in this subcellular fraction. Data are representative of three independent experiments. **d**, Gene Ontology enrichment analysis of 3XF-WT PPIA versus 3XF-Mutant PPIA immunoprecipitation–MS results. Data represent 385 consistently identified proteins in 293T cells (overlapping MS results from two separate experiments are depicted). **e**, Quantification of the fraction of IDRs in the total proteome versus 3XF-WT PPIA-interacting proteins. **P < 0.01; two-sided Wilcoxon rank-sum test (violin plot lines indicate quartiles; n = 385 PPIA target proteins). GFP, green fluorescent protein.

P-bodies, the P-body regulator DEAD-box helicase DDX6, the nucleolar oligomeric protein nucleophosmin 1 (NPM1), which is integral to liquid–liquid phase separation of the nucleolus, and the stress granule initiator G3BP1. We validated our proteomics data by confirming the interaction of endogenous PPIA with its phase-separating substrates by IP–western blot (Fig. 4e,f) and demonstrated that protein expression of IDR-rich PPIA substrates is reduced in human epithelial and haematopoietic cell lines, as well as in mouse HSPCs that are deficient for PPIA (Fig. 4g and Extended Data Fig. 3b,c).

To determine whether the observed changes in PPIA substrate proteins are driven at the transcriptional level, we performed RNA sequencing in *Ppia* heterozygous (serving as wild-type control) and knockout mice (Extended Data Fig. 3d). Our analysis revealed three crucial findings. First, the absence of PPIA does not induce a compensatory upregulation of other chaperones or the ubiquitin-proteasome system, which is responsible for the degradation of misfolded proteins. Instead, we observed robust upregulation of genes involved in cytoplasmic translation. This result functionally links PPIA more closely with protein synthesis, as opposed to protein removal pathways. Second, HSPCs from *Ppia* knockout mice exhibit a molecular signature akin to

that of ageing BM stem cells and show upregulation of the key haematopoietic ageing marker P-selectin (Extended Data Fig. 3e)[44,45]. This suggests that the absence of PPIA triggers cell-autonomous ageing processes, corroborating the ageing-like phenotype we detected in our preceding transplantation studies (Fig. 1). Third, in PPIA-deficient HSPCs, we detected a transcriptional upregulation of PPIA substrates, suggesting a potential compensatory response (Extended Data Fig. 3f). Such upregulation may serve to mitigate the effect of impaired translation in cells deficient in PPIA. Supporting a role of PPIA in protein synthesis, as opposed to refolding or removal, we found no indication of heightened protein aggregation in *PPIA* knockdown cells under homeostatic conditions (Extended Data Fig. 3g).

**PPIA affects liquid–liquid phase separation of its substrates**

IDRs within proteins can initiate liquid–liquid phase separation and trigger the formation of membrane-less organelles[46–48]. For example, the RNA-binding protein PABPC1 engages in phase separation involving its unstructured proline-rich linker region, which is critical for the formation of RNA stress granules[49]. Following *PPIA* knockdown, expression of PABPC1 protein was reduced by 20–30% by western blot,

suggesting the chaperone stabilizes PABPC1 (Fig. 4g). We biochemically confirmed that PABPC1 is a substrate of PPIA (Fig. 4e,f and Extended Data Fig. 2b,c) and that PABPC1 protein expression is reduced following genetic depletion of PPIA in haematopoietic cell lines and primary HSPCs (Extended Data Fig. 3b,c). In response to diverse stresses or unfavourable growth conditions, PABPC1 undergoes phase transition and participates in stress granule formation to sequester cytoplasmic RNA and ribosomes (Supplementary Video 1). This allows cells to temporarily reduce protein translation[50,51]. Treating cells with the oxidative stressor sodium arsenite is a well-established experimental approach to study the formation of stress granules. To determine whether proline isomerization affects the phase separation of PABPC1, we genetically modulated PPIA activity and assessed the dynamics of stress-granule formation. Upon stress induction with sodium arsenite, we observed significantly reduced stress granule formation in the absence of prolyl isomerase activity (Fig. 5a and Supplementary Videos 2 and 3), and cells devoid of PPIA were more susceptible to cell death following arsenite treatment. In addition, reintroduction of the chaperone partially rescued stress granule formation in *PPIA* knockdown cells. Similar to our findings with PABPC1, we discovered that cells lacking PPIA displayed reduced formation of P-bodies, as evidenced by diminished DDX6 staining (Fig. 5b). Furthermore, we determined that PPIA-deficient cells exhibited smaller and more fragmented nucleoli, as indicated by the nuclear distribution of NPM1 (Fig. 5c). The combined data indicate that reduced PPIA activity significantly influences liquid–liquid phase separation in all three examined instances and lowers the ability of cells to form condensates containing these PPIA substrate proteins.

Although visualizing cytoplasmic protein condensates in primary haematopoietic stem cells posed challenges due to the limited extra-nuclear space in these cells, we successfully confirmed interactions between PPIA and PABPC1, DDX6 and NPM1 using proximity ligation assays (PLAs) on highly purified stem cells (Fig. 5d).

Collectively, these findings support the notion that PPIA regulates the folding of its substrates by controlling proline isomerization as early as during translation. Our data suggest that PPIA plays a crucial role in maintaining appropriate translation levels of its substrates. Even when these substrates are transcriptionally upregulated, PPIA remains necessary for their full functionality.

### PPIA and IDR-rich proteins decline with age

We next addressed how our finding that PPIA regulates the function of intrinsically disordered polypeptides relates to the haematopoietic phenotype, which resembles ageing, as observed in the transplantation of *Ppia*[−/−] BM cells. A previous quantitative proteomic analysis of human dermal fibroblasts showed that PPIA is significantly reduced with age[10]. In the haematopoietic compartment, we found a substantial reduction of PPIA protein in HSCs from old mice compared to younger cells

(Fig. 6a). The notable decrease in PPIA protein levels in aged HSCs has recently been corroborated by an independent study (Extended Data Fig. 4a)[16]. Importantly, *Ppia* transcripts were not altered in HSPCs of different ages in our RNA-seq data (Supplementary Data 2). Based on our earlier findings, we would expect reduced PPIA activity to result in lower expression of IDR-rich proteins. Indeed, using two distinct tandem mass spectrometry (MS/MS) methods to quantitatively compare the proteome of young and old mice, we observed a reduction of IDR-rich proteins in HSPCs during ageing, which is not apparent at the transcriptome level (Fig. 6b, Extended Data Fig. 4b,c and Supplementary Data 1, 2, 5 and 6). Consistently, PPIA-depleted haematopoietic stem and progenitor cells also demonstrated a significantly diminished level of intrinsic disorder within their proteome, similar to the structural transition that we observed during natural ageing (Fig. 6c, Extended Data Fig. 4d,e and Supplementary Data 5 and 7). To validate these findings orthogonally, we applied unbiased hierarchical clustering to classify proteome changes in haematopoietic cells expressing PPIA compared to PPIA-deficient cells. The results of this analysis revealed that proteins rich in IDRs are more sensitive to the expression of PPIA than proteins with lower IDR levels (Extended Data Fig. 5a–c). In summary, we show that PPIA levels decline with age in haematopoietic cells. This change is accompanied by a remarkable deficiency in IDR-rich proteins.

## Discussion

Long-lived cells, such as HSCs, are particularly vulnerable to the accumulation of misfolded proteins and subsequent proteotoxic stress[4]. This risk is further exacerbated by the low division rate observed in HSCs, a process that could otherwise facilitate the dilution of protein aggregates. Additionally, the inherently low proteasome activity in these cells also contributes to a heightened proteostatic susceptibility[52]. Thus, ensuring protein integrity is a critical determinant of stem cell function. In this study we have characterized the protein content of mouse haematopoietic stem and progenitor cells and discovered that PPIA is a prevalent chaperone that ensures structural proteome diversity in these cells. Considering the high expression of PPIA, it is tempting to compare its function with another highly abundant prolyl isomerase: in prokaryotes, trigger factor is a well-documented and extensively studied example of a ribosome-associated chaperone that facilitates the folding of nascent polypeptides[53]. Multiple lines of evidence suggest that PPIA also acts as a chaperone in the early stages of protein homeostasis. First, prolyl isomerization is a rate-limiting step during translation, and previous studies have implicated an involvement of PPIA in protein synthesis[18]. Second, PPIA displays close spatial association with functional ribosomes[54]. Third, PPIA can facilitate the expression of substrate proteins[55]. Fourth, PPIA-deficient cells do not exhibit enhanced protein aggregation under homeostatic conditions (Extended Data Fig. 3g). Instead, we observed reduced

---

**Fig. 4 | PPIA activity promotes expression of proteins enriched for IDRs.**
**a**, Schematic of the pulsed SILAC experiment to evaluate protein synthesis. Pulse treatment for 24 h allowed for metabolic labelling of newly translated proteins. Protein synthesis was determined by the heavy to unlabelled ratio quantified by MS, as described in refs. 35,64. Two independent experiments were performed. **b**, Pulsed SILAC was performed and protein extracts from control or *PPIA* knockdown (Kd) cell lines were analysed to measure newly synthetized proteins. The heatmap represents the relative degree of protein synthesis (*n* = 345 overlapping proteins compared between the different cell types). **c**, Uptake of heavy amino acids by control or *PPIA* Kd 293T cells was quantified following a pulsed SILAC experiment. A value of 0.5 indicates the equal presence of light and heavy labelled peptides. *P < 0.05, **P < 0.01 and ****P < 0.0001; two-sided Wilcoxon rank-sum test (comparing 160 PPIA target proteins to 1,280 non-targets; similar findings were observed for HeLa cells). **d**, List of PPIA client proteins involved in protein phase separation based on PhaSepDB2.0. Proteins are listed by their official gene name. Nuclear bodies include nucleoli, Cajal bodies, nuclear speckles, paraspeckles, promyelocytic leukemia

(PML) nuclear bodies and histone locus bodies. **e**, Immunoprecipitation of endogenous PPIA protein in HeLa cells followed by a western blot to detect the PPIA protein partners (poly(A)-binding protein 1 (PABPC1), DEAD-box helicase 6 (DDX6), Ras GTPase-activating protein-binding protein 1 (G3BP1) and nucleophosmin 1 (NPM1)). Data are representative of two independent experiments, confirmed by unbiased MS. **f**, PPIA activity is required for substrate binding. PPIA wild type, but not the G104A mutant, binds to PABPC1, DDX6 and NPM1 in IP–western experiments in HeLa cells. Data are representative of two independent experiments, confirmed by unbiased MS. **g**, Reduced expression of PPIA substrates following knockdown of the chaperone. HeLa cells were stably transduced with negative control or *PPIA* knockdown construct Kd1. Shown are representative immunoblots of *n* = 3 independent biological replicates. Right: a pairwise comparison shows significant reduction of PABPC1, DDX6 and NPM1 in *PPIA* knockdown cells by densitometry. GAPDH expression was used as a reference. Data are means ± s.d.; *P < 0.05, **P < 0.01; two-sided paired Student's *t*-test. DMEM, Dulbecco's modified Eagle medium.

protein translation following PPIA depletion or inhibition (Fig. 4b,c and Extended Data Fig. 3a). Finally, reduced levels of PPIA decrease the expression of its substrate proteins, despite partial transcriptional compensation (Fig. 4c,g and Extended Data Fig. 3b,c,f).

In an effort to define the substrate selectivity of PPIA, we found that proteins enriched in IDRs are frequent targets of the chaperone. IDRs have been implicated in diverse cellular functions, including the phenomenon of liquid–liquid phase separation, which requires protein–protein or protein–nucleic acid interactions to allow the formation of membrane-less organelles. Our data suggest that PPIA engages with its substrates early during translation, presumably to

*cis*-isomerize prolines within IDRs. Over 20% of PPIA substrate proteins participate in liquid–liquid phase separation, and we have found evidence for impaired protein condensation in the absence of the chaperone. For example, PPIA might influence the structures of PABPC1, DDX6 and NPM1. Without the chaperone, their ability to undergo phase transition diminishes. Given the number of key regulators of phase separation among PPIA substrates, our findings indicate that proline isomerization might more generally impact the formation of membrane-less organelles[34,56]. Proline residues play a key role during phase transition of prion-like and unstructured proteins, and can regulate protein solubility and amyloid formation in an isomer-specific

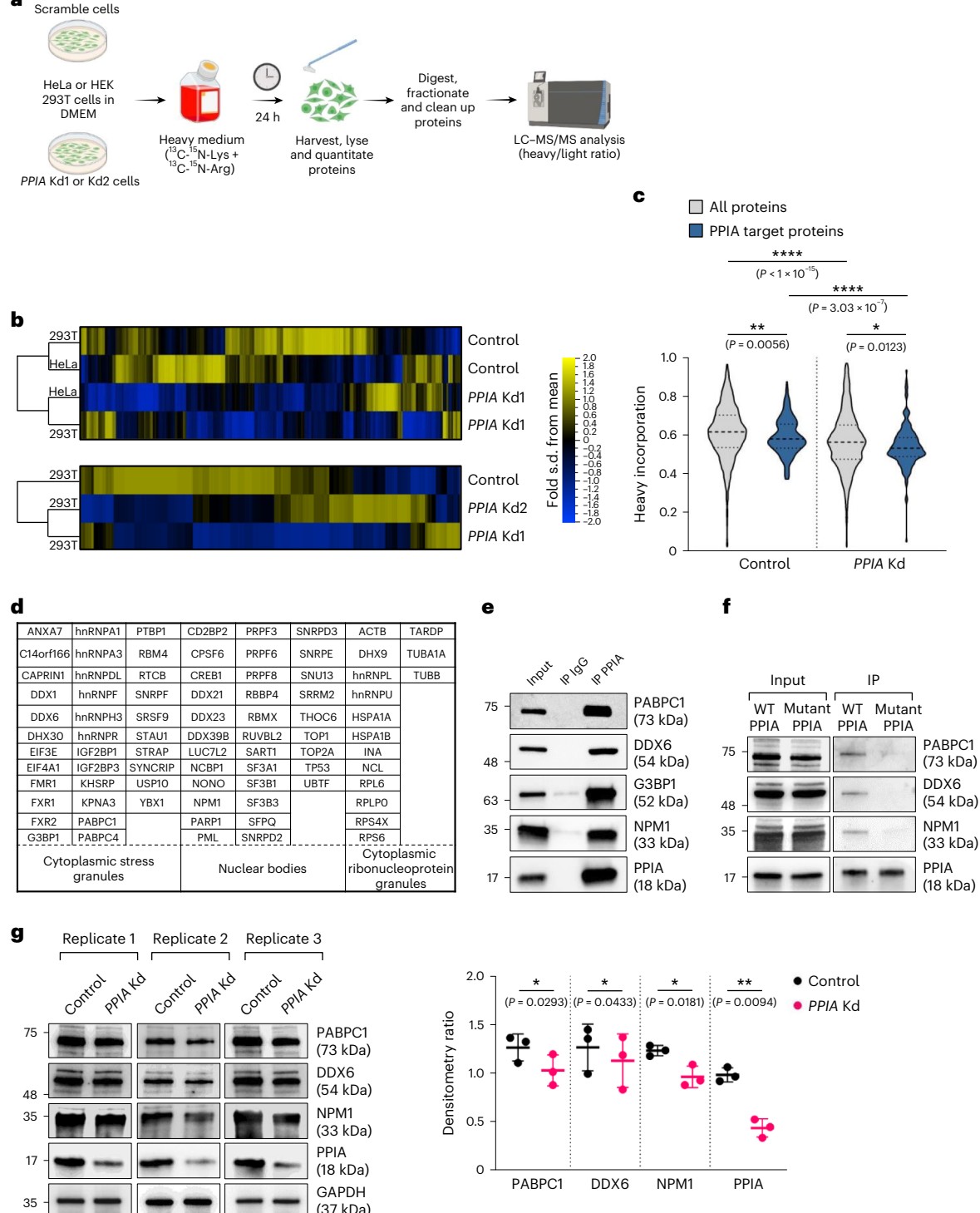

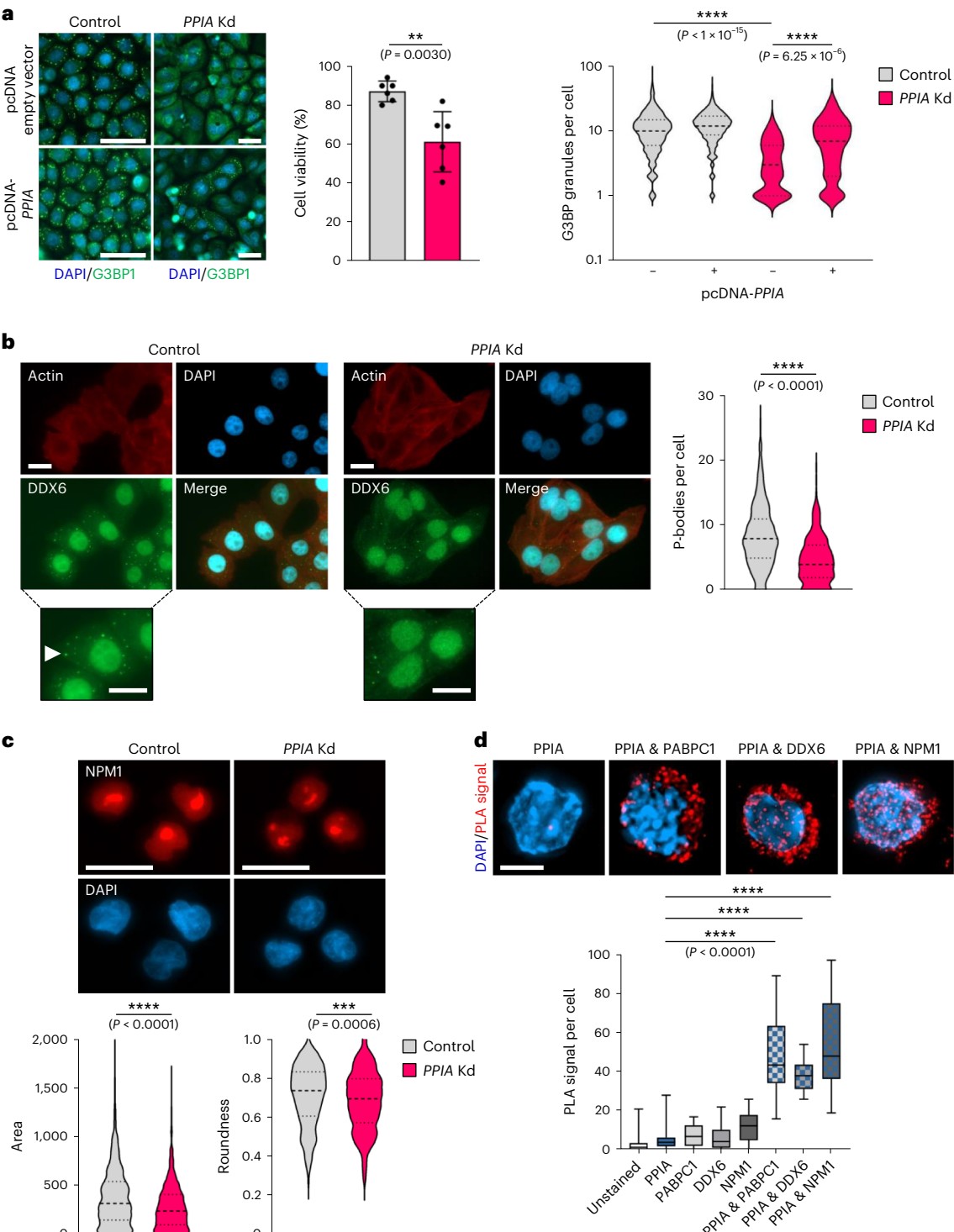

**Fig. 5 | PPIA regulates protein phase separation of its substrates. a**, Stress-granule formation was visualized and quantified with G3BP1 staining after stress induction with sodium arsenite in HeLa control or *PPIA* Kd cells. DAPI, blue; G3BP1, green. Scale bars, 50 μm. *PPIA* knockdown was partially rescued by the reintroduction of knockdown-resistant *PPIA*. Cell viability was measured on an automated cell counter with acridine orange/propidium iodide staining solution using $n = 6$ independently treated replicates per group. Data are means ± s.d.; **$P < 0.01$, ****$P < 0.0001$; two-sided Wilcoxon rank-sum test; $n = 616$ (control), $n = 656$ (control + *PPIA*), $n = 254$ (*PPIA* knockdown) and $n = 293$ (*PPIA* knockdown + *PPIA*) cells were analysed following blinding. Data are representative of three independent experiments. **b**, Staining for DDX6 revealed significantly fewer P-bodies in HeLa cells following *PPIA* knockdown. Scale bars, 20 μm. The arrowhead indicates a representative P-body. ****$P < 0.0001$; two-sided Wilcoxon rank-sum

test; $n = 457$ (control) and $n = 284$ (*PPIA* knockdown) cells were analysed following blinding. Data are representative of three independent experiments. **c**, OCI-AML3 cells that express a stable, gene-edited NPM1-mCherry fusion, exhibited smaller, more fragmented nucleoli following *PPIA* knockdown. Cell designation was blinded for the analyst. Scale bars, 20 μm. ***$P < 0.001$, ****$P < 0.0001$; two-sided Wilcoxon rank-sum test; $n = 564$ (control) and $n = 617$ (*PPIA* knockdown) cells were analysed following blinding. Data are representative of three independent experiments. **d**, PLAs in primary haematopoietic stem cells (lin⁻/c-Kit⁺/Sca1⁺/CD34⁻/CD135⁻) between PPIA and PABPC1, DDX6 and NPM1. Shown is the signal quantification using single-antibody staining as a control. Scale bar, 5 μm. ****$P < 0.0001$; two-sided Wilcoxon rank-sum test. Cells were analysed following blinding. Box plots indicate minima, maxima and quartiles; $n = 20$ randomly chosen cells per group. Data are representative of two independent experiments.

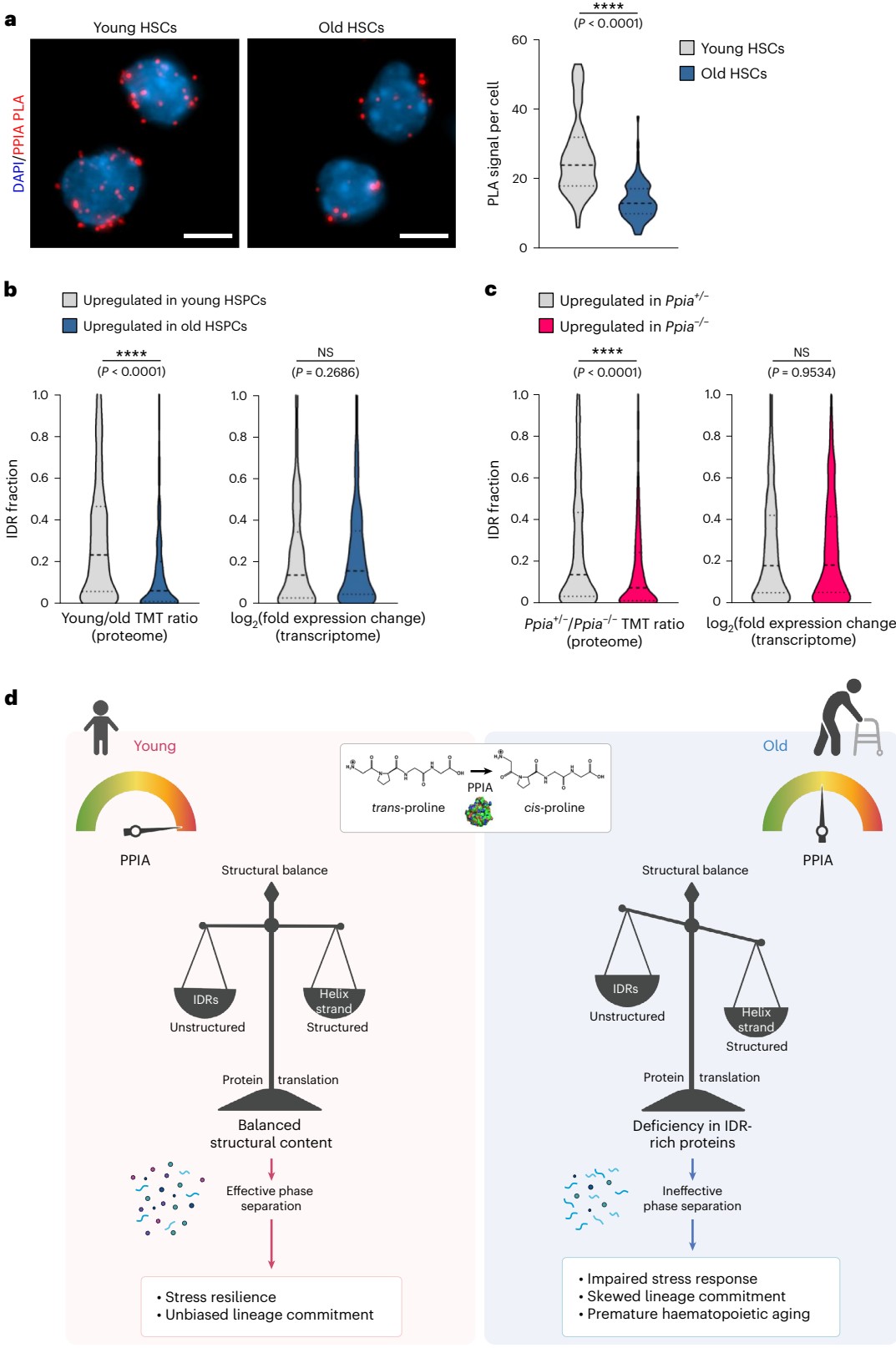

fashion[57–60]. How proline residues within IDRs, which, by definition, do not undergo a proper folding process, affect the overall structure and activity of proteins, remains an intriguing subject of further research[55]. We cannot conclusively determine whether the reduced formation of stress granules, P-bodies or irregular nucleoli in PPIA-depleted cells is due to a decreased quantity of PPIA substrates or their structural changes. Liquid–liquid phase separation is inherently influenced by the

concentration of the interacting components[61], but it is also modulated by their structural characteristics. A recent study underscored the role of PPIA in the condensation of proteins implicated in neurodegenerative conditions[62]. Our findings, along with those of others, indicate that neither inhibition nor depletion of PPIA in vivo instantaneously impacts the formation of stress granules[63]. PPIA inhibitors required more than 12 h of treatment before reduced stress granule formation was evident.

**Fig. 6 | PPIA levels decline with age and contribute to IDR protein deficiency in HSPCs. a**, PLA to quantify PPIA protein levels in mouse HSCs shows decreased PPIA expression in 23-month-old HSCs when compared to 5-month-old cells. DAPI, blue; PPIA, red. Scale bars, 5 µm. ****$P < 0.0001$; two-sided Wilcoxon rank-sum test; $n = 70$ (young) and $n = 101$ (old) cells were analysed following blinding. Data are representative of two independent experiments. **b**, Quantification of IDR content in the mouse HSPC proteome by tandem-mass-tag (TMT) MS/MS and in the transcriptome (RNA-seq). Shown are the top quartiles of proteins/genes upregulated in young or old cells. NS, not significant; ****$P < 0.0001$; determined by two-sided Wilcoxon rank-sum test. Displayed are cumulative results from three independent experiments, with each experiment individually showing consistent outcomes. $n = 726$ proteins and $n = 479$ transcripts were analysed. **c**, Quantification of IDR content in the mouse HSPC proteome (TMT MS/MS)

and transcriptome (RNA-seq). Shown are the top quartiles of proteins/genes upregulated in $Ppia^{+/-}$ or $Ppia^{-/-}$ cells. ****$P < 0.0001$; two-sided Wilcoxon rank-sum test. Displayed are cumulative results from three independent experiments, with each experiment individually showing consistent outcomes. $n = 479$ proteins and $n = 1{,}528$ transcripts were analysed. **d**, Model of PPIA activity and function in the ageing haematopoietic compartment. PPIA supports nascent proteins during translation and affects proline isomerization in IDRs. Therefore, proteins rich in IDRs, some of which can undergo phase separation, may require higher isomerization activity. PPIA expression decreases during haematopoietic ageing, and the aged proteome is consequently depleted of disordered proteins. In conclusion, PPIA deficiency impairs stress response in HSCs, biases lineage commitment, and accelerates HSC ageing.

The slow effect of PPIA inhibition or depletion on liquid–liquid phase separation implies that the proteins involved in condensation must be turned over and replaced, suggesting that PPIA might not directly affect the granules themselves.

The expression of PPIA was significantly decreased in aged HSCs, and loss of PPIA accelerated ageing in the stem-cell compartment. As a consequence, intrinsically disordered proteins, which are challenging to translate[35], are less efficiently synthesized in HSCs and progenitor cells that are aged or genetically depleted of PPIA. We therefore propose that the expression of IDR-rich proteins constitutes a translational bottleneck during ageing, with possible implications for protein phase separation and other stress responses. PPIA's diverse interactions, encompassing cellular functions such as splicing, translation and transcription, highlight its potential to influence various processes within the multifaceted context of ageing. Although our findings show that diminished PPIA levels impact the viability and adaptability of haematopoietic stem cells, the effects on distinct progenitor populations and immune cells warrant further investigation.

In conclusion, our data suggest a role of PPIA in the molecular cascade that drives haematopoietic ageing, particularly in the synthesis of intrinsically disordered proteins and the preservation of a structurally diverse proteome (Fig. 6d). The capacity to synthesize disordered proteins might represent a previously unrecognized determinant of cellular health, one that is not captured through transcript-based methods.

## Online content

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

## Methods

### Mice

All animal experiments and care procedures were conducted at the Massachusetts General Hospital or Baylor College of Medicine facilities in accordance with the Institutional Animal Care and Use Committee (IACUC) protocols approved at each institution, in compliance with all relevant ethical regulations, and following guidelines from the National Institutes of Health Guide for the Care and Use of Laboratory Animals. The animal facilities were approved by the Association for Assessment and Accreditation for Laboratory Animal Care International (AAALAC). The generation of the $Ppia^{-/-}$ mice has been described previously[19]. C57BL/6 and the congenic (CD45.1⁺) B6.SJL strain were purchased from The Jackson Laboratory (000664 and 002014). $Ppia^{+/-}$, $Ppia^{-/-}$ and C57BL/6 wild-type animals were kept under pathogen-free conditions. All mice were housed in ventilated cages, on a standard rodent diet of chow and water ad libitum, under a 12-h light–dark cycle. $Ppia^{-/-}$ mice were born at sub-Mendelian ratios but displayed no abnormal phenotype after multiple generations of backcrossing to the C57BL/6 genetic background. Animals with signs of sickness or infection were excluded from the study.

### 2D electrophoresis gels

We used the Zoom IPGRunner system (Invitrogen) to separate proteins in two dimensions. We isolated the HSPCs of young male and female C57BL/6J mice (four to eight months of age) and lysed them according to the manufacturer's instructions with urea, CHAPS, dithiothreitol (DTT) and ampholytes. CyDyes DIGE fluors (minimal dyes) were used according to the vendor's instructions (Amersham) with fluorophores Cy3 and Cy5 for post-lysis labelling to ensure that only 1–2% of lysines were labelled in a given protein. Labelling intensities were measured with a Typhoon FLA 9000 scanner and quantified with DeCyder 7.0 and ImageQuant software (GE Healthcare). We normalized the total protein abundance based on protein size and lysine concentration for spots with known identity by MS/MS. PPIA quantity was identical in two independent experiments using different fluorophores.

### MS/MS

**Characterization of the mouse HSPC proteome following 2D electrophoresis.** Following trypsinolysis, we analysed digested peptides by reverse-phase liquid chromatography electrospray ionization MS using a Waters NANO-ACQUITY-UPLC system coupled to a Thermo LTQ linear ion-trap mass spectrometer. To identify proteins, we searched the MS/MS spectra against the non-redundant NCBI protein database using the SEQUEST program (http://proteomicswiki.com/wiki/index.php/SEQUEST). Two independent experiments were performed.

**PPIA protein complex identification following 3XF-PPIA immuno-precipitation.** Immunopurified samples were analysed by MS-based proteomics, as previously described[65]. Minor modifications from the previously cited protocol are listed here. Digested peptides were injected into a nano-HPLC 1000 system (Thermo Scientific) coupled to an LTQ Orbitrap Elite mass spectrometer (Thermo Scientific) for the first repeat and a Q Exactive Plus (Thermo Scientific) mass spectrometer for second-repeat samples. Separated peptides were directly electro-sprayed into the mass spectrometer, controlled by Xcalibur software (Thermo Scientific) in data-dependent acquisition mode, selecting fragmentation spectra of the top 25 and 35 strongest ions for first samples and second samples, respectively. MS/MS spectra were searched against the target-decoy human RefSeq database (released January 2019, containing 73,637 entries) with the software interface and search parameters previously described[65]. Variable modification of methionine oxidation and lysine acetylation was allowed. Protein abundance was calculated with the iBAQ algorithm, and the relative protein amounts between samples were compared with an in-house processing algorithm[66]. Hits were limited to proteins that were identified using strict false discovery rate (FDR) levels following peptide spectral matching. Substrates with a preference for wild-type PPIA were defined as having at least 1.5-fold higher abundance of peptide spectral matches (PSMs) compared to mutant PPIA.

**Tandem-mass-tag isobaric labelling for MS analysis of the mouse HSPC proteome.** Whole BM was isolated from the femurs and tibias of four-month-old and 28-month-old male C57BL/6J mice or 10–12-month old PPIA heterozygous ($Ppia^{+/-}$) and knockout ($Ppia^{-/-}$) male mice. Magnetic depletion of lineage-positive haematopoietic cells was performed using the EasySep mouse haematopoietic progenitor cell isolation kit (Stem Cell Technologies), and lineage-depleted stem and progenitor cells were submitted to MS analysis. Cells were lysed with RIPA lysis buffer (Sigma-Aldrich) supplemented with XPert protease inhibitor cocktail (GenDepot), and 50-µg protein samples were subjected to acetone precipitation at −20 °C for 3 h. After centrifugation (12,000g for 5 min), pellets were denatured and reduced with 30 µl of 6 M urea, 20 mM DTT in 150 mM Tris-HCl, pH 8.0, at 37 °C for 40 min, then alkylated with 40 mM iodacetamide in the dark for 30 min. The reaction mixture was diluted tenfold using 50 mM Tris-HCl pH 8.0 before overnight digestion at 37 °C with trypsin (1:25 enzyme to substrate ratio). Digestions were terminated by adding an equal volume of 2% formic acid, and then desalted using Oasis HLB 1-ml reverse-phase cartridges (Waters). Eluates were dried by vacuum centrifugation. The protein digests were labelled by mixing with the appropriate TMT reagent according to the TMTsixplex Isobaric Label reagent protocol (Thermo Scientific). Following incubation at room temperature for 1 h, the reaction was quenched with hydroxylamine to a final concentration of 0.3% (vol/vol). After labelling, the individual reaction mixtures were combined, dried in a vacuum centrifuge to near dryness, then reconstituted in 0.5% formic acid containing 2% acetonitrile and desalted using Oasis HLB 1-ml reverse-phase cartridges (Waters).

An aliquot of TMT tryptic digest (in 2% acetonitrile/0.1% formic acid in water) was analysed by LC–MS/MS on an Orbitrap Fusion Tribrid mass spectrometer (Thermo Scientific) interfaced with an UltiMate 3000 binary RSLCnano system (Dionex). For trapping the sample, the UHPLC was equipped with an Acclaim PepMap 100 trap column (100 µm × 2 cm, C18, 3 µm) and washed with solvent A at a flow rate of 6 µl min⁻¹ for 7 min. Peptides were continuously separated onto an analytical C18 column (100-µm inner diameter × 30 cm, 3 µm) at flow rate of 350 nl min⁻¹. Gradient conditions were as follows: 5% for 8 min; 5–25% B for 200 min; 25–37% B for 22 min; 37–90% B for 10 min; 90% B held for 10 min (solvent A, 0.1% formic acid in water; solvent B, 0.1% formic acid in acetonitrile). The peptides were analysed using a data-dependent acquisition method. Orbitrap Fusion was operated with measurement of FTMS1 at a resolution of 120,000 (at $m/z$ 200), a scan range of 380–1,500 $m/z$, AGC target 2E5 and a maximum injection time of 50 ms during a maximum 3-s cycle time. The most abundant multiply charged parent ions were selected for HCD MS2 at a resolution of 15,000 (at $m/z$ 200) in the Orbitrap MS, with HCD NCE 40, a 1.6-$m/z$ isolation window, AGC target 5E4 and a maximum injection time of 120 ms, and dynamic exclusion was employed for 40 s.

Proteome Discoverer v.1.4 (Thermo Scientific) with SEQUEST HT search engines was used for the spectra preprocessing, and HCD MS2 spectra were used for peptide identification and quantitation based on TMT reporter ions. The spectra were also searched against the decoy database using a target FDR of 1% or 5% using the Percolator. For trypsin, up to two missed cleavages were allowed. The MS tolerance was set to 10 ppm and the MS/MS tolerance to 0.02 Da. Oxidation of methionine was set as a variable modification, and carbamidomethylation on cysteine residues and TMT labelling on lysine and at the peptide N terminus were set as fixed modifications.

**Label-free quantitative proteomic profiling of mouse HSPC global proteome ('365' profiling).** Whole BM was isolated from the femurs and tibias of three-month-old and 21-month-old male C57BL/6J mice.

Magnetic depletion of lineage-positive haematopoietic cells was performed using the EasySep mouse haematopoietic progenitor cell enrichment kit (Stem Cell Technologies), and lineage-depleted stem and progenitor cells were submitted to MS analysis. Following sample lysis and overnight trypsin digestion, reconstituted peptidic fractions were loaded onto a nano-HPLC 1000 system (Thermo Fisher Scientific) coupled to an Orbitrap Fusion Lumos Tribrid mass spectrometer (Thermo Fisher Scientific), with identical acquisition settings as previously described[67]. The trap and capillary HPLC columns have been described previously[65]. The search of resultant MS/MS spectra against the target-decoy mouse RefSeq database (released June 2015, containing 58,549 entries) was carried out with the Proteome Discoverer 2.1 interface (Thermo Fisher) with the Mascot 2.4 algorithm (Matrix Science). The allowed variable modifications were methionine oxidation and protein N-terminal acetylation. The search settings were as follows: a precursor mass tolerance of 20 ppm, a maximum of two missed trypsin cleavages and a fragment ion mass tolerance of 0.5 Da. Assigned peptides were filtered with 1% FDR. The in-house iFOT data-processing algorithm[51,66] was used to calculated the label-free relative abundance of proteins in samples (Supplementary Data 1).

Gene Ontology analyses were performed with the DAVID bioinformatic database (https://david.ncifcrf.gov). The data represent 385 consistently identified proteins in 293T cells from two independent biological replicates. The degree of native protein disorder was determined using the openly available web interface IUPred2A (https://iupred2a.elte.hu/)[68,69]. IUPred2A is a biophysics-based model that predicts intrinsically disordered protein regions in specific proteins with a confidence score between 0 and 1 for each residue, corresponding to the probability of the given protein being in a disordered state. A disordered region was defined as a protein segment having a confidence score greater than 0.5. The IDR computational analysis was free of redundant information to avoid over-representation of duplicated proteins in the MS/MS data.

## Transplantations

All donor and recipient animals were gender-matched and between three and six months of age. Separate experiments were conducted in male and female mice with identical results. Experiments had a statistical power of >80%, and transplants were initiated with at least five animals per group. A priori power calculations were conducted to ascertain the required sample size for each of two equal-sized groups. The effect size was established at 15% of the pooled standard deviations, estimated at 10% based on historical data from our group. This analysis was predicated on performing a two-tailed independent samples test, appropriate for testing without a predetermined direction. Power was validated using post hoc verification. Transplant recipient animals were randomly assigned at the time of irradiation, and donor cells were pooled from up to three animals. Ppia[+/−] animals were indistinguishable from wild-type animals in all experiments tested. Ppia[+/−] mice were generated by backcrossing into C57BL/6J mice for over ten generations. Transplantation studies are representative of two independent biological replicates.

C57BL/6-B6.SJL wild-type mice (CD45.1[+]) were lethally irradiated with a Cs137 source at a single dose of 9.5 Gy up to 24 h before transplantation.

**PB and BM cell analysis.** Cells were injected into the tail vein of C57BL/6-B6.SJL recipient mice in 100 μl of PBS; 375,000 nucleated BM cells of male or female C57BL/6J Ppia[+/−] or Ppia[−/−] mice (CD45.2[+]) were co-injected with the same number of CD45.1[+] competitor cells. PB chimerism was assessed at weeks 5, 8, 12 and 24 (shown are the 24-week analyses). Trendwise differences between Ppia knockout and heterozygous cells emerged at weeks 8 and 12 (P < 0.1) and became statistically significant at the 24-week analysis. Final BM collection occurred at week 28.

**Serial transplantations.** Equal numbers of BM cells (500,000 cells) from donor mice were mixed with 500,000 competitor cells from C57BL/6-B6.SJL wild-type mice and injected into lethally irradiated recipient mice. Two months after primary transplantation, 1,000,000 nucleated BM cells from the primary recipients were collected for a second time and again after two months for a third round of transplantation. Final evaluation was performed seven months after the third transplantation.

**Rescue experiments.** Lineage[−], c-Kit[+], Sca1[+] cells were pooled from two 18-month-old C57BL/6 male mice and divided into two groups for lentiviral transduction overnight. The collected cells were grown in tissue culture incubators and serum-free medium (StemSpan SFEM, Stem Cell Technologies), supplemented with murine thrombopoietin (TPO; 20 ng ml[−1], PeproTech), stem cell factor (SCF; 10 ng ml[−1], PeproTech) and the β-catenin agonist CHIR99021 (250 nM, Stemgent). Concentrated lentivirus (pLVX-EF1alpha, Takara Bio) encoding mouse Ppia ('rescue') or the reverse complement ('control') was added to the cells. The next day, cells were washed in PBS, and 5,000 transduced HSPCs were injected along with 500,000 competing total BM cells in the recipient's background into each irradiated recipient (four-month-old female C57BL/6-B6.SJL mice). The recipient mice were followed for six months with regular fluorescence-activated cell sorting (FACS) analysis of the PB to quantify reconstitution.

## Cell analysis and FACS

First, freshly isolated PB and BM were used for analysis. BM cells were initially depleted of lineage-positive cells with MACS LD columns (Miltenyi Biotec), as previously described[70]. The cells were then analysed with an LSR II instrument and isolated with an Aria I fluorescence-activated cell sorter (BD Biosciences).

The following antibody combinations were used for cell phenotyping: HSPC (c-Kit[+], lineage[−]), LKS (c-Kit[+], Sca1[+], lineage[−]), CMP (c-Kit[+], Sca1[−], lineage[−], CD16/32[−], CD34[+]), CLP (c-Kit[int.], Sca1[int.], lineage[−], CD127[+], CD34[+]) and HSC (c-Kit[+], Sca1[+], lineage[−], CD135[−], CD34[−], CD150[+]). Immunostainings were performed by incubating cells with anti-c-Kit (clone 2B8, BD Biosciences or Life Technologies), anti-Sca1 (clone D7, Caltag Medystems or Thermo Fisher Scientific), anti-CD16/32 (clone 93, eBioscience), anti-CD34 (clone RAM34, BD Biosciences), anti-CD135 (clone A2F10.1, BD Biosciences), anti-CD150 (clone TC15-12F12.2, BioLegend), anti-CD127 (clone SB/199, BioLegend) and anti-CD45.1/2 (clones A20 and 104, BioLegend) antibodies for 30 min (PB) or 60 min (BM) at 4 °C, before FACS analyses.

The antibodies used for lineage depletion were anti-CD11b (clone M1/70, BD Biosciences), anti-Ly-6G and Ly-6C (clone RB6-8C5, BD Biosciences), anti-CD8α (clone 53-6.7, BD Biosciences), anti-CD3ε (clone 145-2C11, BD Biosciences), anti-CD4 (clone GK1.5, BD Biosciences), anti-TER-199 (clone TER-119, BD Biosciences), anti-CD45R (clone RA3-6B2, BD Biosciences) and streptavidin (S32365, Thermo Fisher Scientific). The sources of the samples were blinded to the FACS analyst.

## Cell culture and drug treatments

Biochemical assays were performed in 293T or HeLa cells, which were maintained at 37 °C in a humidified incubator containing 5% CO$_2$. Cell lines were purchased from ATCC (293T CRL-3216; HeLa CCL-2) or DMSZ (NB4 ACC-207; OCI-AML3 ACC-582), cultured with the medium composition recommended by the supplier, and monitored for signs of infection, including mycoplasma contamination. The ATCC cell lines were confirmed by short tandem repeat profiling and human papillomavirus positivity (HeLa).

Stable 293T or HeLa control and PPIA Kd1/Kd2 cell lines were generated using pLKO.1 lentiviral vectors encoding short-hairpin RNAs targeting the human PPIA protein (clone ID TRCN0000049171 (Kd1) or clone ID TRCN0000049170 (Kd2), Horizon Discovery) designed by The RNAi Consortium (TRC). Cell lines stably transduced with a

pLKO.1-TRC empty vector encoding a non-targeting sequence (clone ID TRC TRCN0000241922, Horizon Discovery) served as controls. Following puromycin selection (2 µg ml⁻¹, Gibco, Fisher Scientific), *PPIA* knockdown efficiency was assessed by measuring PPIA protein expression by western blots in stably transduced cells (Extended Data Fig. 2d). The two constructs *PPIA* Kd1 and *PPIA* Kd2 showed >80% knockdown efficiency by immunoblot and were tested independently.

Control and *PPIA* Kd1 Hela cells were transfected with pcDNA3.1-PPIA vector or corresponding empty pcDNA3.1 control vector for 48 h. Following stress induction with sodium arsenite (50 µM, Sigma-Aldrich) for 1 h, immunostaining for G3BP1 protein, a marker of stress granule assembly, was performed using a rabbit polyclonal anti-G3BP1 antibody (cat. no. 13057-2-AP, Proteintech). The cells were mounted using Prolong gold antifade mounting medium containing 4′,6-diamidino-2-phenylindole (DAPI; Invitrogen) and were imaged at ×20 magnification on a Celldiscoverer 7 confocal microscope (Zeiss) operated with the ZEN Pro imaging software (Zeiss). The exposure time and gain were maintained at a constant level for all samples, and stress granule analysis was carried out with ImageJ software. Cell viability was measured on a Cellometer Auto 2000 automated cell counter with ViaStain acridine orange/propidium iodide staining solution (Nexcelom Bioscience). This staining solution discriminates live and dead nucleated cells using dual fluorescence. Cell viability was measured 1 h after stress induction with sodium arsenite.

Staining for P-bodies was performed in an identical manner in HeLa cells using an anti-DDX6 antibody (cat. no. 14632-1-AP, Proteintech). Visualization of nucleoli was conducted in OCI-AML3 cells (cat. no. ACC-582, DSMZ), following knock-in of mCherry into the endogenous NPM1 locus. *Z*-stacks were obtained at ×100 magnification (Celldiscoverer 7, Zeiss) in live cells following immobilization with Cell-Tak (Corning) at 37 °C in a humidified chamber with 5% $CO_2$. Before this, the cells were transduced and puromycin-selected for control or *PPIA* knockdown-vector 1 or treated for 24–48 h with PPIA inhibitor TMN355 (10 µM, Selleckchem). The size and roundness of nucleoli were quantified with Fiji[71].

**Generation of NPM1-mCherry knock-in OCI-AML3 cells.** *sgRNA design*. For gene knock-in experiments in acute myeloid leukaemia (AML) cell lines, we utilized previously described methods[72]. We determined a protospacer sequence for NPM1 near the C-terminal exon 12 using the CRISPRscan platform[73] and a 20-nt guide RNA sequence. Synthetic single-guide RNAs (sgRNAs) for homology directed repair were purchased from Synthego.

*Editing of NPM1 in OCI-AML3 cells*. A double-stranded DNA template encoding the mCherry protein for HDR (NPM1_mCherry) was synthesized as a linear DNA fragment (Twist Biosciences). The HDR template was designed with ~100-bp and ~200-bp homology arms, with the left homology arm designed from NPM1 intron 11 DNA sequence and the right homology arm from the NPM1 3′ UTR DNA sequence, respectively. Following polymerase chain reaction (PCR) amplification (KAPA HiFI HotStart ReadyMix; Roche) with HDR Primer F and Primer R (Millipore Sigma), the amplicons were purified with AMPure XP beads (Beckman Coulter). To generate the fusion protein, stop codons were replaced with a GSG linker followed by mCherry and a stop codon.

*Cas9-sgRNA pre-complexing and transfection*. To obtain Cas9-sgRNA RNPs, 1 µg of Cas9 protein (PNA bio) was incubated with 1 µg of synthetic sgRNA (Synthego) for 20 min at room temperature, then 1 µg of PCR-amplified HDR DNA was added to the RNP mixture. $2.50 × 10^5$ OCI-AML3 cells were electroporated in buffer R (Thermo Fisher) using the Neon Transfection System. The following electroporation conditions were used for OCI-AML3 cells: 1,400 V, 10 ms, three pulses.

**Sequences.** NPM1-mCherry:

TTAACTCTCTGGTGGTAGAATGAAAAATAGATGTTGAACTATGCAAAGAGACATTTAATTTATTGATGTCTATGAAGTGTTGTGGTTCCTTAACCACATTTCTTTTCTTTTTTTTTCCAGGCCATTCAGGACCTTTGGCAATGGAGAAAATCACTAGGAAGCGGAGTGAGCAAGGGCGAGGAGGATAACATGGCCATCATCAAGGAGTTCATGCGCTTCAAGGTGCACATGGAGGGCTCCGTGAACGGCCACGAGTTCGAGATCGAGGGCGAGGGCGAGGGCCGCCCCTACGAGGGCACCCAGACCGCCAAGCTGAAGGTGACCAAGGGTGGCCCCCTGCCCTTCGCCTGGGACATCCTGTCCCCTCAGTTCATGTACGGCTCCAAGGCCTACGTGAAGCACCCCGCCGACATCCCCGACTACTTGAAGCTGTCCTTCCCCGAGGGCTTCAAGTGGGAGCGCGTGATGAACTTCGAGGACGGCGGCGTGGTGACCGTGACCCAGGACTCCTCCCTGCAGGACGGCGAGTTCATCTACAAGGTGAAGCTGCGCGGCACCAACTTCCCCTCCGACGGCCCCGTAATGCAGAAGAAGACCATGGGCTGGGAGGCCTCCTCCGAGCGGATGTACCCCGAGGACGGCGCCCTGAAGGGCGAGATCAAGCAGAGGCTGAAGCTGAAGGACGGCGGCCACTACGACGCTGAGGTCAAGACCACCTACAAGGCCAAGAAGCCCGTGCAGCTGCCCGGCGCCTACAACGTCAACATCAAGTTGGACATCACCTCCCACAACGAGGACTACACCATCGTGGAACAGTACGAACGCGCCGAGGGCCGCCACTCCACCGGCGGCATGGACGAGCTGTACAAGTAG

HDR Primer F:

GTTCACATTTTTTATGACTGATTAAAGTGTTTGGAATTAAATTACATCTGAGTATAAATTTTCTTGGAGTCATATCTTTATCTAGAGTTAACTCTCTGGTGGTAGAATGAAAAATAGATGT

HDR Primer R:

TTCTCACTCTGCATTATAAAAAGGACAGCCAGATATCAACTGTTACAGAAATGAAATAAGACGGAAAATTTTTTAACAAATTGTTTAAACTATTTTCCTACTTGTACAGCTCGTCCATGC

sgNPM1wt:

UCCAGGCUAUUCAAGAUCUC

**Overview of cell type usage across figures.** Figures 1, 2, 5d and 6 are based on freshly isolated murine HSPCs and HSCs. Figures 3b–e and 4c show experiments with 293T cells. Figures 4e–g and 5a,b were performed in HeLa cells. Figure 5c was conducted with OCI-AML3 cells following knock-in of mCherry at the 3′ end of *NPM1*.

**Immunoprecipitations and western blots**
Immunoprecipitations of 3XF-PPIA and 3XF-Mutant PPIA transiently transfected cells were performed in 293T cells (one T175 flask per condition). 3XF-Mutant PPIA (G104A mutant) has reduced catalytic activity due to blocked substrate access to the active site[32]. At 48 h post transfection, 293T cells were washed twice with cold PBS and mechanically collected with a cell scraper. Cells were lysed using PBS buffer supplemented with 1% Triton X-100 and Xpert protease inhibitor cocktail (GenDepot) for 10 min, with end-over-end rotation. Cell lysates were centrifuged at 4,000*g* for 4 min at 4 °C. The supernatant was collected and immunoprecipitation was performed on the cytoplasmic fraction of the cells. For immunoprecipitation, the cell lysate was mixed with 4 µl of mouse monoclonal anti-FLAG antibody (clone M2, 1 µg µl⁻¹, Millipore Sigma) in a total volume of 900 µl for 1 h 30 min at 4 °C, then 50 µl of Protein G Dynabeads (Thermo Fisher) were added to the cell lysate for 3 h 30 min with end-over-end rotation. The beads were washed twice with 800 µl of ice-cold PBS for 5 min with end-over-end rotation at 4 °C, and the protein complexes were eluted with 50 µl of 3XFLAG peptide (Sigma-Aldrich) for 5 min at room temperature. Immunoblotting against a fraction of the lysate was used to validate that the expression levels of 3XF-PPIA and 3XF-Mutant PPIA were equal.

Immunoprecipitation against endogenous PPIA was performed in parental HeLa cells (two T175 flasks per condition). The cells were washed three times with ice-cold PBS, harvested mechanically, and pelleted at 1,200*g* for 5 min at 4 °C. The cells were lysed using IP lysis buffer (PBS supplemented with 1% Triton X-100 and 1% Xpert Protease Inhibitor Cocktail) and centrifuged at 4,000*g* for 5 min at 4 °C. The supernatant was collected and a fraction was stored at −80 °C for input

material. The lysate was then mixed with either 10 μg of anti-PPIA antibody (ab58144, Abcam) or normal mouse IgG$_1$ (IgG, immunoglobulin; Cell Signaling Technology) as a negative control in a total volume of 950 μl. Incubations with target antibody or isotype control were carried out for 1 h 15 min with end-over-end rotation at 4 °C, followed by incubation with 125 μl of Protein G Dynabeads overnight at 4 °C. The next day, the Protein G beads were washed once with 850 μl of ice-cold lysis buffer for 5 min with end-over-end rotation at 4 °C, followed by a similar wash with 850 μl of ice-cold PBS supplemented with 1% Xpert protease inhibitor cocktail. The protein complexes were eluted with 60 μl of elution buffer (50 mM Tris-HCl, pH 7.4, 1% SDS, 10 mM EDTA) by vortexing and applying three incubation cycles of 5 min at 65 °C.

Western blots were performed with a rat monoclonal anti-HA high-affinity antibody (clone 3F10, Millipore Sigma), a rabbit polyclonal anti-histone H3 antibody (ab1791, Abcam), a rabbit polyclonal anti-cyclophilin A antibody (2175, Cell Signaling Technology), a rabbit polyclonal anti-PABPC1 antibody (4992, Cell Signaling Technology), a rabbit polyclonal anti-DDX6 antibody (14632-1-AP, Proteintech), a rabbit polyclonal anti-G3BP1 antibody (13057-2-AP, Proteintech), a rabbit polyclonal anti-NPM1 antibody (10306-1-AP, Proteintech), a mouse monoclonal anti-β-tubulin antibody (86298, Cell Signaling Technology) and a mouse monoclonal anti-glyceraldehyde 3-phosphate dehydrogenase (anti-GAPDH) antibody (ab204481, Abcam).

### Pulsed SILAC

The workflow of the pulsed SILAC experiment performed in this study is described in Fig. 4a. First, control and *PPIA* Kd HeLa or 293T cells were cultured for five days in standard DMEM (containing light/unlabelled variants of lysine and arginine). Once the cells reached a similar confluence level (~50%), heavy isotope ($^{13}$C-$^{15}$N-lysine and $^{13}$C-$^{15}$N-arginine)-containing DMEM (Thermo Fisher Scientific) was added in excess to the cells for 24 h. The amino-acid concentrations were 0.46 mM L-lysine-2HCl and 0.47 mM L-arginine-HCl. Cells were collected and 100 μg of protein cell lysates from each cell type and condition were subjected to acetone precipitation, then denaturation, reduction and alkylation before overnight in-solution digestion at 37 °C with trypsin to generate peptides for MS. Digestions were terminated by adding an equal volume of 2% formic acid, and then desalted with Oasis HLB 1-ml reverse-phase cartridges (Waters) according to the vendor's protocol.

**LC–MS/MS analysis.** An aliquot of the tryptic digest was analysed by LC–MS/MS on an Orbitrap Fusion Tribrid mass spectrometer (Thermo Scientific) interfaced with an UltiMate 3000 Binary RSLCnano System (Dionex), as previously described[74]. In our experiments, dynamic exclusion was employed for 40 s.

**Data processing and analysis.** The raw proteomic files were processed with the Proteome Discoverer 1.4 software (Thermo Scientific), and the MS/MS spectra were searched against the UniProt Homo sapiens database using the SEQUEST HT search engine. The spectra were also searched against the decoy database using a peptide target FDR set to <1% and <5%, for stringent and relaxed matches, respectively. The search parameters allowed for a maximum of two missed trypsin cleavages, and the MS/MS tolerance was set at 0.6 Da. Carbamidomethylation on cysteine residues was used as a fixed modification, and oxidation of methionine as well as SILAC heavy arginine ($^{13}$C$_6$-$^{15}$N$_4$) and SILAC heavy lysine ($^{13}$C$_6$-$^{15}$N$_2$) were set as variable modifications. Quantification of SILAC pairs was performed with the Proteome Discoverer software. Precursor ion elution profiles of heavy versus light peptides were determined with a MS tolerance of 3 ppm. The area under the curve was used to determine a SILAC ratio for each peptide.

### De novo translation assay

Haematopoietic stem cells (c-Kit$^+$, Sca1$^+$, lineage$^-$, CD135$^-$, CD34$^-$) were collected from male and female C57BL6/J mice, four to six months of age, and expanded ex vivo using a previously published protocol[75]. Following 24-h treatment with PPIA inhibitor TMN355 (10 μM) or DMSO control, the translation rates were measured by microscopy with a fluorescent puromycin analogue (Click-IT Plus OPP, Alexa 488 picolyl azide, Thermo Fisher) following a 2-h pulse with the bio-orthogonal label according to the vendor's protocol. Quantification was performed at ×40 magnification (Celldiscoverer 7, Zeiss) using Fiji software[71].

### PLA

Whole BM was obtained from the hind-limb long bones and hip bones of young and old male C57BL/6J mice (five months old and 23 months old, respectively). Lineage-positive cells were isolated using the Direct Lineage Cell depletion kit (Miltenyi Biotec) and magnetically depleted with an AutoMACS Pro Separator (Miltenyi Biotec). The lineage-negative fraction was resuspended at a concentration of 10$^8$ cells per millilitre and stained on ice for 15 min with the combination of antibodies characterizing HSCs described in the 'Cell analysis and FACS' section. Cell sorting was carried out on an Aria II FACS instrument (BD Biosciences). Finally, isolated HSCs were cytospinned, attached onto a Cellview slide (543979, Greiner Bio-one) in the presence of Cell-Tak (Corning), and fixed in 4% paraformaldehyde.

To quantify PPIA expression, PLAs were performed on isolated HSCs with the Duolink in Situ Red Starter Kit Mouse/Rabbit (DUO92101, Millipore Sigma), adapting the vendor's protocol for HSCs. Briefly, HSCs were permeabilized with PBS + 0.5% Triton X-100 for 7 min, washed with PBS, and blocked in 5% donkey serum for 30 min at room temperature. After a short wash in PBS, the slides were incubated in a humidity chamber for 1 h at 37 °C with Duolink blocking solution. The primary antibodies (mouse anti-cyclophilin A antibody, ab58114, and rabbit anti-cyclophilin A antibody, ab41684; both from Abcam) were applied overnight at 4 °C in a humidity chamber. To quantify interactions between PPIA and its substrates, the mouse anti-cyclophilin A antibody was used in combination with rabbit anti-PABPC1, anti-DDX6 or anti-NPM1 antibodies (Proteintech 10970, 14632, 10306) in HSCs derived from mice, six to eight months of age. After washing the samples twice with Duolink buffer A, diluted anti-mouse PLUS and anti-rabbit MINUS PLA probes were added to the samples for 1 h at 37 °C in a pre-heated humidity chamber. Following two washes with buffer A, the cells were incubated with a DNA ligase previously diluted in Duolink ligation buffer for 30 min at 37 °C. The samples were washed twice in Duolink buffer A under gentle shaking, and incubated with a diluted DNA polymerase solution for 1 h 40 min at 37 °C in the dark. Finally, the slides were rinsed twice in 1× wash buffer B for 10 min and once in 0.01× wash buffer B for 1 min at room temperature and mounted with Duolink in situ mounting medium containing DAPI. For each antibody, a negative control experiment was performed where only one antibody or no antibody was incubated with the PLA probes. Fluorescence was visualized with a Celldiscoverer 7 confocal microscope (Zeiss) at ×100 magnification, and the images were processed to include background subtraction and orthogonal projection with ZEN Pro imaging software (Zeiss). The analyst was blinded to the origin of the samples during PLA staining and spot counting. An average of 90 cells per condition were counted, and the shown fluorescence microscopy images are representative of two independent biological replicates.

### Misfolded protein quantification

To quantify the relative abundance of misfolded protein aggregates in HeLa cells, we utilized a Proteostat Aggresome detection kit (ENZ-51035-K100, Enzo Life Sciences). The Proteostat aggresome detection assay was performed according to the manufacturer's instructions. Briefly, cells seeded on glass slides were washed with PBS, fixed with 4% formaldehyde for 30 min at room temperature, permeabilized (0.5% Triton X-100, 3 mM EDTA) for 30 min on ice under gentle shaking, and stained with Proteostat dye (1:20,000 dilution) for 1 h at room temperature. Nuclei were counterstained with DAPI.

Cells treated with 10 μM MG132 (proteasome inhibitor) for 16 h were used as a positive control. Samples stained with DAPI only served as a background control for Proteostat quantification. The cells were imaged with an Olympus Fluoview FV3000 confocal microscope with excitation/emission (Proteostat) = 488/632 nm and (DAPI) = 350/435 nm. Signal quantification was performed with Fiji software[71].

## RNA sequencing

For a young versus old comparison, wild-type HSPCs were isolated from the hind-limb long bones of male C57BL/6J mice, aged four to six months or 31–33 months, respectively. c-Kit+ cells were stained and magnetically isolated from the lineage-depleted cell suspension using the EasySep mouse CD117 (c-Kit) positive selection kit (Stem Cell Technologies), following the manufacturer's instructions. After overnight growth in serum-free medium (StemSpan SFEM, Stem Cell Technologies), supplemented with murine TPO (20 ng ml$^{-1}$, PeproTech), SCF (10 ng ml$^{-1}$, PeproTech) and the β-catenin agonist CHIR99021 (250 nM, Stemgent), HSPCs were collected as cell pellets. Immediately after collection, RNA extraction was carried out with the RNeasy Plus Mini kit with genomic DNA Eliminator columns (QIAGEN) in combination with on-column DNaseI digestion (QIAGEN), according to the vendor's protocol.

Total RNA-seq libraries were generated and prepared for multiplexing on the Illumina platform with the TruSeq stranded total RNA library prep (Illumina) according to the manufacturer's protocol. The libraries included ERCC ExFold RNA spike-in mixes (Thermo Fisher Scientific) to assess the platform dynamic range. RNA spike-in mixes confirmed high fidelity between two independent next-generation sequencing (NGS) runs ($R^2$ = 0.991 and 0.943, respectively; Supplementary Data 2). The resultant libraries were quality-checked on a Bioanalyzer 2100 instrument (Agilent) and quantified with a Qubit fluorometer (Thermo Fisher Scientific). Further quantification of the adapter ligated fragments and confirmation of successful P5 and P7 adapter incorporations were assessed with the KAPA universal library quantification kit for Illumina (Roche), run on a ViiA7 real-time PCR system (Applied Biosystems). Multiplexed and equimolarly pooled library products were re-evaluated on the Bioanalyzer 2100 and diluted to 18 pM for cluster generation by bridge amplification on the cBot system. The libraries were then loaded onto a rapid run mode flowcell v.2, followed by paired-end 100-cycle sequencing run on a HiSeq2500 instrument (Illumina). The PhiX Control v3 adapter ligated library (Illumina) was spiked-in at 2% by weight to ensure balanced diversity and to monitor clustering and sequencing performance. We obtained a minimum of 50 million reads per sample.

For the *Ppia* heterozygous versus knockout comparison, cells were isolated from mice, aged 10–12 months, as outlined above, and immediately subjected to RNA isolation (without overnight culture). Total RNA libraries were prepared using the SMARTer Stranded Total RNA-Seq kit v.2 (Takara Bio, 634418) and Unique Dual Index kit (Takara Bio, 634752). Paired-end sequencing was performed for 150 cycles using an Illumina NovaSeq 6000 system.

**Data processing.** Fastq file generation was achieved with the Illumina's BaseSpace Sequence Hub. Demultiplexing was based on sample-specific barcodes. All bioinformatic analyses were performed with Linux command line tools. After removing the short sequence reads that did not pass quality control and discarding reads containing adaptor sequences with Cutadapt v.1.12[76], sequence reads were assembled and mapped against the mouse MM9 reference genome (Genome Reference Consortium) with TopHat2/Bowtie2 v.2.1.0[77]. Gene expression changes were quantified with Cufflinks and Cuffdiff v.2.1.1[78], and data were normalized by calculating the fragments per kilobase per million mapped reads (FPKM). Analysis of murine RNA-seq data was validated by two independent biological replicates (young versus old) or three independent replicates (*Ppia* heterozygous versus knockout).

## Statistics

All statistical analyses were performed using Stata v.15.1 and GraphPad Prism v.10 software. Unsupervised hierarchical clustering was performed with Morpheus using default parameters, and gene set enrichment analyses were performed with GSEA v.4.3.2 based on gene set permutations[79,80]. The gene sets are available in Supplementary Data 8. Dashed lines mark medians and dotted lines represent the lower and upper quartiles in violin plots. Comparisons for MS were pruned for low-scoring peptides and rank-normalized. The treatment designation or cell genotype in microscopy-based or FACS analyses was blinded to the person performing quantification to reduce experimental bias. No data points or animals were excluded from the analysis of completed experiments. Randomization was not feasible within the experimental design. We did not formally test data for normality and homoscedasticity; however, we employed the Wilcoxon rank-sum test, which utilizes ranks rather than actual values, allowing robust statistical calculations even when distributions are skewed and variances are unequal.

## Software

All open-source and commercial software and proteomic databases used to analyse MS/MS data, RNA-sequencing data and microscopy pictures are described in the Methods. All statistical analyses were performed using Stata v.15.1 and GraphPad Prism v.10. Image analysis was performed with Fiji/ImageJ 2.00/1.52p and ZEN Pro 3.1. The 2D gel captures were analysed with DeCyder 7.0 and ImageQuant (GE Healthcare). Pulsed SILAC data were analysed with Qlucore Omics Explorer 3.5 software. FACS data were analysed with FlowJo v.10. Gene set enrichment analyses were performed with GSEA v.4.3.2. The 3D molecular structure of the PPIA protein was visualized with PyMOL v.2.5.2 (licensed by A.C.). Figures 1c,g, 2a, 4a and 6d were created with BioRender.com (licensed by L.M.).

### Reporting summary

Further information on research design is available in the Nature Portfolio Reporting Summary linked to this Article.

## Data availability

All data necessary for interpreting, verifying and extending the research in this Article have been co-submitted as Source Data and Supplementary Information files. Raw data have been deposited in the repositories outlined below and are available without restrictions. Mass spectrometry data obtained after 3XF-PPIA immunoprecipitation (Fig. 3c) are deposited with the ProteomeXchange Consortium via the MassIVE repository (MSV000083867) with the dataset identifier PXD014025 (https://massive.ucsd.edu/). The datasets generated in the mouse HSC proteome profiling (Extended Data Fig. 4b) have been deposited with the ProteomeXchange Consortium via the MassIVE repository (MSV000083845) with the dataset identifier PXD013995. Proteome data following isobaric labelling and pulsed SILAC are available as Supplementary Information and at MassIVE (MSV000093125, MSV000093126, MSV000093127) and ProteomeXchange (PXD046245, PXD046246, PXD046247). For the transcriptomic analysis of murine HSPCs (Supplementary Data 2), raw and processed RNA-seq data have been deposited with the Gene Expression Omnibus database under accession code GSE151125. Previously published proteome data that were re-analysed are available under accession codes PXD007048 (proteome data) and GSE115353 (transcriptomics). The PPIA structure was previously submitted to the Research Collaboratory for Structural Bioinformatics Protein Data Bank under accession code 7ABT. PhaSepDB2.0 contains a database of 593 phase-separation proteins and 7,679 membrane-less organelle entries (http://db.phasep.pro/)[81], curated from literature and databases. IUPred2A was used to compute the likelihood of structural disorder per residue (https://iupred2a.elte.hu/)[68,69]. Heatmaps in Extended Data Fig. 5 were computed with Morpheus (https://software.broadinstitute.org/morpheus).

The members of the Cyclophilin family depicted in Extended Data Fig. 1c are PPIL6 (NP_775943.1), NKTR (NP_005376.2), PPIG (NP_004783.2), PPIE (NP_006103.1), PPIH (NP_006338.1), PPID (NP_005029.1), RANBP2 (NP_006258.3), PPIA (NP_066953.1), PPIF (NP_005720.1), PPIB (NP_000933.1), PPIC (NP_000934.1), SDCCAG-10 (NP_005860.2), PPIL1 (NP_057143.1), PPIL2 (NP_055152.1), PPIL3 (NP_115861.1), PPIL4 (NP_624311.1) and PPWD1 (NP_056157.1). All other data supporting the findings of this study are available from the corresponding author on reasonable request. Source data are provided with this paper.

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

## Acknowledgements

A.C. was supported by the Cancer Prevention and Research Institute of Texas (CPRIT-RR140038), the Ted Nash Long Life Foundation, the American Society of Hematology (Bridge grant 2023-20) and the National Institutes of Health (NIH) (grants NIH R01DK115454 and NIH R01GM142143). L.M. is the recipient of an NIDDK T32 Hematology Training Grant (2T32DK060445). We also thank J. Ino Hsu for helpful discussions, C. Gillespie for editorial assistance, and L. Prickett-Rice, K. Folz-Donahue, J.M. Sederstrom and A. White for FACS support. We acknowledge the Genomic and RNA Profiling Core (supported by NIH-NIDDK P30DK56338 Center grant, NIH-NCI P30CA125123 Center grant and NIH 1S10OD02346901 S10 grant), the Cancer Genomics Center at the University of Texas (CPRIT-RP180734), the Cytometry and Cell Sorting Core (supported by CPRIT Core Facility Support Award (CPRIT-RP180672) and NIH (CA125123 and RR024574) grants) and the Mass Spectrometry Proteomics Core (supported by NIH-NCI P30CA125123 Center grant and CPRIT Core Facility Support Award (CPRIT-RP170005)) at Baylor College of Medicine for their technical support. We thank S. Jung and A. Jain at the BCM Mass Spectrometry Proteomics Core for their assistance with '365' profiling and MS/MS data analysis, L. Li and S. Pan at the Clinical and Translational Proteomics Service Center, D. Kraushaar at the RNA Profiling Core, and X. Chen at the Cancer Genomics Center of the University of Texas Health Science Center for their expert help with the generation and analysis of the pulsed SILAC and TMT MS data, as well as with RNA sequencing.

## Author contributions

L.M. designed and conducted molecular assays, interpreted results and wrote the manuscript. P.I. conducted molecular assays and interpreted results. S.E.M. helped with the acquisition of microscopy images. C.G.L., X.L. and G.K.D. supported in vitro experiments. D.B.S., C.T.H., D.S.K. and B.S. helped with in vitro assays and functional haematopoietic assays and edited the manuscript. E. Spooner and J.C.K.K. were involved in protein identification. B.C.B. supported the in vivo studies and discussions, interpreted data and edited the manuscript. B.D.S., X.C., F.T.F.T., M.A.G. and E. Sahin supported data analysis and edited the manuscript. D.T.S. co-supervised the haematopoietic aspects of this study, interpreted data and edited the manuscript. A.C. supervised this study and was involved in all experimental aspects, conceived the project, analysed data and wrote the manuscript.

## Competing interests

The authors declare no competing interests.

## Additional information

**Extended data** is available for this paper at https://doi.org/10.1038/s41556-024-01387-x.

**Correspondence and requests for materials** should be addressed to André Catic.

**a**

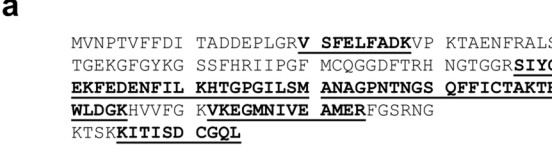

```
MVNPTVFFDI TADDEPLGRV SFELFADKVP KTAENFRALS
TGEKGFGYKG SSFHRIIPGF MCQGGDFTRH NGTGGRSIYG
EKFEDENFIL KHTGPGILSM ANAGPNTNGS QFFICTAKTE
WLDGKHVVFG KVKEGMNIVE AMERFGSRNG
KTSKKITISD CGQL
```

**b**

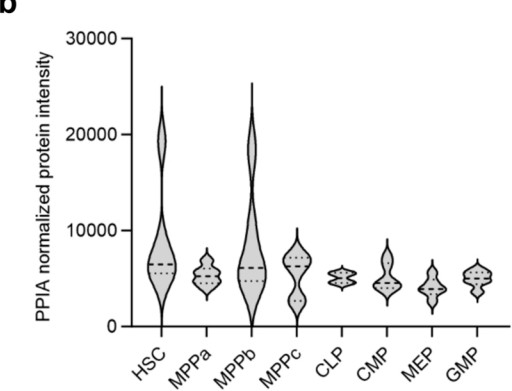

**c**

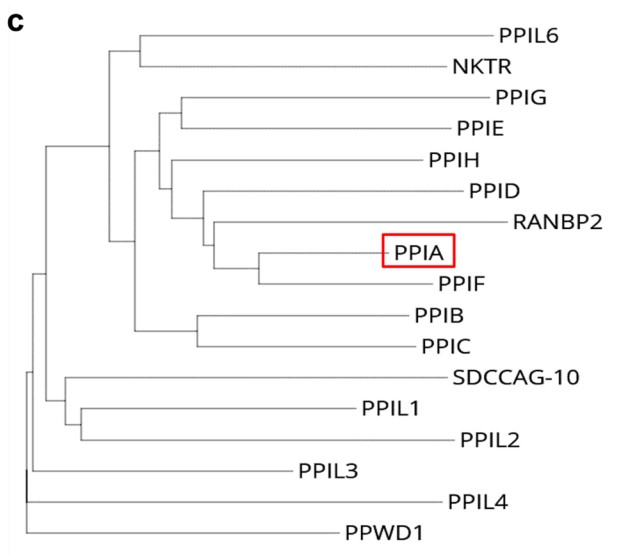

**d**

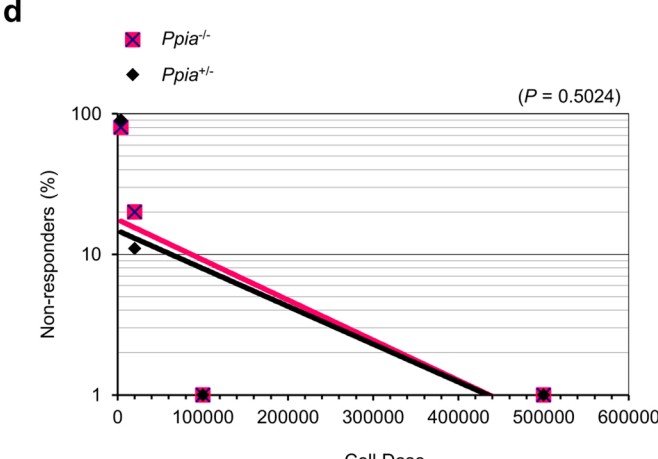

**Extended Data Fig. 1 | PPIA mass spectrometry and limiting dilution assay. a**, Identification coverage of PPIA. Spots were excised from 2-D SDS PAGE for extraction and trypsin digestion. MS/MS spectra cover 49.8% of the entire PPIA protein sequence (underlined; representative of two independent mass spectrometry identifications). **b**, Analysis of proteomic data in the haematopoietic compartment based on Zaro et al.[16]. PPIA is robustly expressed in haematopoietic stem cells (HSCs), but also expressed in progenitor compartments throughout haematopoiesis. Violin plots represent PPIA abundance in six replicate animals. MPP, multipotent progenitors; CLP, common lymphoid progenitors; CMP, common myeloid progenitors; MEP, megakaryocyte-erythroid progenitors; GMP, granulocyte-monocyte progenitors. **c**, Phylogenetic tree of the Cyclophilin protein family in humans. The tree is based on protein sequence alignments from the NCBI RefSeq database with Clustal Omega. The red box indicates PPIA. Accession numbers

for the individual proteins are: PPIL6 (NP_775943.1), NKTR (NP_005376.2), PPIG (NP_004783.2), PPIE (NP_006103.1) PPIH (NP_006338.1), PPID (NP_005029.1), RANBP2 (NP_006258.3), PPIA (NP_066953.1), PPIF (NP_005720.1), PPIB (NP_000933.1), PPIC (NP_000934.1), SDCCAG-10 (NP_005860.2), PPIL1 (NP_057143.1), PPIL2 (NP_055152.1), PPIL3 (NP_115861.1), PPIL4 (NP_624311.1), and PPWD1 (NP_056157.1). **d**, Limiting dilution transplantations of *Ppia* heterozygous (*Ppia*$^{+/-}$) and knockout (*Ppia*$^{-/-}$) bone marrow. 500,000 competitor cells (CD45.1$^+$) were co-injected with 4,000, 20,000, 100,000, or 500,000 nucleated bone marrow cells of *Ppia*$^{+/-}$ or *Ppia*$^{-/-}$ mice. Reconstitution of peripheral CD45.2$^+$ cells was assayed 20 weeks after transplantation and differences were compared using a two-tailed Poisson *t*-test. No significant difference exists between *Ppia* heterozygous and knockout donors; (a representative example of two independent experiments is shown with *n* = 5 animals per group for the two high donor cell doses and *n* = 10 animals per group for the two low donor cell doses).

**a**

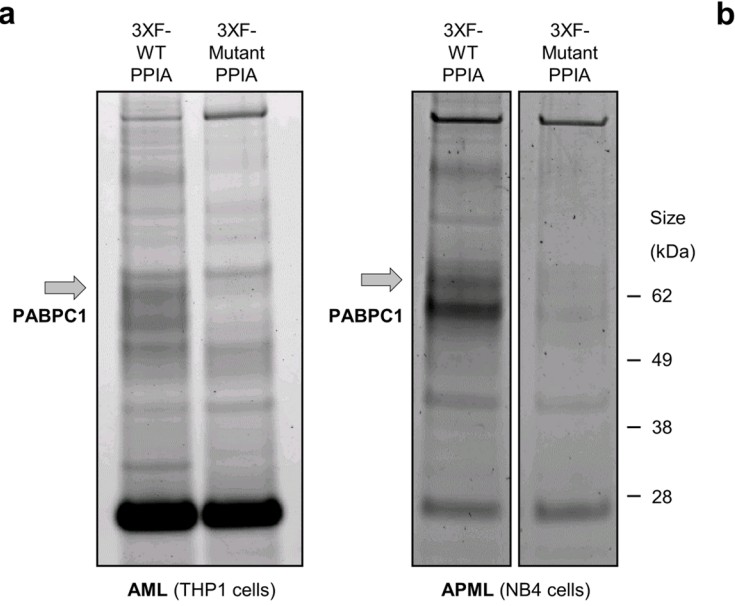

**b**

```
MNPSAPSYPM ASLYVGDLHP DVTEAMLYEK FSPAGPILSI
RVCRDMITRR SLGYAYVNFQ QPADAERALD TMNFDVIKGK
PVRIMWSQRD PSLRKSGVGN IFIKNLDKSI DNKALYDTFS
AFGNILSCKV VCDENGSKGY GFVHFETQEA AERAIEKMNG
MLLNDRKVFV GRFKSRKERE AELGARAKEF TNVYIKNFGE
DMDDERLKDL FGKFGPALSV KVMTDESGKS KGFGFVSFER
HEDAQKAVDE MNGKELNGKQ IYVGRAQKKV ERQTELKRKF
EQMKQDRITR YQGVNLYVKN LDDGIDDERL RKEFSPFGTI
TSAKVMMEGG RSKGFGFVCF SSPEEATKAV TEMNGRIVAT
KPLYVALAQR KEERQAHLTN QYMQRMASVR AVPNPVINPY
QPAPPSGYFM AAIPQTQNRA AYYPPSQIAQ LRPSPRWTAQ
GARPHPFQNM PGAIRPAAPR PPFSTMRPAS SQVPRVMSTQ
RVANTSTQTM GPRPPAAAAA ATPAVRTVPQ YKYAAGVRNP
QQHLNAQPQV TMQQPAVHVQ GQEPLTASML ASAPPQEQKQ
MLGERLFPLI QAMHPTLAGK ITGMLLEIDN SELLHMLESP
ESLRSKVDEA VAVLQAHQAK EAAQKAVNSA TGVPTV
```

**c**

| ACTL6A | MTA1 | SSRP1 | XRCC5 | DDX1 | SFPQ | C14orf166 |
|--------|------|-------|-------|------|------|-----------|
| CDC5L | PARP1 | SUB1 | | DDX17 | YBX1 | EEF1A1 |
| CREB1 | PML | SUPT16H | | DDX21 | | hnRNPUL1 |
| CTBP1 | PNN | TADA2B | | DDX5 | | ILF3 |
| CTBP2 | RBBP4 | TFAM | | G3BP1 | | KHSRP |
| FOXC1 | RBBP5 | TFCP2 | | hnRNPDL | | RBMX |
| GTF2I | RBBP7 | TP53 | | hnRNPK | | |
| HDAC1 | RUVBL1 | TRIM28 | | ILF2 | | |
| HDAC2 | RUVBL2 | UBP1 | | NACA | | |
| HIC2 | SND1 | UBTF | | NONO | | |
| DNA-binding proteins | | | | DNA- & RNA-binding proteins | | RNA-binding proteins |

**d**

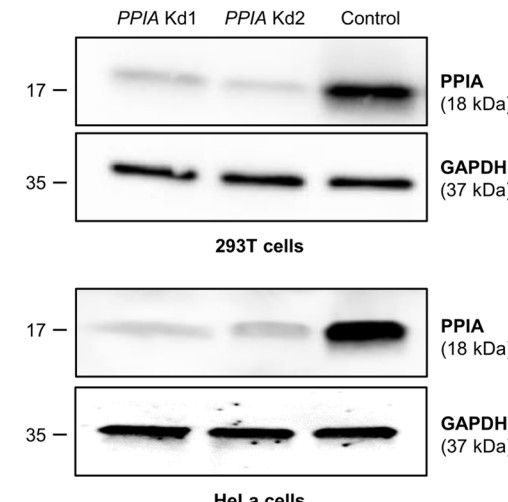

**e**

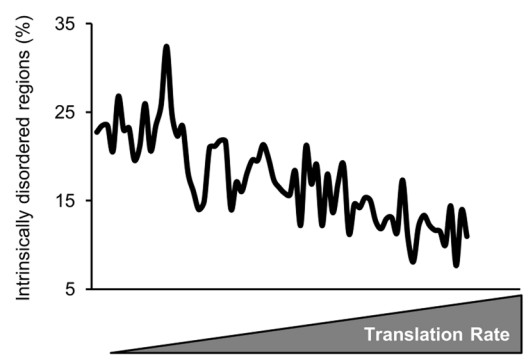

**Extended Data Fig. 2 | See next page for caption.**

**Extended Data Fig. 2 | PPIA functional validation experiments. a**, Interaction between PPIA and PABPC1 in haematopoietic cells. Co-IP followed by MS/MS identifies PABPC1 as an interactor of PPIA in the human haematopoietic cell lines THP1 (acute monocytic leukemia, AML) and NB4 (acute promyelocytic leukemia, APML). Cells were transduced with 3XF-tagged PPIA (3XF-WT-PPIA or 3XF-Mutant(G104A)-PPIA, respectively) and IP was performed with an anti-3XFLAG antibody. The arrow indicates the band for PABPC1 protein; (representative of two independent experiments shown). **b**, Identification coverage of PABPC1. Co-immunoprecipitated bands were excised after SDS-PAGE (Fig. 2b), trypsin digested, and identified by MS/MS. Coverage for PABPC1 extends to 26.4% of the protein, including the N-terminal RNA-binding domain, the C-terminal domain interacting with cap-binding proteins, and the unstructured linker region; (representative of two independent experiments shown). **c**, List of nucleotide-binding PPIA client proteins. Proteins listed by their official gene names. Nucleotide binding was determined based on UniProt classifications. **d**, Efficiency of *PPIA* knockdown in 293T cells and HeLa cells. Cells were stably transduced with pLKO.1-TRC control, TRC *PPIA* Kd1, or TRC *PPIA* Kd2 lentiviral vectors, respectively. Then, cell lysates were prepared and loaded onto a SDS-PAGE in order to measure PPIA protein expression by westernblot using a rabbit polyclonal anti-PPIA antibody. Glyceraldehyde 3-phosphate dehydrogenase (GAPDH) was used as a loading control for protein normalization; (representative of three independent experiments shown). **e**, Presence of intrinsically disordered regions correlates with a slower translation rate. Reanalysis of data from Schwanhäusser et al.[35] showing the inverse correlation between protein translation speed and percentage of intrinsically disordered regions[36] in the whole proteome. Proteins with more intrinsically disordered regions translate at a slower rate than structured proteins. Details on the calculation of protein synthesis rates have been described by the original authors[35].

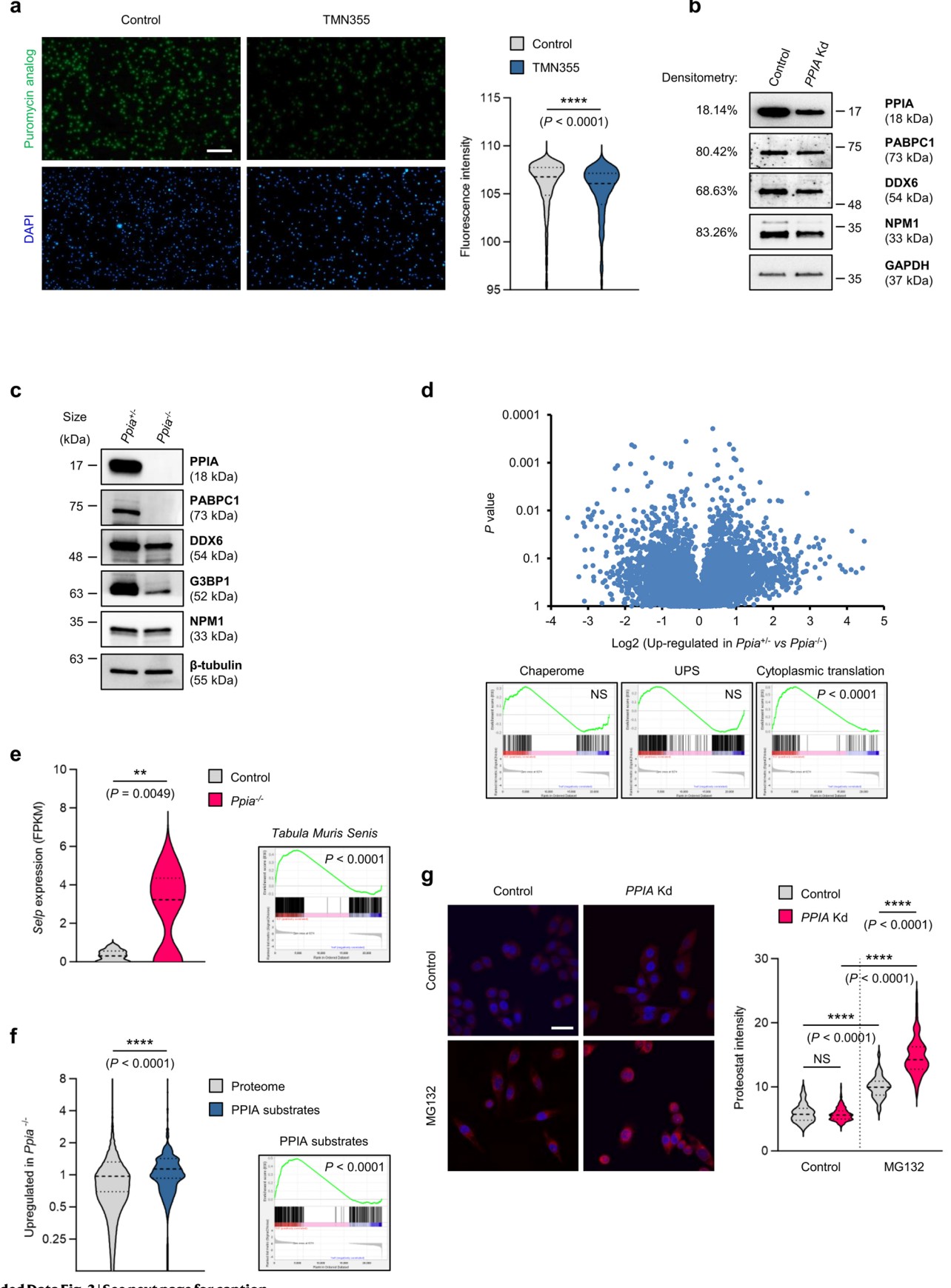

**Extended Data Fig. 3 | See next page for caption.**

**Extended Data Fig. 3 | PPIA's effect on transcription and translation in haematopoietic cells. a**, Impaired translation in haematopoietic stem and progenitor cells following pharmacological PPIA inhibition. Haematopoietic stem cells (lin⁻/cKit⁺/Sca1⁺/CD34⁻/CD135⁻) were isolated and expanded *in vitro* as previously published[75]. Translation rates were measured through bio-orthogonal labeling with a fluorescently labeled puromycin analog for two hours in cells that were pre-treated with DMSO or PPIA inhibitor TMN355 (10 µM, 24 h). Shown is a representative of three independent biological replicates, in which $n = 1,880$ (DMSO) and $n = 1,303$ (TMN355) cells were analysed and fluorescence was measured per cell. Scale bar, 100 µm (****$P < 0.0001$, two-sided Wilcoxon rank-sum test). Bottom panel depicts DAPI counterstain. **b**, Reduced expression of PPIA substrates in OCI-AML3 cells following *PPIA* knockdown. Western blot analyses to detect protein expression of PPIA and PPIA protein partners PABPC1, DDX6, and NPM1 in the OCI-AML3 cell line. GAPDH was used as a loading control for protein normalization and densitometry was measured relative to GAPDH expression. The images represent results of two independent experiments. **c**, Protein expression of PPIA client proteins is decreased in *Ppia*⁻/⁻ HSPCs. Western blot analyses to detect protein expression of PPIA and PPIA protein partners PABPC1, DDX6, G3BP1, and NPM1 in mouse lineage-depleted bone marrow cells. The images represent results of *Ppia*⁺/⁻ and *Ppia*⁻/⁻ animals ($n = 1$ each). β-tubulin was used as a loading control for protein normalization. This experiment was performed once. **d**, *Ppia* knockout versus heterozygous haematopoietic stem and progenitor cells (lin⁻/c-Kit⁺) upregulate genes involved in translation. Volcano plot shows comparable up- and down-regulation of genes. Gene set enrichment analysis of haematopoietic stem and progenitor cells of *Ppia* knockout or heterozygous animals. $N = 3$ independent animals were analysed per group. Genes encoding the entire mouse chaperome or the ubiquitin-proteasome system (UPS) were not significantly increased in knockout cells. However, the gene ontology 'cytoplasmic translation' was significantly upregulated in knockout cells. **e**, *Ppia* knockout cells show the transcriptional signature of aging. Haematopoietic stem and progenitor cells (lin⁻/cKit⁺) of three *Ppia* knockout animals compared to three heterozygous animals show significant upregulation of the aging marker gene P-selectin[45]. In addition, we observed a strong gene set enrichment resembling aged haematopoietic stem cells[44]. Statistics derived using two-sided Wilcoxon rank-sum test ($n = 3$ mice per group). **f**, Genes encoding PPIA substrates are upregulated in *Ppia* knockout haematopoietic stem and progenitor cells. Relative FPKM changes shown in cumulative violin plots representing three animals per genotype (*Ppia* knockout versus heterozygous); y-axis represented in log2. Gene set enrichment analysis of haematopoietic stem and progenitor cells (lin⁻/c-Kit⁺) of *Ppia* knockout or heterozygous animals shows significant upregulation of PPIA substrates ($n = 307$) compared to the overall proteome ($n = 6,114$) in knockout animals. Statistics derived using two-sided Wilcoxon rank-sum test. **g**, No increased spontaneous aggregation of misfolded proteins in absence of PPIA. Left panel: Protein misfolding was quantified using the molecular rotor ProteoStat with affinity for aggregated proteins. Increased protein aggregation causes the dye to stop spinning and emit fluorescence. We analysed misfolding in HeLa cells transduced with either scramble lentivirus or following *PPIA* knockdown, and used proteasome inhibition as positive control (MG132, 10 µM, 16 h). Scale bar, 80 µm. Right panel: Relative mean intensity per cell was plotted and calculated after blinding. A total of 8 independently treated replicates were analysed per group in this assessment. Statistics calculated using two-sided Wilcoxon rank-sum test ($n = 154$ control cells, $n = 147$ *PPIA* knockdown cells for, $n = 75$ for MG132-treated control cells, and $n = 86$ for MG132-treated *PPIA* knockdown cells; the experiment was performed once).

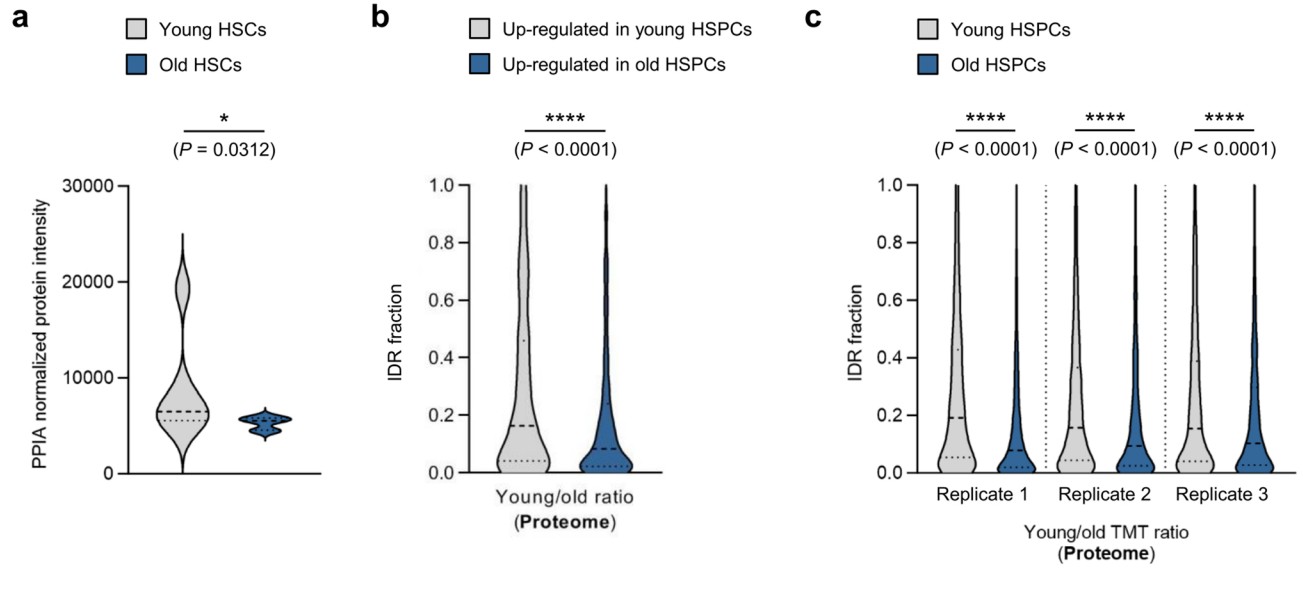

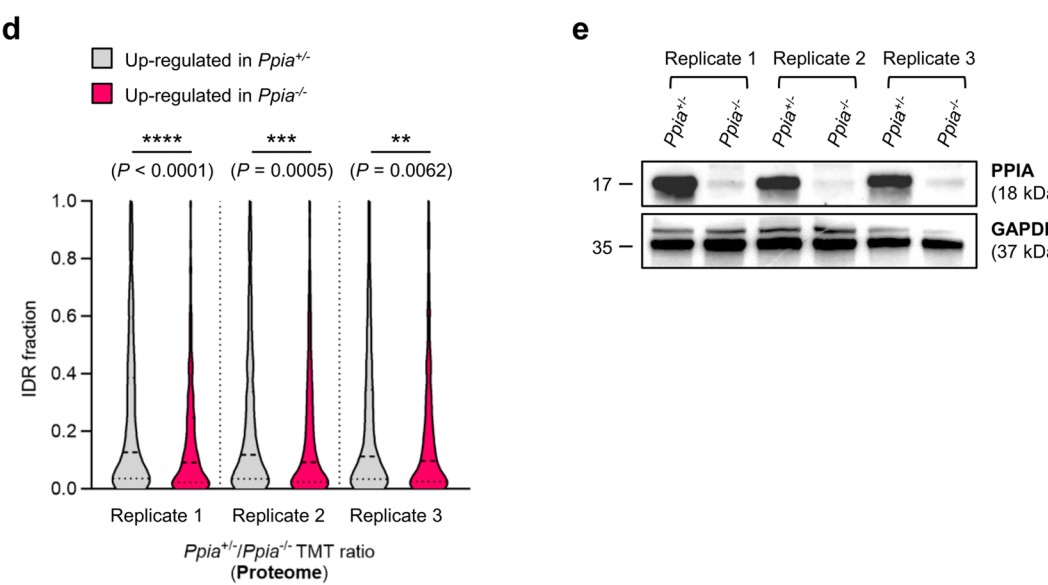

**Extended Data Fig. 4 | Mass spectrometry in the haematopoietic compartment. a**, Reanalysis of proteomic data in haematopoietic stem cells based on Zaro et al.[16]. PPIA is significantly downregulated in aged haematopoietic stem cells (HSC). Violin plots represent PPIA abundance in n = 6 replicate animals. Statistics calculated using two-sided Wilcoxon rank-sum test (*P < 0.05).
**b**, Quantification of IDR content in the top quartiles of proteins upregulated in the young and old mouse HSPC proteome by label-free MS/MS, respectively. Analysis of MS/MS data following '365' proteome profiling was validated by a total of two independent biological replicates. ****P < 0.0001, determined by two-sided Wilcoxon rank-sum test (n = 1,703 proteins per sample analysed).
**c**, Aged HSPCs show lower levels of intrinsic disorder in their proteome. Shown are individual comparisons of three independent replicates with the top upregulated proteins, separated by median, in either young or old cells. Quantitative mass spectrometry was performed following isobaric labelling. ****P < 0.0001;

two-sided Wilcoxon rank-sum test (n = 4,789, n = 4,789, and n = 4,706 proteins per sample analysed). **d**, Ppia knockout results in lower levels of proteome disorder. Shown are individual comparisons of three independent replicates with the top upregulated proteins, separated by median, in either Ppia heterozygous (equivalent to wild type) or knockout cells. Quantitative mass spectrometry was performed following isobaric labelling. **P < 0.01, ***P < 0.001, ****P < 0.0001; two-sided Wilcoxon rank-sum test (n = 4,788, n = 4,453, and n = 4,637 proteins per sample analysed). **e**, Validation of Ppia knockout efficiency in the sets of Ppia+/- and Ppia-/- mice used for TMT MS/MS experiments shown in Fig. 6c. Livers were homogenized with RIPA buffer and cell lysates were loaded onto a SDS-PAGE gel in order to measure PPIA protein expression by western blot using a rabbit polyclonal anti-PPIA antibody. Glyceraldehyde 3-phosphate dehydrogenase (GAPDH) was used as a loading control for protein normalization.

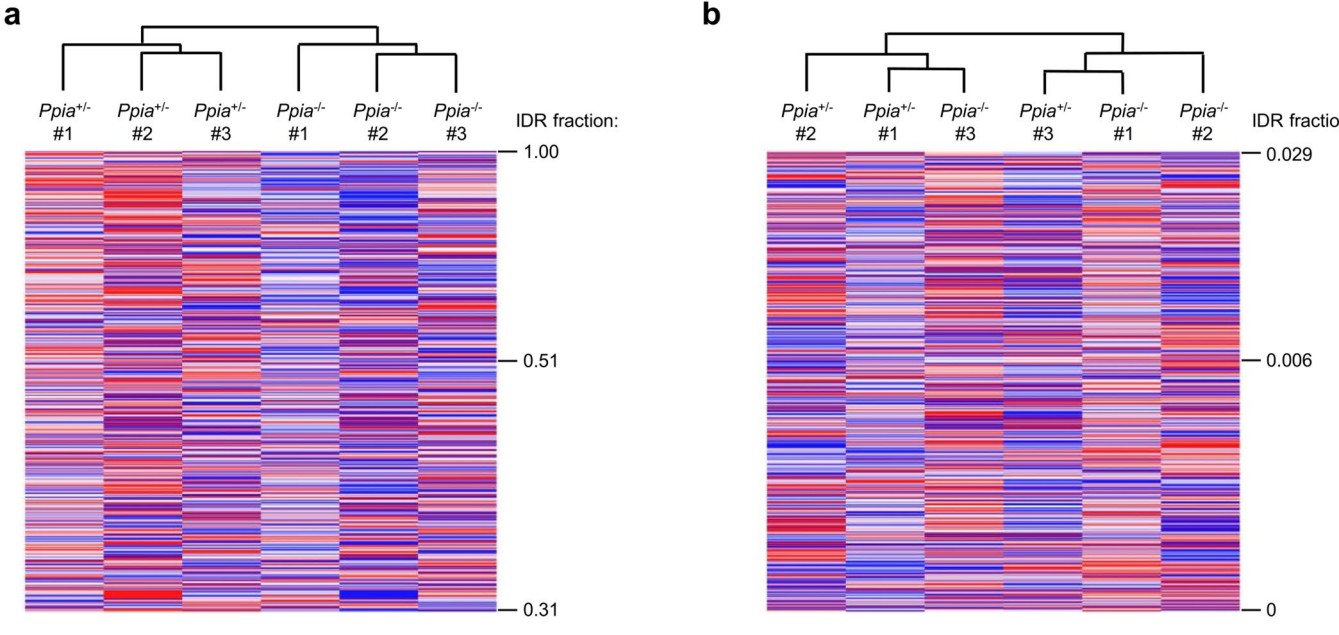

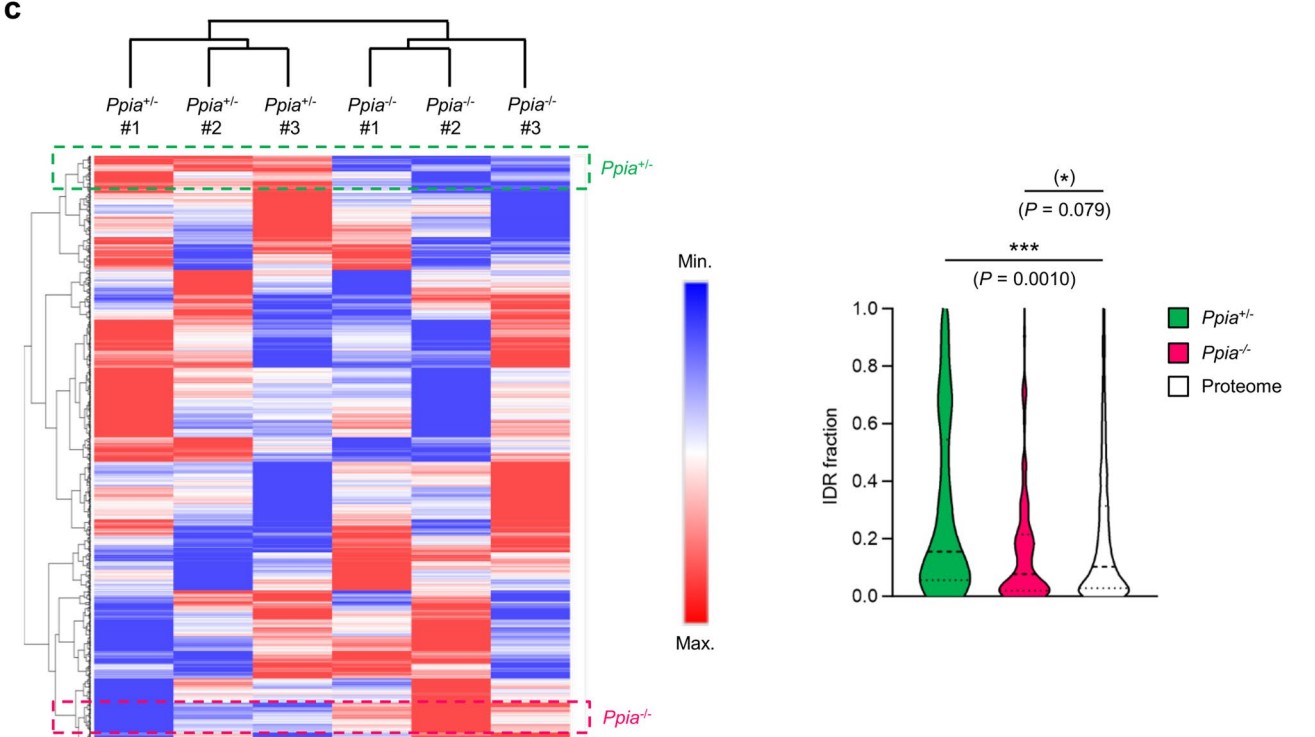

**Extended Data Fig. 5 | Clustering of proteomic data. a**, PPIA expression affects highly disordered proteins more than proteins with low levels of disorder. Proteins from haematopoietic stem and progenitor cells (lin⁻) of three *Ppia* heterozygous or knockout animals were quantified by mass spectrometry following isobaric labeling and sorted from high levels to low levels of disorder (top to bottom). The top quartile of proteins with high degrees of intrinsic disorder cluster according to genotype of the origin cells after unbiased hierarchical cluster analysis. The degree of disorder of the top (100%), median (51%), and bottom protein (31%) within the highest quartile is depicted. **b**, IDR-poor proteins are less dependent on PPIA. Proteins at the bottom quartile of disorder fail to cluster by genotype, indicating that PPIA expression mostly

affects highly disordered proteins. The degree of disorder of the top (2.9%), median (0.6%), and bottom protein (0%) within the lowest quartile is depicted. **c**, Structural disorder correlates with PPIA expression. Performing a reciprocal analysis, we conducted unbiased hierarchical clustering by genotype and protein disorder. The top cluster with consistent upregulation of proteins in PPIA expressing cells (marked with green box) showed significantly higher levels of structural disorder compared to the total proteome (*$P < 0.05$, ***$P < 0.001$, two-sided Wilcoxon rank-sum test). The protein cluster with consistent upregulation in PPIA-deficient cells (marked with red box) showed a trend for lower levels of intrinsic disorder compared to the total proteome ($P = 0.079$; three independent animals were analysed per genotype).

# Reporting Summary

## Statistics

For all statistical analyses, confirm that the following items are present in the figure legend, table legend, main text, or Methods section.

| n/a | Confirmed | |
|---|---|---|
| ☐ | ☒ | The exact sample size (*n*) for each experimental group/condition, given as a discrete number and unit of measurement |
| ☐ | ☒ | A statement on whether measurements were taken from distinct samples or whether the same sample was measured repeatedly |
| ☐ | ☒ | The statistical test(s) used AND whether they are one- or two-sided<br>*Only common tests should be described solely by name; describe more complex techniques in the Methods section.* |
| ☒ | ☐ | A description of all covariates tested |
| ☒ | ☐ | A description of any assumptions or corrections, such as tests of normality and adjustment for multiple comparisons |
| ☐ | ☒ | A full description of the statistical parameters including central tendency (e.g. means) or other basic estimates (e.g. regression coefficient) AND variation (e.g. standard deviation) or associated estimates of uncertainty (e.g. confidence intervals) |
| ☐ | ☒ | For null hypothesis testing, the test statistic (e.g. *F*, *t*, *r*) with confidence intervals, effect sizes, degrees of freedom and *P* value noted<br>*Give P values as exact values whenever suitable.* |
| ☒ | ☐ | For Bayesian analysis, information on the choice of priors and Markov chain Monte Carlo settings |
| ☒ | ☐ | For hierarchical and complex designs, identification of the appropriate level for tests and full reporting of outcomes |
| ☒ | ☐ | Estimates of effect sizes (e.g. Cohen's *d*, Pearson's *r*), indicating how they were calculated |

*Our web collection on statistics for biologists contains articles on many of the points above.*

## Software and code

Policy information about availability of computer code

| | |
|---|---|
| Data collection | BD FACS DiVa software 9.0 was used to acquire flow cytometry data. Commercial softwares used to collect MS/MS data, RNA sequencing data, and capture microscopy pictures are indicated in the Methods section of the manuscript. No custom algorithm or custom software was used. |
| Data analysis | All open source and commercial software and proteomic databases used to analyse MS/MS data, RNA sequencing data and microscopy pictures are described in the Methods section of the manuscript. All statistical analyses were performed using Stata v.15.1 and GraphPad Prism 10. Image analysis was performed with FIJI/ImageJ 2.00/1.52p and ZenPro 3.1. 2D gel captures were analysed with DeCyder 7.0 and ImageQuant(GE Healthcare). Pulsed SILAC data was analysed with Qlucore Omics Explorer 3.5 software. FACS data was analysed with FlowJo version 10. Gene Set Enrichment Analyses were performed with version 4.3.2. The 3-D molecular structure of PPIA protein was visualized with PyMOL v.2.5.2 (licensed by A.C.). Rendered graphics were created with BioRender (licensed by L.M.). |

For manuscripts utilizing custom algorithms or software that are central to the research but not yet described in published literature, software must be made available to editors and reviewers. We strongly encourage code deposition in a community repository (e.g. GitHub). See the Nature Portfolio guidelines for submitting code & software for further information.

## Data

Policy information about <u>availability of data</u>

All manuscripts must include a <u>data availability statement</u>. This statement should provide the following information, where applicable:

- Accession codes, unique identifiers, or web links for publicly available datasets
- A description of any restrictions on data availability
- For clinical datasets or third party data, please ensure that the statement adheres to our <u>policy</u>

All data necessary for interpreting, verifying, and extending the research in this article have been co-submitted as source data and supplementary information files. Raw data have been deposited in the repositories outlined below and are available without restrictions.

Mass spectrometry data obtained after 3XF-PPIA immunoprecipitation (Fig. 3c) are deposited with the ProteomeXchange Consortium via the MassIVE repository (MSV000083867) with the dataset identifier PXD014025 (https://massive.ucsd.edu/). The datasets generated in the mouse HSC proteome profiling (Extended Data Fig. 4b) have been deposited to the ProteomeXchange Consortium via the MassIVE repository (MSV000083845) with the dataset identifier PXD013995. Proteome data following isobaric labelling and pulsed SILAC is available as supplementary information and at MassIVE (MSV000093125, MSV000093126, MSV000093127) and ProteomeXchange (PXD046245, PXD046246, PXD046247).

For transcriptomic analysis of murine HSPCs (SI-Mouse HSPCs RNA seq), raw and processed RNA-seq data have been deposited with the Gene Expression Omnibus (GEO) database under accession code GSE151125.

Previously published proteome data that were re-analysed are available under accession codes PXD007048 (proteome data) and GSE115353 (transcriptomics).

The PPIA structure was previously submitted to the Research Collaboratory for Structural Bioinformatics – Protein Data Bank under accession 7ABT.

PhaSepDB2.0 contains a database of 2,957 non-redundant phase-separation proteins and membrane-less organelles (http://db.phasep.pro/), curated from literature and databases.

IUPred2A was used to compute the likelihood of structural disorder per residue (https://iupred2a.elte.hu/).

## Research involving human participants, their data, or biological material

Policy information about studies with <u>human participants or human data</u>. See also policy information about <u>sex, gender (identity/presentation), and sexual orientation</u> and <u>race, ethnicity and racism</u>.

| | |
|---|---|
| Reporting on sex and gender | N/A |
| Reporting on race, ethnicity, or other socially relevant groupings | N/A |
| Population characteristics | N/A |
| Recruitment | N/A |
| Ethics oversight | N/A |

Note that full information on the approval of the study protocol must also be provided in the manuscript.

# Field-specific reporting

Please select the one below that is the best fit for your research. If you are not sure, read the appropriate sections before making your selection.

☒ Life sciences ☐ Behavioural & social sciences ☐ Ecological, evolutionary & environmental sciences

For a reference copy of the document with all sections, see nature.com/documents/nr-reporting-summary-flat.pdf

# Life sciences study design

All studies must disclose on these points even when the disclosure is negative.

| | |
|---|---|
| Sample size | A priori power calculations were conducted to ascertain the required sample size for each of two equal-sized groups. The effect size was established at 15% of the pooled standard deviations, estimated at 10% based on historical data from our group. This analysis was predicated on performing a two-tailed independent samples test, appropriate for testing without a predetermined direction. Power was validated using post hoc verification. |
| Data exclusions | No data were excluded from this study. Animals with signs of sickness were euthanized. |
| Replication | For transplantation studies involving Ppia heterozygous versus knockout donor cells, two separate experiments were conducted, one with |

| Replication | male and one with female donor mice, showing identical results. Other experiments that were performed in duplicates or triplicates are described in the Methods section and Figure Legends. |
|---|---|
| Randomization | Randomization was not feasible within the experimental design. However, in transplantation studies, transplant recipient animals were randomly assigned at the time of irradiation and donor cells were pooled from two or three animals. Experiments were age and gender matched and independently conducted with male/female mice. |
| Blinding | For transplantation studies, animal handling, microscopy, and FACS, experiments and analysis were handled by separate researchers. The FACS technician performing population statistics was blinded to the genotype of the donor mice.<br>For all imaging analyses, the experimenter was blinded to the origin of the samples.<br>For pulse SILAC and MS/MS, the researcher operating the mass spectrometrer and generating the raw proteomic data was blinded to the origin of the samples. |

# Reporting for specific materials, systems and methods

We require information from authors about some types of materials, experimental systems and methods used in many studies. Here, indicate whether each material, system or method listed is relevant to your study. If you are not sure if a list item applies to your research, read the appropriate section before selecting a response.

## Materials & experimental systems

| n/a | Involved in the study |
|---|---|
| ☐ | ☒ Antibodies |
| ☐ | ☒ Eukaryotic cell lines |
| ☒ | ☐ Palaeontology and archaeology |
| ☐ | ☒ Animals and other organisms |
| ☒ | ☐ Clinical data |
| ☒ | ☐ Dual use research of concern |
| ☒ | ☐ Plants |

## Methods

| n/a | Involved in the study |
|---|---|
| ☒ | ☐ ChIP-seq |
| ☐ | ☒ Flow cytometry |
| ☒ | ☐ MRI-based neuroimaging |

## Antibodies

| Antibodies used | Antigen & Fluorophore -  (Vendor & Catalog #) - Dilution:<br><br>FACS - Stem and progenitor cells:<br><br>CD45.1-PE  (Biolegend 110708)  1:200<br>CD45.1-APC  (eBioscience 17-0453-82) 1:200<br>CD45.2-PE Cy5.5 (eBioscience 35-0454-82) 1:200<br>CD45.2-Pac Blue (Biolegend 109820)  1:200<br>cKit-APC  (BD Biosciences 553356)  1:200<br>cKit-APC Cy7 (eBioscience 47-1171-82) 1:50<br>SA-Pac Orange (Invitrogen S32365)  1:200<br>SA-Pac Blue (Invitrogen S11222)  1:200<br>Sca1-Pac Blue (Biolegend 108120)  1:200<br>Sca1-PE Cy5.5 (Invitrogen MSCA18)  1:300<br>Sca1-PE Cy7  (Biolegend 108114)  1:200<br>CD34-FITC  (eBioscience 11-0341-85) 1:50<br>CD127-APC Cy7  (eBioscience 47-1271-82) 1:200<br>CD135-PE BD (Biosciences 553842)  1:50<br>CD150-PE Cy7  (Biolegend 115914)  1:200<br>CD16/32-PE-Cy7  (eBioscience 25-0161-82) 1:200<br><br>FACS - Peripheral blood:<br><br>CD45.1-PE  (Biolegend 110708)  1:125<br>CD45.2-BV421  (Biolegend 109832)  1:125<br>CD3-PE Cy7  (eBioscience 25-0031-81) 1:125<br>B220-PerCP Cy5.5 (eBioscience 45-0452-80) 1:125<br>MAC1-APC  (eBioscience 17-0112-81) 1:125<br>GR1-APC eF780 (eBioscience 47-5931-80) 1:125<br><br>Western and IP antibodies:<br><br>PPIA    (CST)  2175  1:1,000<br>PPIA    (Abcam)  ab1791  1:1,000<br>PPIA    (Abcam)  ab58144  IP<br>FLAG    (Sigma)  F1804  1:1,000<br>IgG1 control  (CST)  5415  1:1,000<br>HA    (Sigma)  11867423001 1:1,000 |
|---|---|

H3   (Abcam)  ab1791  1:1,000
PABPC1   (CST)  4992  1:1,000
Caprin1   (Proteintech) 15112-1-AP 1:1,000
DDX6   (Proteintech) 14632-1-AP 1:1,000
NPM1   (Proteintech) 10306-1-AP 1:5,000
G3BP1   (Proteintech) 13057-2-AP 1:1,000
beta-Tubulin   (CST)  86298  1:3,000
GAPDH   (Abcam)  ab204481 1:10,000

Proximity Ligation Assay:

PPIA   (Abcam)  ab58114  1:100
PPIA   (Abcam)  ab41684  1:100
PABPC1   (Proteintech) 10970-1-AP 1:100
DDX6   (Proteintech) 14632-1-AP 1:100
NPM1   (Proteintech) 10306-1-AP 1:100

Immuno-Fluorescence:

G3BP1   (Proteintech) 13057-2-AP 1:500
DDX6   (Proteintech) 14632-1-AP 1:100

**Validation**

All antibodies were commercially purchased and validated for their respective application by the manufacturer.

For flow cytometry:
B220-PerCP Cy5.5 (eBioscience/Thermo Scientific, Cat. #45-0452-80, Clone RA3-6B2). This RA3-6B2 antibody has been tested by flow cytometric analysis of mouse splenocytes. Advanced Verification: this antibody was verified by relative expression to ensure that the antibody binds to the antigen stated.

c-Kit (CD117)-APC (BD Biosciences, Cat. #553356, Clone 2B8). Application: Flow cytometry (Routinely rested). A single-cell suspension of BALB/c bone marrow was simultaneously stained with FITC Rat Anti-Mouse CD45R/B220 (Cat. Nos. 553087/553088, both panels) and either APC Rat IgG2b, κ Isotype Control (Cat. No. 553991) or APC Rat Anti-Mouse CD117 (Cat. No. 553356) monoclonal antibodies. Flow cytometry was performed on a BD FACSCalibur flow cytometry system.

c-Kit (CD117)-APC-Cy7 (eBioscience/Thermo Scientific, Cat. #47-1171-82, Clone 2B8). Applications Tested: this 2B8 antibody has been tested by flow cytometric analysis of mouse bone marrow cells.

Gr-1 (Ly-6G/Ly-6C)-APC eF780 (eBioscience/Thermo Scientific, Cat. #47-931-80, Clone RB6-8C5). This RB6-8C5 antibody has been tested by flow cytometric analysis of mouse bone marrow cells. Advanced verification: this Antibody was verified by relative expression to ensure that the antibody binds to the antigen stated.

MAC1 (CD11b)-APC (eBioscience/Thermo Scientific, Cat. #17-0112-81, Clone M1/70). The M1/70 antibody has been tested by flow cytometric analysis of mouse splenocytes.

Sca1-Pac Blue (BioLegend, Cat. #108120, Clone D7). Each lot of this antibody is quality control tested by immunofluorescent staining with flow cytometric analysis.

Sca1-PE Cy5.5 (Invitrogen/Thermo Scientific, Cat. #MSCA18, Clone D7). The antibody has been discontinued and validation information is no longer available on the website.

Sca1-PE Cy7 (BioLegend, Cat. #108114, Clone D7). Each lot of this antibody is quality control tested by immunofluorescent staining with flow cytometric analysis.

Streptavidin-Pac Orange (Invitrogen/Thermo Scientific, Cat. #S32365). For Use With (Application): Flow Cytometry, Immunoassays, Histochemical Applications, Blot Analysis. Not an antibody, no precise validation data is available. Certificate of Analysis only.

Streptavidin-Pac Blue (Invitrogen/Thermo Scientific, Cat. #S11222). For Use With (Equipment): Flow Cytometry, Immunoassays, Histochemical Applications, Blot Analysis. Not an antibody, no precise validation data is available. Certificate of Analysis only.

CD3-PE Cy7 (eBioscience/Thermo Scientific, Cat. #25-0031-81, Clone 145-2C11). This 145-2C11 antibody has been tested by flow cytometric analysis of mouse thymocytes and splenocytes.

CD16/32-PE Cy7 (eBioscience/Thermo Scientific, Cat. #25-0161-82, Clone 93). This 93 antibody has been tested by flow cytometric analysis of mouse splenocytes.

CD34-FITC (eBioscience/Thermo Scientific, Cat. #11-0341-85, Clone RAM34). This RAM34 antibody has been tested by flow cytometric analysis of mouse bone marrow cells.

CD45.1-PE (BioLegend, Cat. # 110708, Clone A20). Each lot of this antibody is quality control tested by immunofluorescent staining with flow cytometric analysis.

CD45.1-APC (eBioscience/Thermo Scientific, Cat. # 17-043-82, Clone A20). Applications Tested: the A20 antibody has been tested by flow cytometric analysis of mouse splenocytes.

CD45.2-PE Cy5.5 (eBioscience/Thermo Scientific, Cat. #35-0454-82, Clone 104). Applications Tested: this 104 antibody has been

tested by flow cytometric analysis of BALB/c splenocytes.

CD45.2-Pac Blue (BioLegend, Cat. #109820, Clone 104). Each lot of this antibody is quality control tested by immunofluorescent staining with flow cytometric analysis.

CD45.2-BV421 (BioLegend, Cat. #109832, Clone 104). Each lot of this antibody is quality control tested by immunofluorescent staining with flow cytometric analysis.

CD127-APC Cy7 (eBioscience/Thermo Scientific, Cat. #47-1271-82, Clone A7R34). This A7R34 antibody has been tested by flow cytometric analysis of mouse splenocytes.

CD135-PE (BD Biosciences, Cat. #553842, Clone A2F10.1). Application: Flow cytometry (Routinely Tested). In flow cytometric analysis, the A2F10 antibody recognizes Flt3-transfected Y3 cells (rat myeloma), but not the parent cell line in addition to recognizing early B lymphoid lineage cells in juvenile and adult bone marrow.

CD150-PE Cy7 (BioLegend, Cat. #115914, Clone TC15-12F12.2). Each lot of this antibody is quality control tested by immunofluorescent staining with flow cytometric analysis.

For Western blots and Immunoprecipitation assays:
PPIA (Cell Signaling Technology, Cat. #2175). Cyclophilin A Antibody detects endogenous levels of total Cyclophilin A protein. Application: Western Blotting, Dilution 1:1000. Species reactivity is determined by testing in at least one approved application (e.g., western blot).

PPIA (Abcam, Cat. # ab41684). Replenishment batches of our polyclonal antibody, ab41684 are tested in WB. Previous batches were additionally validated in ICC/IF.

PPIA (Abcam, Cat. #58144, Clone 1F4-1B5). Our Abpromise guarantee covers the use of ab58144 in the following tested applications: WB, ICC/IF, IP, flow cytometry. Western blot: Use at an assay dependent concentration. Predicted molecular weight: 18 kDa.

FLAG (Millipore Sigma, Cat. #F1804, Clone M2). Application: for highly sensitive and specific detection of FLAG fusion proteins by immunoblotting, immunoprecipitation, immunohistochemisty, immunofluorescence and immunocytochemistry. Optimized for single banded detection of FLAG fusion proteins in mammalian, plant, and bacterial expression systems. Specificity: Conforms. Detects a single band of protein on a Western Blot from mammalian crude cell lysates. Sensitivity test: Conforms. Detects 2 ng of FLAG-BAP fusion protein by Dot Blot using Chemiluminescent Detection.

IgG1 Isotype Control (Cell Signaling Technology, Cat. #5415). Mouse (G3A1) mAb IgG1 Isotype Control is not directed against any known antigen. It functions as an isotype control for mouse IgG1 monoclonal antibodies. Species reactivity is determined by testing in at least one approved application (e.g., western blot).

HA tag (Sigma-Aldrich, Cat. #11867423001, Clone 3F10). Quality: function tested in western blot. Use Anti-HA High Affinity for the detection of native influenza hemagglutinin protein and recombinant proteins that contain the HA epitope using Dot blots, ELISA, Immunocytochemistry, Immunoprecipitation, and Western blots.

Histone H3 (Abcam, Cat. #1791). WB: Detects a band of approximately 17 kDa (predicted molecular weight: 15 kDa). Specificity: based only on sequence homology, we expect the antibody to react with multiple variants of H3 such as H3.1, H3.2 and H3.3.

PABPC1 (Cell Signaling Technology, Cat. #4992). PABP1 Antibody detects endogenous levels of total PABP1 and PABP3 proteins. Species reactivity is determined by testing in at least one approved application (e.g., western blot).

Caprin1 (Proteintech, Cat. #15112-1-AP). Positive WB detected in HEK-293 cells, mouse brain tissue, HEK-293T cells, rat brain tissue, HeLa cells, Jurkat cells, NIH/3T3 cells.

DDX6 (Proteintech, Cat. #14632-1-AP). 14632-1-AP targets DDX6 in WB, IP, IHC, IF, ELISA applications and shows reactivity with human, mouse, rat samples. Positive WB detected in HeLa cells, HEK-293 cells, HepG2 cells, Jurkat cells, K-562 cells, C2C12 cells, mouse testis tissue.

NPM1 (Proteintech, Cat. #10306-1-AP). 10306-1-AP targets B23/NPM1 in WB, IP, IHC, IF, CoIP, ChIP, ELISA applications and shows reactivity with human, rat samples. Positive WB detected in COLO 320 cells, Jurkat cells, multi-cells, K-562 cells, HeLa cells, HEK-293 cells.

G3BP1 (Proteintech, Cat. #13057-2-AP). 13057-2-AP targets G3BP1 in WB, RIP, IP, IHC, IF, FC, CoIP, ELISA applications and shows reactivity with human, rat, mouse samples. Positive WB detected in C6 cells, HEK-293 cells, human brain tissue, Neuro-2a cells, HeLa cells, HepG2 cells, MCF-7 cells, Jurkat cells, mouse kidney tissue, rat kidney tissue, mouse brain tissue, rat brain tissue. The protein is known to run at higher than expected molecular weight in SDS PAGE.

Beta-Tubulin (Cell Signaling Technology, Cat. #86298, Clone D3U1W). β-Tubulin (D3U1W) Mouse mAb recognizes endogenous levels of total β-tubulin protein. Species reactivity is determined by testing in at least one approved application (e.g., western blot).

GAPDH (Abcam, Cat. #ab204481, Clone EPR16884). WB: detects a band of approximately 36 kDa (predicted molecular weight: 36 kDa). Species reactivity: reacts with: Mouse, Rat, Human.

For Proximity Ligation Assays:
PPIA (Abcam, Cat. # ab41684). Replenishment batches of our polyclonal antibody, ab41684 are tested in WB. Previous batches were additionally validated in ICC/IF.

PPIA (Abcam, Cat. #58144, Clone 1F4-1B5). Our Abpromise guarantee covers the use of ab58144 in the following tested applications:

WB, ICC/IF, IP, flow cytometry. Western blot: Use at an assay dependent concentration. Predicted molecular weight: 18 kDa.

PABPC1 (Proteintech, Cat. #10970-1-AP). 10970-1-AP targets PABPC1, PABP in WB, IP, IHC, IF, FC, ELISA applications and shows reactivity with human, mouse, rat samples. Positive IF detected in MCF-7 cells.

DDX6 (Proteintech, Cat. #14632-1-AP). 14632-1-AP targets DDX6 in WB, IP, IHC, IF, ELISA applications and shows reactivity with human, mouse, rat samples. Positive IF detected in HeLa cells, hTERT-RPE1 cells.

NPM1 (Proteintech, Cat. #10306-1-AP). 10306-1-AP targets B23/NPM1 in WB, IP, IHC, IF, CoIP, chIP, ELISA applications and shows reactivity with human, rat samples. Positive IF detected in HeLa cells.

Immuno-fluorescence:
G3BP1 (Proteintech, Cat. #13057-2-AP). 13057-2-AP targets G3BP1 in WB, RIP, IP, IHC, IF, FC, CoIP, ELISA applications and shows reactivity with human, rat, mouse samples. Positive IF detected in sodium arsenite treated HeLa cells.

DDX6 (Proteintech, Cat. #14632-1-AP). 14632-1-AP targets DDX6 in WB, IP, IHC, IF, ELISA applications and shows reactivity with human, mouse, rat samples. Positive IF detected in HeLa cells, hTERT-RPE1 cells.

# Eukaryotic cell lines

Policy information about cell lines and Sex and Gender in Research

| | |
|---|---|
| Cell line source(s) | Cell lines were purchased from ATCC (293T CRL-3216; HeLa CCL-2) or DMSZ (NB4 ACC-207; OCI-AML3 ACC-582), cultured with the medium composition recommended by the supplier, and monitored for signs of infection, including mycoplasma contamination. |
| Authentication | The ATCC cell lines were confirmed by STR profiling and HPV positivity (HeLa). |
| Mycoplasma contamination | Cells were monitored for signs of infection, including mycoplasma contamination. Negative tests were recorded using the Lonza MycoAlert Mycoplasma Detection Kit. |
| Commonly misidentified lines (See ICLAC register) | The study did not involve misidentified cell lines. |

# Animals and other research organisms

Policy information about studies involving animals; ARRIVE guidelines recommended for reporting animal research, and Sex and Gender in Research

| | |
|---|---|
| Laboratory animals | We used C57BL/6 wild-type mice or derived Ppia-/- and Ppia+/- mice of both sexes, multiple ages as indicated in the Methods section. As recipients in transplant experiments, we used female C57BL/6.SJL mice from the Jackson Laboratory (catalot no. 002014). Animals were housed in groups of 5 or 4 (if weight >25g) in ventilated cages in a pathogen-free high-barrier facility under ambient temperature and humidity. The mice were on a standard rodent diet of chow and water ad libitum, under a 12-hour light/dark cycle.<br><br>Animal ages (details described in the Methods section):<br>Transplant donors and recipients: 3-6 months of age, except for Fig. 2 where the donors were 18 months of age.<br>Age-dependent transcriptome and proteome analyses: young animals are 3-6 months of age, aged animals are over 20 months of age.<br>Ppia genotype-dependent transcriptome and proteome analyses: mice are 10-12 months of age.<br>General proteome and PLA interaction assays: mice are 4-8 months of age. |
| Wild animals | The study did not involve wild animals. |
| Reporting on sex | Experiments were performed using single-sex donor cells. No differences were observed between male or female donors. |
| Field-collected samples | The study did not involve samples collected from the field. |
| Ethics oversight | All animal experiments and care procedures were conducted at the Massachusetts General Hospital or the Baylor College of Medicine facilities in accordance with the Institutional Animal Care and Use Committee (IACUC) protocols approved at each institution, in compliance with all relevant ethical regulations, and following guidelines from the National Institutes of Health Guide for the Care and Use of Laboratory Animals (approved protocol #AN6745). The animal facilities were approved by the Association for Assessment and Accreditation for Laboratory Animal Care International (AAALAC). |

Note that full information on the approval of the study protocol must also be provided in the manuscript.

# Flow Cytometry

## Plots

Confirm that:

☒ The axis labels state the marker and fluorochrome used (e.g. CD4-FITC).

☒ The axis scales are clearly visible. Include numbers along axes only for bottom left plot of group (a 'group' is an analysis of identical markers).

☒ All plots are contour plots with outliers or pseudocolor plots.

☒ A numerical value for number of cells or percentage (with statistics) is provided.

## Methodology

| | |
|---|---|
| Sample preparation | Samples were prepared using standard protocols, with details outlined in the Methods section. |
| Instrument | BD Biosciences LSR Fortessa for analysis and Aria II for sorting. |
| Software | Analysis was performed using pre-installed BD Biosciences software FACS DiVa 9.0 and confirmed post-hoc with FlowJo v.10. |
| Cell population abundance | Relative and absolute cell numbers were calculated between various samples. While the relative numbers are shown in the manuscript, the absolute cell numbers showed consistent changes based on genotype of the donor animals. |
| Gating strategy | Standard gating strategy was performed, as detailed in the Methods section. Dead cells were excluded based on FSC/SSC scatter. |

☒ Tick this box to confirm that a figure exemplifying the gating strategy is provided in the Supplementary Information.

