## [Peer Review File · Nature Cell Biology]

Peer Review Information

Journal: Nature Cell Biology

Manuscript Title: Cyclophilin A supports translation of intrinsically disordered proteins and affects haematopoietic stem cell aging

Corresponding author name(s): Dr Andre Catic

Editorial Notes:

Reviewer Comments & Decisions:

Decision Letter, initial version:

Dear Professor Catic,

I once again apologize for the delay.

Your manuscript "Cyclophilin A supports translation of intrinsically disordered proteins and mitigates haematopoietic stem cell aging", has now been seen by 3 referees, who are experts in stem cell senescence (referee 1); proteostasis and proteomics (referee 2); and biomolecular condensation (referee 3), and whose comments are pasted below. In light of their advice, we regret that we cannot offer to publish the study in Nature Cell Biology.

As you will see, although the reviewers find this work potentially interesting, they raise a number of concerns that question the strength of the data and of the conclusions that can be drawn. These concerns include unclear effects on hematopoietic cells (please see comments from Reviewer #1), concerns whether loss of PPIA affects IDR protein specifically as opposed to overall effects on the cellular proteome, as well as questions about effects on biomolecular condensation, and validation of targets (all Reviewers), and in light of the points they raise, we find the present data-set too preliminary to pursue.

We are very sorry that we could not be more positive on this occasion, but we thank you for the opportunity to consider this work.

With kind regards,
Daryl Jason David

Daryl J.V. David, PhD

Senior Editor, Nature Cell Biology
Consulting Editor, Nature Communications
Nature Portfolio

Heidelberger Platz 3, 14197 Berlin, Germany
Email: daryl.david@nature.com
ORCID: <https://orcid.org/0000-0002-9253-4805>

Reviewers' comments:

Reviewer #1 (Remarks to the Author):

The study from Maneix et al. aims to elucidate the role of PPIA as a co-translational chaperone in HSPCs and to identify PPIA loss as a driver of hematopoietic stem cell aging. The authors performed proteomic analysis to show that proteins with intrinsically disordered regions (IDR) are PPIA substrates and are mainly involved in liquid-liquid phase separation and the formation of membrane-less complexes with other proteins or nucleic acids. Interestingly, they found that HSPC aging (in mice) associates with decreased PPIA protein level and reduced synthesis of proteins with IDR, likely leading to reduced stress resistance and stem cell dysfunctions

The main findings of the study are interesting, the topic is quite novel and the data are well-presented. However, some of the conclusions with regard to the role of PPIA-proteins with IDR in the aging phenotype of hematopoietic stem cells are not fully supported by the data. Specifically, the molecular link between PPIA depletion and the aging phenotype through loss of liquid-liquid phase separation remains weak. My main concern is the lack of a functional interplay between loss of PPIA, loss of IDR proteins and the impact that this unbalanced cellular proteome may have on the self-renewal and transplantation fitness of murine HSPC. Similarly, whether restoration of PPIA function and increased synthesis of proteins with IDR and phase separation are sufficient to rescue age-associated dysfunctions of hematopoiesis remains to be elucidated.

The original proteomic analysis by 2D gel electrophoresis and mass spec was performed in murine HSPCs that is notoriously a very heterogeneous cell population encompassing multipotent progenitors, committed progenitors and a rare subset of long-term HSCs. Validation of (some) of the identified targets in Fig.1 should be performed across HSPC subsets and in a sorted subset of most primitive HSCs.

The data presented in the transplantation experiments with wt and PPIAA KO HSPCs shown in Fig. 1c-h are clearly showing that PPIA deficiency leads to accelerated exhaustion of HSPCs but how much this dysfunction is related to PPIA-dependent regulation of the cellular proteome and deficiency of IDR proteins?

The experiments aimed at gaining molecular insights into the changes caused by PPIA depletion and at identifying the client proteins of PPIA are not performed in hematopoietic cells (instead in HeLa or 293T). What is the reason for not performing these experiments in murine HSPCs? The authors show that in PPIA KO HSPCs the levels of some interactors (G3BP1, DDX6, PABPC1..) is reduced by WB but some validation of interaction between PPIA and IDR proteins should be also performed in wt-HSPCs (by Co-IP or PLA that will require less cells).

In Fig. 3 The authors biochemically identified and confirmed that PABPC1 is a client of PPIA and show

that it undergoes phase transition and participate in stress granules at least in Hela cells. The authors should attempt similar set of experiments (modulation of PPIA activity, assess dynamics of stress granules formation and resistance to oxidative stress) in HSPCs. In addition they should show that reintroduction of the chaperone rescues stress granule formation in aged HSPCs. Such rescue experiments should be accompanied by functional evaluation of aged HSPCs in transplantation assays to really determine how much the axis PPIA-IDR-rich proteins-effective phase separation contributes to age-associated dysfunctions of HSPCs.

Reviewer #2 (Remarks to the Author):

The manuscript by Maneix et al. studied the role of Cyclophilin A (PPIA) in haematopoietic stem cell aging. The authors carried out transplantation assays in mouse models to describe premature aging phenotype upon PPIA loss. Using interaction proteomics, the authors defined PPIA target proteins and observed an enrichment of these to contain intrinsically disordered regions (IDRs). PPIA knock-down caused decreased translation and PPIA modulation affected stress granule formation. Finally, proteomics analyses revealed a correlation for PPIA and IDR in young and old HSCs.

The study addresses an interesting topic and the need to better define the role of PPIA in controlling proteostasis, IDR-containing proteins and a possible involvement in cell ageing. In the current state, the manuscript seems premature in that a number of crucial conclusions, which are critical for the logical flow of the manuscript, lack statistical rigor and necessary validation.

Major points:

- 1) The evidence for PPIA-mediated premature ageing is somewhat circumstantial. From the manuscript, it remains unclear how loss of PPIA shapes the cellular proteome and proteostasis. Consequently, a number of cellular pathways may be responsible for the observed effects. A more rigorous characterization of cellular changes, including classical molecular aging markers is necessary to claim the connection between PPIA and haematopoietic aging. Do PPIA levels and the specified interactions partners change when comparing old versus young HSPCs?
- 2) Several key conclusions drawn are based on various omics data, including proteome, interactome, transcriptome and translational changes. However, a number of these experiments lack a sufficient number of replicates to draw any conclusions with statistical significance. These design limitations in combination with a general lack of data availability (raw data, continuous quantification and statistical significance data) make it difficult to agree with the claims. Especially in that light, validation of the key claims by orthogonal methods is required. All key experiments that have not been validated orthogonally need to be carried out with at least three independent biological replicates.
- 3) Does PPIA loss affect translation via known cellular translation pathways regulated under stress and hence the observed reduced translation rate?
- 4) Figure 3b should be repeated including the G104A mutant to validate the findings.
- 5) Figure 3c, to confirm the interaction, at least one of the bait and prey should be monitored at the

endogenous level without overexpression.

6) The number of replicates need to be stated clearly, ideally in the experimental model or figure legend. Implying $n=3$ when no statement can be found is not sufficient. Drawing significance from distribution (violin) plots when there are not at least 3 biological replicates is not appropriate.

Consequently, these data sets are not sufficient as single supportive evidence for major claims.

7) Among the identified PPIA target proteins are a number of mitochondrial proteins (Figure 2c). Does loss of PPIA cause mitochondrial dysfunction and does that alter translation and the stem cell ageing phenotype?

8) A more unbiased representation of data is prudent to assess their quality. How do interactions change across all protein in the WT versus PPIA mutant comparison? How were interactors selected? For the transcriptome (Fig. 2f), also heat plots per replicate should be shown. A bit more insight into which proteins go up/down would be interesting. The percentage of heavy incorporated is very low, additional validation that there is still sufficient dynamic range to monitor dynamics would be important. The transcriptome data set is quite useful and more data should be shown (e.g. quality control/correlation across replicates, some clustering or pathway analyses).

9) The claim that PPIA and PABPC1 regulate phase separation needs further examination. Reduction of protein translation, as is observed upon PPIA loss, is sufficient to induce stress granules. While these are phase separated, it is not clear that PPIA and PABPC1 drive the physical aspects of phase separation. How are IDR and other target proteins changed upon rescue of PPIA?

Minor points

10) PPIA Kd2 and Kd1 (293T) show major difference in expression pattern (Figure 2f). The author could a) cluster the target proteins based on function, b) should represent fold change and significance of target proteins relative to control.

11) Extended data Fig. 3c: Lower right blot midsection is unlabeled. Is that correct?

Reviewer #3 (Remarks to the Author):

Comment on 'Cyclophilin A supports translation of intrinsically disordered proteins and mitigates haematopoietic stem cell aging' by Maneix et al.

In this paper, the authors have identified Peptidyl-3 Prolyl Isomerase A (PPIA or Cyclophilin A) as a dominant chaperone in haematopoietic stem and progenitor cells (HSPC) as it accounts for 14% of the cytosolic proteome and its mRNA accounts for over 0.5% of all mRNAs of HSPC. Total knockout of PPIA leads to 1) myeloid skewing, an immunophenotypic but not functional increase in stem cells, and 2) impaired self-renewal with 3) accelerated exhaustion, which are three hallmarks of haematopoietic aging. Using an assay targeting differential co-immunoprecipitation between the wild-type PPIA and a G104A mutant PPIA, the authors revealed approximately 400 substrates of the wild-type enzyme. Since they performed the co-immunoprecipitation in the cytosolic cell fraction, these results suggest PPIA and substrates interact between translation and nuclear translocation. When compared to the global proteome, immunoprecipitated PPIA substrates feature higher levels of IDRs.

By surveying proline residues within PDB, the authors found that the majority of cis-prolines located in bends and turns. Their conclusion is that "PPIA and related enzymes predominantly isomerize prolines within unstructured protein regions." This conclusion is incorrect as the bends and turns are categorized as structured regions that connecting helices and beta-strands. "Unstructured protein regions" are regions of proteins that they donot have stable (i.e. highly populated) conformations and are "invisible" in PDB. Further studies are required to justify this conclusion.

Using pulsed SILAC in HeLa and 293T cells with either normal or reduced levels of PPIA, the authors found that loss of PPIA significantly reduced expression of the whole proteome, PPIA clients and non-clients, in both cell types. As far as the levels of PPIA target proteins are concerned, they were lower than for other proteins and further reduced when PPIA was knockdown. In addition, the fraction of protein IDRs in the HSPC proteome of PPIA knockout animals was significantly reduced when compared with heterozygous siblings.

Largely based on these aforementioned observations, the authors claim that PPIA supports de novo translation of its target proteins. Unfortunately, this claim is not even close to be supported by the data reported.

The authors went on to confirmed a few clients of PPIA using a combination of IP and western blotting. One of the confirmed clients is PABPC1. "Following PPIA knockdown, expression of PABPC1 protein was reduced by 20-30% in pulsed SILAC experiments, suggesting the chaperone engages with PABPC1 during translation." This conclusion is questionable as the pulse was 24 hours. Many other mechanisms other than, or in addition to, limited translation can account for the 20-30% reduction.

Next, the authors turned their attention to liquid-liquid phase separation (LLPS), a biophysical property of multivalent proteins including many IDR-containing ones. Essentially the authors showed that PPIA KD perturbed condensates of PABPC1-GFP (I have a hard time to figure whether PABPC1-GFP is transiently expressed or is expressed at endogenous levels) and the formation of stress granules marked by G3BP1 upon the treatment with sodium arsenite. It is unclear whether the reduced translation of PPIA's clients or the lack of PPIA activity per se was claimed to cause the perturbation to condensates of PABPC1-GFP or stress granule formation. Nevertheless LLPS is just one aspect of the global proteome

Overall, the authors didn't make sufficient effort to connect pieces of observations for a coherent model explaining the role of PPIA in ageing of haematopoietic stem cells. I can't recommend its further consideration for pulication in Nature Cell Biology.

To help them improving their study for future submission, I listed some suggestions below:

Major issues

- 1)When doing differential analysis of proteome to assess the effects of PPIA on IDR-enriched clients, the authors shall divide PPIA clients and nonclients into IDR-containing, IDR-noncontaining, four categories.
- 2)Genetic manipulation or pharmacological perturbation for days are rather slow to assess acute and direct functions. I suggest the authors to use an AID system or PROTAC system (if available) to chemically KD PPIA and assess whether the reported effects are direct or indirect.

Minor issues

- 1) There are no legends for SI-Videos.

**Although we cannot publish your paper, it may be appropriate for another journal in the Nature Portfolio. If you wish to explore the journals and transfer your manuscript please use our manuscript transfer portal. You will not have to re-supply manuscript metadata and files, but please note that this link can only be used once and remains active until used. For more information, please see our manuscript transfer FAQ page.

Note that any decision to opt in to In Review at the original journal is not sent to the receiving journal on transfer. You can opt in to In Review at receiving journals that support this service by choosing to modify your manuscript on transfer. In Review is available for primary research manuscript types only.

Author Rebuttal to Initial comments

We thank the reviewers for their time and expertise in critiquing our initial manuscript. The insightful feedback has greatly assisted us in improving the robustness and clarity of our work. We are confident that addressing the highlighted limitations resulted in significant enhancements to our manuscript. To maintain transparency, responses to critiques and subsequent amendments to the manuscript are in blue font.

Reviewer #1 (Remarks to the Author):

The study from Maneix et al. aims to elucidate the role of PPIA as a co-translational chaperone in HSPCs and to identify PPIA loss as a driver of hematopoietic stem cell aging. The authors performed proteomic analysis to show that proteins with intrinsically disordered regions (IDR) are PPIA substrates and are mainly involved in liquid-liquid phase separation and the formation of membrane-less complexes with other proteins or nucleic acids. Interestingly, they found that HSPC aging (in mice) associates with decreased PPIA protein level and reduced synthesis of proteins with IDR, likely leading to reduced stress resistance and stem cell dysfunctions

The main findings of the study are interesting, the topic is quite novel and the data are well-presented.

We thank reviewer #1 for his/her positive assessment, who clearly understood the novelty and importance of our study to the field.

However, some of the conclusions with regard to the role of PPIA-proteins with IDR in the aging phenotype of hematopoietic stem cells are not fully supported by the data. Specifically, the molecular link between PPIA depletion and the aging phenotype through loss of liquid-liquid phase separation remains weak. My main concern is the lack of a functional interplay between loss of PPIA, loss of IDR proteins and the impact that this unbalanced cellular proteome may have on the self-renewal and transplantation fitness of murine HSPC. Similarly, whether restoration of PPIA function and increased synthesis of proteins with IDR and phase separation are sufficient to rescue age-associated dysfunctions of hematopoiesis remains to be elucidated.

We concur with reviewer #1's concerns about the molecular link between PPIA depletion, the aging phenotype, and the loss of liquid-liquid phase separation. To address these concerns, we have performed additional experiments to provide compelling evidence for the functional interplay between loss of PPIA, loss of intrinsically disordered proteins (IDPs), and the impact on self-renewal and transplantation fitness of murine HSPCs. The new data and analyses are summarized below:

1. Functional interplay between loss of PPIA and HSPC fitness:

To demonstrate the functional interplay, we have performed critical rescue experiments to evaluate the effect of PPIA on self-renewal and transplantation fitness in murine haematopoietic stem and progenitor cells (new Fig. 2a-d). Mouse PPIA was introduced using a lentiviral construct in aged stem and progenitor cells, which significantly improved reconstitution following competitive transplantation. We followed these animals for >6 months, which indicates that the enhanced phenotype is contributed by the long-lived stem cell pool. We further verified that the PPIA protein is increased in the blood of animals after receiving "rescued" cells and that the resulting improved haematopoiesis is not skewed towards myeloid lineages.

2. Functional interplay between PPIA and IDR protein expression:

PPIA is a widely expressed chaperone with a broad substrate base. While PPIA recognizes many cellular proteins with a larger fraction of disordered regions on average, it is not exclusive for disordered proteins. We, therefore, used the term "selective" instead of "specific" in our manuscript. Nonetheless, to strengthen the evidence for the interaction between PPIA and substrate proteins, which contain many well-recognized and important players in liquid-liquid phase separation, we investigated the expression and function of the stress granule protein PABPC1, the nucleolar organizer NPM1, and the P-body protein DDX6. We found that

depletion of PPIA leads to reduced expression of these three essential proteins (new Fig. 4g and Extended Data Fig. 3b). Further, to substantiate that PPIA directly interacts with these three proteins, we performed co-immunoprecipitation with either wildtype PPIA or the binding-impaired G104A mutant [1]. Consistent with our previous proteomic studies, we found reduced levels of PABPC1, NPM1, and DDX6 when performing immunoprecipitation with the mutant PPIA (new Fig. 4f). The dependence of substrate expression on PPIA was also noted by the group of Dr. Zweckstetter [2]. Importantly, reduced levels of PPIA substrates in PPIA-deficient cells are not driven by transcription (new Extended Data Fig. 3f).

The original proteomic analysis by 2D gel electrophoresis and mass spec was performed in murine HSPCs that is notoriously a very heterogeneous cell population encompassing multipotent progenitors, committed progenitors and a rare subset of long-term HSCs. Validation of (some) of the identified targets in Fig.1 should be performed across HSPC subsets and in a sorted subset of most primitive HSCs.

We acknowledge the potential limitations of using bulk murine cells for our proteomic analysis in the absence of single-cell mass spectrometry technology. PPIA is widely expressed across stem and progenitor cells subsets, and we focused on PPIA's overall impact on the aging haematopoietic phenotype. We are confident that haematopoietic stem cell-specific changes are contributing to this phenotype due to the long-term functional impact of PPIA deficiency and rescue in transplantation assays. However, we note that our study did not claim that PPIA is a chaperone specific for a subset of haematopoietic stem and progenitor cells.

Recently, the laboratory of Dr. Irv Weissman published a detailed proteome catalog across different murine bone marrow stem and progenitor subsets [3]. We re-examined this valuable resource and found robust expression of PPIA in mouse haematopoietic stem cells (new Extended Data Fig. 1b). Importantly, the Weissman group also observed significantly reduced PPIA protein expression in aged HSCs (new Extended Data Fig. 4a), consistent with the PPIA quantification we performed in young versus aged haematopoietic stem cells (Fig. 6a).

The data presented in the transplantation experiments with wt and PPIA KO HSPCs shown in Fig. 1c-h are clearly showing that PPIA deficiency leads to accelerated exhaustion of HSPCs but how much this dysfunction is related to PPIA-dependent regulation of the cellular proteome and deficiency of IDR proteins?

We recognize that with current technologies, it is difficult to directly quantify the exact contribution of PPIA-dependent proteome regulation to the observed stem and progenitor dysfunction. To substantiate our conclusion, we have performed additional experiments to support the relevance of PPIA-dependent regulation of IDPs PABPC1, NPM1, and DDX6, and their expression and contribution to stress granules, nucleoli, and P-bodies, respectively (new Figs. 4g and 5b,c). Our new data serve as a significant initial step in identifying PPIA as a crucial chaperone for IDP expression and activity, thereby influencing stem cell functionality. Importantly, we show that PPIA depletion results in a skewed proteome, disfavoring IDPs (Fig. 6c and new Extended Data Figs. 4d,e and 5a-c). Identifying how loss of individual or multiple IDPs directly contributes to haematopoietic aging is beyond the scope of the current manuscript (see also response to reviewer #2, major point 1).

The experiments aimed at gaining molecular insights into the changes caused by PPIA depletion and at identifying the client proteins of PPIA are not performed in hematopoietic cells (instead in Hela or 293T). What is the reason for not performing these experiments in murine HSPCs? The authors show that in PPIA KO HSPCs the levels some interactors (G3BP1, DDX6, PABPC1..) is reduced by WB but some validation of interaction between PPIA and IDR proteins should be also performed in wt-HSPCs (by Co-IP or PLA that will require less cells).

The reviewer highlights a crucial concern regarding the interaction of PPIA with substrates and the possibility that this is cell type-specific. PPIA belongs to the extensive family of cyclophilins, which, in turn, is one of four distinct families of prolyl isomerases, including FKBP, Parvulins, and PTPA. Consequently, compensation attenuates the effects of PPIA depletion, especially in constitutive knockout mouse models. We found more robust responses when using acute knockdown approaches. The critical proteomic and transcriptomic studies

were performed in primary haematopoietic cells, and we confirmed diminished expression of PABPC1, DDX6, and NPM1 in the haematopoietic cell line, AML3, after knockdown of PPIA (new Extended Data Fig. 3b). Further, we conducted western blots in primary mouse haematopoietic stem and progenitor cells after *Ppia* knockout to validate our findings (Extended Fig. 3c). Unfortunately, visualization of P-bodies and stress granules by microscopy proved difficult in the haematopoietic system (our unpublished observation), prompting us to use epithelial cell lines for these assays. To specifically address the interaction between PPIA and IDPs, we performed Proximity Ligation Assays (PLA) in primary haematopoietic stem cells, which confirmed that PPIA interacts with PABPC1, DDX6, and NPM1 in mouse and human cells (new Fig. 5d).

In Fig. 3 The authors biochemically identified and confirmed that PABPC1 is a client of PPIA and show that it undergoes phase transition and participate in stress granules at least in Hela cells. The authors should attempt similar set of experiments (modulation of PPIA activity, assess dynamics of stress granules formation and resistance to oxidative stress) in HPSCs. In addition they should show that reintroduction of the chaperone rescues stress granule formation in aged HSPCs. Such rescue experiments should be accompanied by functional evaluation of aged HSPCs in transplantation assays to really determine how much the axis PPIA-IDR-rich proteins-effective phase separation contributes to age-associated dysfunctions of HSPCs.

We concur with the reviewer that performing experiments in primary haematopoietic cells would be optimal. Regrettably, quantitatively visualization of stress granules in these cells was not feasible because of the restricted size of the cytoplasm. Nonetheless, to obtain a deeper understanding of the disrupted pathways in PPIA-depleted stem and progenitor cells, we conducted RNA-sequencing analyses. We found that a majority of altered gene programs were related to initial stages of proteostasis, unlike disruptions of protein repair or clean-up pathways that lead to the accumulation of misfolded proteins. Notably, a compensatory upregulation of cytoplasmic translation emerges as one of the most prominent gene ontologies observed in *Ppia* knockout cells. Conversely, we did not observe molecular chaperones or the ubiquitin-proteasome system being transcriptionally activated by PPIA depletion. Our results further substantiate our hypothesis that PPIA plays a crucial role in protein biosynthesis (new Extended Data Fig. 3d). To directly test this, we carried out protein synthesis assays and found that *de novo* translation is markedly diminished in primary haematopoietic stem cells following treatment with a PPIA-specific inhibitor (new Extended Data Fig. 3a).

In our revised manuscript, we present unequivocal evidence that PPIA engages with substrate proteins, a majority of which possess a high level of intrinsic disorder, based on predictions with the widely-accepted IUPred2A algorithm [4]. Furthermore, we report that the decrease in PPIA—an occurrence also observed during aging in haematopoietic stem cells—is associated with a significant reduction of IDPs within the proteome. Notably, numerous PPIA substrate proteins are essential for various biological processes, especially in mRNA processing. It is also noteworthy that in *Ppia* knockout animals, we detected a significant transcriptional upregulation of PPIA substrates (new Extended Data Fig. 3f). This phenomenon underscores the essential role of PPIA and PPIA substrates within the cellular environment.

We propose that the intimate link between prolyl isomerase and protein synthesis could be similar of the role of trigger factor, a functional homolog of human PPIA in Prokarya, in the folding of the nascent polypeptide chain as it emerges from translating ribosomes. Drawing from our observations and those documented in other studies, we postulate that PPIA may fulfill a comparable function in protein folding in Eukarya.

Reviewer #2 (Remarks to the Author):

The manuscript by Maneix et al. studied the role of Cyclophilin A (PPIA) in haematopoietic stem cell aging. The authors carried out transplantation assays in mouse models to describe premature aging phenotype upon PPIA loss. Using interaction proteomics, the authors defined PPIA target proteins and observed an enrichment of these to contain intrinsically disordered regions (IDRs). PPIA knock-down caused decreased translation and PPIA modulation affected stress granule formation. Finally, proteomics analyses revealed a correlation for PPIA and IDR in young and old HSCs.

The study addresses an interesting topic and the need to better define the role of PPIA in controlling proteostasis, IDR-containing proteins and a possible involvement in cell ageing. In the current state, the manuscript seems premature in that a number of crucial conclusions, which are critical for the logical flow of the manuscript, lack statistical rigor and necessary validation.

We appreciate the constructive criticism of reviewer #2, who considered our study an interesting topic. We respect your perspective and have thus worked diligently to strengthen our manuscript. While we understand your initial assessment, we would like to politely disagree that our revised manuscript remains in a premature state. We showed that PPIA is the first chaperone with a preference for intrinsically disordered proteins (IDPs) and provide evidence that PPIA and its substrates decline with age within the haematopoietic system, which, to the best of our knowledge, are two impactful, novel findings. In response to this reviewer's comments, we provide substantial new evidence derived from additional sets of experiments that were performed over the past year to address any perceived limitations of our study. As requested by this reviewer, we have also enhanced the statistical robustness by increasing the number of biological replicates in pivotal experiments. Importantly, we present triplicates of biologically independent analyses of isobaric proteomic studies comparing both young versus old as well as *Ppia* heterozygous versus knockout cells in the haematopoietic system (new Extended Data Fig. 4c-e). We believe the additions we made significantly address the concerns raised, and bring more depth, clarity, and precision to our work.

Major points:

1) The evidence for PPIA-mediated premature ageing is somewhat circumstantial. From the manuscript, it remains unclear how loss of PPIA shapes the cellular proteome and proteostasis. Consequently, a number of cellular pathways may be responsible for the observed effects. A more rigorous characterization of cellular changes, including classical molecular aging markers is necessary to claim the connection between PPIA and haematopoietic aging. Do PPIA levels and the specified interactions partners change when comparing old versus young HSPCs?

We recognize that multiple pathways can affect haematopoietic aging. To address this issue, we have now included data from RNA sequencing of three biological replicates, comparing heterozygous and knockout *Ppia* stem and progenitor cells (new Extended Data Fig. 3d). Notably, we find that the knockout cells exhibit a pronounced gene set enrichment that reflects data published in the recent *Tabula Muris Senis* study and is consistent with the transcriptome of aged haematopoietic stem cells [7] (new Extended Data Fig. 3e). Moreover, P-Selectin has been reported by several research groups [8, 9] as the most consistently upregulated marker of aged stem and progenitor cells. In line with these findings, we observed a significant upregulation of this gene in *Ppia* knockout cells compared to heterozygous cells. Importantly, our experiments demonstrate that augmenting PPIA expression in aged stem and progenitor cells ameliorates their performance in a competitive transplantation assay (new Fig. 2a-d). This finding substantiates the proposed causal relationship between PPIA activity and the aging process within this cellular compartment.

Nevertheless, while our findings underscore the significance of PPIA in aging, it is important to acknowledge that aging is a multifaceted process [10]. Given that PPIA interacts with proteins involved in diverse cellular

pathways, such as splicing, translation, and transcription, we assert that it would be an oversimplification and potentially misleading to concentrate solely on a single aging-driving mechanism.

In line with our data, we propose that structural, in addition to any functional features, characterize protein alterations with age. Consequently, attempts to redress PPIA deficiency by overexpressing individual substrate proteins is unlikely to further our understanding of this intricate process. We note that the substantial transcriptional upregulation of PPIA substrates observed in knockout models suggests the existence of compensatory mechanisms, thereby ensuring adequate expression of these critical RNA-processing proteins and the survival of *Ppia* knockout cells (new Extended Data Fig. 3f).

2) Several key conclusions drawn are based on various omics data, including proteome, interactome, translome and transcriptome changes. However, a number of these experiments lack a sufficient number of replicates to draw any conclusions with statistical significance. These design limitation in combination with a general lack of data availability (raw data, continuous quantification and statistical significance data) make it difficult to agree with the claims. Especially in that light, validation of the key claims by orthogonal methods is required. All key experiments that have not been validated orthogonally need to be carried out with at least three independent biological replicates.

Our results were derived using a diverse range of orthogonal methods that substantially reinforce the reliability of our data. To further bolster the robustness of our findings, we have undertaken additional proteomics experiments—specifically, isobaric profiling—of young and aged cells, as well as *Ppia* heterozygous and knockout cells. These experiments were conducted with each condition assessed independently in biological triplicates (new Extended Data Fig. 4c-e). The raw and normalized proteome-wide quantification obtained through isobaric labeling with tandem mass tags (TMT) is now available as supplementary material.

In addition, we performed a new RNA-seq analysis on *Ppia* heterozygous and knockout haematopoietic stem and progenitor cells in triplicate to further enhance the significance of PPIA on transcription and transcriptional compensation (new Extended Data Fig. 3d-f). Together, this comprehensive approach adds substantial weight to our conclusions and strengthens our understanding of PPIA's role in cellular processes.

3) Does PPIA loss affect translation via known cellular translation pathways regulated under stress and hence the observed reduced translation rate?

The reviewer raises an important point as to why protein translation of proteins is impaired when PPIA is depleted or inhibited. We propose that one explanation involves compromised ribosomal biogenesis. NPM1, a master regulator of nucleolar structure and function, is a PPIA substrate. Genetic depletion or pharmacologic inhibition of PPIA reduces NPM1 levels and impairs the shape and size of nucleoli (new Fig. 5c), which may disrupt ribosomal biogenesis. In support of this, the upregulation of ribosomal genes represents one of the most significant ontological changes in *Ppia* knockout cells (new Extended Data Fig. 3d), while we observed no significant upregulation of genes encoding molecular chaperones or components of the ubiquitin-proteasome system.

In addition to an effect of PPIA on ribosomal biogenesis through NPM1, our data also suggest that PPIA may participate in early folding of nascent polypeptides, and thereby affect the translation efficiency of its substrates. We found several results that would corroborate such a model: First, we determined cytoplasmic association of PPIA with nuclear proteins, suggesting interactions during translation. Second, PPIA substrate translation is sensitive in the SILAC study. Third, using acute pharmacological inhibition of PPIA, we observed a robust reduction of global protein translation (new Extended Data Fig. 3a). Fourth, transcriptional upregulation of genes encoding PPIA target proteins suggests a compensatory mechanism and associates the chaperone with these substrates (new Extended Data Fig. 3f).

4) and 5) Figure 3b should be repeated including the G104A mutant to validate the findings. Figure 3c, to confirm the interaction, at least one of the bait and prey should be monitored at the endogenous level without overexpression.

Previous IP-Western experiments for PPIA and its substrates were performed at endogenous protein levels (Fig. 4e). Per the reviewer's request, we have transfected HeLa cells with either 3xP-PPIA or 3xP-PPIA(G104A). Immunoprecipitation was then conducted, followed by probing with antibodies against endogenous PABPC1, DDX6, and NPM1 (new Fig. 4f). The data demonstrate a reduced ability of the mutant PPIA to interact with the three substrate proteins. Furthermore, we verified the interaction between PPIA and the aforementioned proteins at endogenous levels in haematopoietic stem cells using proximity ligation assays (new Fig. 5d).

6) The number of replicates need to be stated clearly, ideally in the experimental model or figure legend. Implying n=3 when no statement can be found is not sufficient. Drawing significance from distribution (violin) plots when there are not at least 3 biological replicates is not appropriate. Consequently, these data sets are not sufficient as single supportive evidence for major claims.

The number of replicates is now clearly stated in the experimental model or figure legend. In addition, we have performed additional as well as orthogonal experiments to substantiate the statistical validity of our findings. Specifically, the primary omics experiments, including isobaric proteomics and the new RNA-seq analysis of *Ppia* knockout and heterozygous haematopoietic stem and progenitor cells, are based on triplicates of independent biological replicates (n=3). The results of the cumulative and individual assays are shown in aggregated form in the main figures (Fig. 6) and individually as supplemental material (new Extended Data Fig. 4), respectively. Raw data are included as supplementary information.

7) Among the identified PPIA target proteins are a number of mitochondrial proteins (Figure 2c). Does loss of PPIA cause mitochondrial dysfunction and does that alter translation and the stem cell ageing phenotype?

Like reviewer #2, we were intrigued by the abundance of mitochondrial proteins amongst PPIA target proteins. However, as a caveat, this observation may have resulted from organelle contaminations due to the use of detergents in our pulldown assay: It is possible that mitochondrial proteins have leaked and inadvertently bound to PPIA post lysis.

In the transcriptional profiling of *Ppia* knockout haematopoietic stem and progenitor cells, we noted an increased expression of nuclear encoded mitochondrial genes, especially those involved in oxidative phosphorylation (see **Figure A** below). Consistently, we also found an increase in mitochondrial proteins in *Ppia* knockout cells based on isobaric labeling and mass spectrometry (not shown). However, we do not have evidence linking absence of the chaperone directly to the enhanced translation of these proteins, given that *Ppia* knockout enhances transcription of their encoding genes. Additionally, for other PPIA substrates, we observed decreased translation in the absence of PPIA, which contrasts with the increased levels observed in the case of mitochondrial proteins.

To further examine this using a more acute cell line-based knockdown model, we performed a series of oxygen consumption assays, utilizing a Seahorse instrument on 293T and HeLa epithelial cells with and without PPIA depletion. These assays did not show a consistent alteration in mitochondrial activity (see **Figure B** below).

Figure A: A subset of nuclear-encoded mitochondrial genes is upregulated in *Ppia* knockout cells. Haematopoietic stem and progenitor cells (lin-/cKit+) show significant transcriptional upregulation of gene sets involved in oxidative phosphorylation. Analysis performed with GSEA software package ver. 4.3.2.

Figure B: No change in mitochondrial function following knockdown of PPIA. PPIA-deficiency in 293T cells (shown) and HeLa cells (not shown) shows no consistent alterations in mitochondrial respiration or ATP production, as measured by Seahorse assay (n>10). P-values were not significant (ns) as calculated by the two-sided Wilcoxon rank-sum test.

Taken together, we do not believe that PPIA modifies mitochondria in a distinct, directional manner, which would enable us to draw conclusions about changes in the organelle's activity. The augmentation in the number of several mitochondrial proteins, which occurred independently of respiration changes, is likely caused by transcriptional changes in nuclear-encoded mitochondrial genes within the haematopoietic compartment. While these transcriptional shifts may be associated with PPIA activity, they could also mirror an altered composition of the stem and progenitor cell pool. Specifically, a shift may have occurred in the knockout animals transitioning from quiescent stem cells towards committed progenitor cells that exhibit a higher mitochondrial abundance [11]. Such a shift in cell composition would be consistent with the functional changes that we observed in transplanted animals with higher levels of CD150-positive stem cells (Fig. 1f).

8) A more unbiased representation of data is prudent to assess their quality. How do interactions change across all protein in the WT versus PPIA mutant comparison? How were interactors selected? For the translome (Fig. 2f), also heat plots per replicate should be shown. A bit more insight into which proteins go up/down would be interesting. The percentage of heavy incorporated is very low, additional validation that there is still sufficient dynamic range to monitor dynamics would be important. The transcriptome data set is quite useful and more data should be shown (e.g. quality control/correlation across replicates, some clustering or pathway analyses).

Following the reviewer's recommendation, we performed an integrative analysis of the transcriptome as well as of the proteome of haematopoietic stem and progenitor cells. The findings are summarized below:

Transcriptome: The absence of PPIA does not instigate a chaperone response or enhance the expression of the ubiquitin-proteasome system. Instead, we witnessed an upregulation of cytosolic translation-related genes, notably ribosome subunits, as one of the most distinct responses (new Extended Data Fig. 3d). This observation implies that the role of PPIA is within the functional context of *de novo* protein synthesis as opposed to later stages of proteostasis. We hypothesize that PPIA might fulfill a role in co-translation protein folding akin to bacterial trigger factor that monitors the folding of the nascent chain. Like trigger factor, PPIA possesses prolyl isomerase activity (though employing a different catalytic domain), which is integral to protein synthesis.

The proposed involvement of PPIA in early translation is further supported by the lack of accumulation of misfolded proteins in PPIA-deficient cells (new Extended Data Fig. 3g). An increase in protein misfolding was only observed in the presence of proteasome inhibitors, suggesting that the cellular protein clearance machinery adequately compensates for PPIA deficiency.

Proteome: An unbiased analysis of protein changes based on the level of structural disorder within proteins also verifies that intrinsically disordered proteins cluster according to genotype (*Ppia* heterozygous versus *Ppia* knockout), while no such clustering is observed for proteins that lack disordered regions (new Extended Data Fig. 5a,b). In order to reciprocally validate the quality of these clusters, we conducted an unbiased hierarchical analysis of all proteins identified in the isobaric labeling experiment of *Ppia* expressing or knockout cells. The cluster that was upregulated the most across *Ppia* expressing cells is enriched for IDPs involved in cellular processes commonly found amongst its substrate proteins (i.e., mRNA binding and processing), while the top cluster upregulated in *Ppia* knockout cells exhibits lower-than-average levels of intrinsic disorder (new Extended Data Fig. 5c). Also of note, while genes encoding cytosolic ribosomal proteins were upregulated in *Ppia* deficient cells, the respective proteins were reduced in these cells, suggesting that transcriptional upregulation compensates for a translational bottleneck in cells lacking PPIA.

PPIA substrate proteins are enriched for essential RNA binding proteins. In the constitutive *Ppia* knockout model, overall expression of these vital proteins was only moderately reduced and likely masked by transcriptional compensation: the genes of PPIA substrates are significantly upregulated in knockout animals, potentially limiting the effect of chaperone loss at the protein level (new Extended Data Fig. 3f). Notably, when

we examined the expression of PPIA substrate proteins in cell line-based *PPIA* knockdown systems (new Fig. 3g, Extended Data Fig. 3b), we observed a significant reduction of the substrates shortly after knockdown, which gradually diminished over time.

Translatome: By re-evaluating our SILAC data in cell lines, we confirmed that cytosolic ribosomal proteins are among the most repressed proteins in *PPIA*-deficient cells and that knockdown cells cluster separately from *PPIA*-expressing cells, even when comparing two different cell types (293T and HeLa cells). The top-scoring gene ontologies are now presented in **Fig. C** (below).

In summary, our results suggest the involvement of *PPIA* in early translation predominantly affects proteins with a high degree of structural disorder. In this context, the pronounced repression of IDPs in aged or *PPIA*-deficient haematopoietic cells is striking. Furthermore, observations derived from knockout models indicate potential transcriptional compensation.

Figure C: Gene ontology analysis of proteins most affected in *de novo* translation following *PPIA* depletion. Pulsed SILAC analysis of 293T and HeLa cells following *PPIA* knockdown shows that proteins most reduced in *PPIA*-depleted conditions are involved in mRNA splicing and translation, consistent with the ontologies of *PPIA* substrates. Listed are the top unique ontologies and p-values.

9) The claim that *PPIA* and *PABPC1* regulate phase separation needs further examination. Reduction of protein translation, as is observed upon *PPIA* loss, is sufficient to induce stress granules. While these are phase separated, it is not clear that *PPIA* and *PABPC1* drive the physical aspects of phase separation. How are IDR and other target proteins changed upon rescue of *PPIA*?

We appreciate the reviewer's question about how *PPIA* influences phase separation, a topic that we find most intriguing. Prior studies support a role of *PPIA* in stress granule formation as *PPIA* co-localizes with these structures *in vivo* and *PPIA* inhibition reduces their formation [13, 14].

To address the reviewer's query, it is indeed counterintuitive why depletion of a chaperone, which increases cellular stress, reduces stress granule formation. Based on our results and published findings by others, we surmise that this can be explained by *PPIA* enhancing the synthesis and/or function of proteins associated with stress granule assembly. Reintroducing *PPIA* partially rescues the stress granule deficiency in knockdown cells, suggesting that *PPIA* might increase the concentration of substrate proteins that are involved in phase separation of these organelles [15], or it might alter substrate protein structure, thereby modifying function and phase mixing/demixing potential [16].

Our work represents a major effort in characterizing *PPIA* as a chaperone modulating the levels (and likely structure) of proteins involved in phase separation. Consistent with our conclusion, the Zweckstetter lab recently reported that *PPIA* can modify the ability of a protein to phase separate *in vitro* [16]. However, providing a detailed structural understanding of *PPIA*'s effect on substrate proteins *in vivo* is beyond the scope of our manuscript. Future studies will be required to thoroughly address the reviewer's question.

Minor points

10) PPIA Kd2 and Kd1 (293T) show major difference in expression pattern (Figure 2f). The author could a) cluster the target proteins based on function, b) should represent fold change and significance of target proteins relative to control.

Please see response to point 8 and **Fig. C** above.

11) Extended data Fig. 3c: Lower right blot midsection is unlabeled. Is that correct?

The unlabeled lanes of the total protein stain were not used for the immunoblot on the left side. We removed this subpanel for clarification.

Reviewer #3 (Remarks to the Author):

Comment on 'Cyclophilin A supports translation of intrinsically disordered proteins and mitigates haematopoietic stem cell aging' by Maneix et al.

In this paper, the authors have identified Peptidyl-3 Prolyl Isomerase A (PPIA or Cyclophilin A) as a dominant chaperone in haematopoietic stem and progenitor cells (HSPC) as it accounts for 14% of the cytosolic proteome and its mRNA accounts for over 0.5% of all mRNAs of HSPC. Total knockout of PPIA leads to 1) myeloid skewing, an immunophenotypic but not functional increase in stem cells, and 2) impaired self-renewal with 3) accelerated exhaustion, which are three hallmarks of haematopoietic aging. Using an assay targeting differential co-immunoprecipitation between the wild-type PPIA and a G104A mutant PPIA, the authors revealed approximately 400 substrates of the wild-type enzyme. Since they performed the co-immunoprecipitation in the cytosolic cell fraction, these results suggest PPIA and substrates interact between translation and nuclear translocation. When compared to the global proteome, immunoprecipitated PPIA substrates feature higher levels of IDRs.

By surveying proline residues within PDB, the authors found that the majority of cis-prolines located in bends and turns. Their conclusion is that "PPIA and related enzymes predominantly isomerize prolines within unstructured protein regions." This conclusion is incorrect as the bends and turns are categorized as structured regions that connecting helices and beta-strands. "Unstructured protein regions" are regions of proteins that they do not have stable (i.e. highly populated) conformations and are "invisible" in PDB. Further studies are required to justify this conclusion.

We agree and have eliminated these data from the revised manuscript.

Using pulsed SILAC in HeLa and 293T cells with either normal or reduced levels of PPIA, the authors found that loss of PPIA significantly reduced expression of the whole proteome, PPIA clients and non-clients, in both cell types. As far as the levels of PPIA target proteins are concerned, they were lower than for other proteins and further reduced when PPIA was knockdown. In addition, the fraction of protein IDRs in the HSPC proteome of PPIA knockout animals was significantly reduced when compared with heterozygous siblings.

Largely based on these aforementioned observations, the authors claim that PPIA supports *de novo* translation of its target proteins. Unfortunately, this claim is not even close to be supported by the data reported.

We disagree with reviewer #3's notion. On the contrary, a substantial body of literature supports this assertion:

First, early characterizations of PPIA activity suggest the potential role of prolyl isomerization in co-translational protein folding [17, 18].

Second, comprehensive research in prokaryotes revealed that trigger factor, a major prolyl isomerase in the biosphere, participates in co-translational isomerization [19].

Third, various studies demonstrated unequivocally that PPIA inhibitors obstruct protein translation [20, 21]. Our experiments corroborate these findings through both genetic and pharmacologic PPIA inhibition, observing significant impairment in *de novo* translation via SILAC, and in orthogonal puromycin-incorporation assays (Fig. 4a-c and new Extended Data Fig. 3a).

Fourth, utilizing a tRNA analog as a UV-activated chemical trap, PPIA has been located in close vicinity to functional ribosomes [22].

Fifth, the transcriptional profile of *Ppia* knockout animals, exhibiting increased expression of cytosolic ribosome genes, implies a compensatory mechanism responding to a translation block (new Extended Data Fig. 3f).

Sixth, in proximity ligation assays, we discerned interaction between cytosolic PPIA and NPM1, which is primarily found in the nucleus, suggesting their engagement prior to NPM1's nuclear import (new Fig. 5d).

Seventh, previous work showed that PPIA enhances the expression of alpha-synuclein [2], supporting our global analysis that PPIA enhances substrate levels.

In the revised manuscript, we have emphasized this prior research. Together with our new findings that PPIA is transcriptionally linked to the cytoplasmic translation machinery and acutely affects translation rates (new Extended Data Fig. 3a,d), our results support the overall interpretation and conclusion that PPIA interacts with nascent polypeptides.

The authors went on to confirmed a few clients of PPIA using a combination of IP and western blotting. One of the confirmed clients is PABPC1. “Following PPIA knockdown, expression of PABPC1 protein was reduced by 20-30% in pulsed SILAC experiments, suggesting the chaperone engages with PABPC1 during translation.” This conclusion is questionable as the pulse was 24 hours. Many other mechanisms other than, or in addition to, limited translation can account for the 20-30% reduction.

We did not find enhanced decay of PPIA substrates in PPIA-depleted cells following chase with cycloheximide for up to 24 hours (data not shown). We also did not observe increased protein aggregation or activation of the ubiquitin-proteasome system (new Extended Data Fig. 3d,g). Furthermore, puromycin incorporation studies indicated reduced translation efficiencies following acute PPIA inhibition even during short 2-hour chase experiments. However, to address the reviewer’s concern, we have revised our manuscript to discuss potential alternative effects of PPIA. Furthermore, we are amenable to changing the title of the manuscript to: “Cyclophilin A Preferentially Targets Intrinsically Disordered Proteins and Mitigates Haematopoietic Aging”.

Next, the authors turned their attention to liquid-liquid phase separation (LLPS), a biophysical property of multivalent proteins including many IDR-containing ones. Essentially the authors showed that PPIA KD perturbed condensates of PABPC1-GFP (I have a hard time to figure whether PABPC1-GFP is transiently expressed or is expressed at endogenous levels) and the formation of stress granules marked by G3BP1 upon the treatment with sodium arsenite. It is unclear whether the reduced translation of PPIA’s clients or the lack of PPIA activity per se was claimed to cause the perturbation to condensates of PABPC1-GFP or stress granule formation. Nevertheless LLPS is just one aspect of the global proteome

We concur that liquid-liquid phase separation (LLPS) represents only one facet of the global proteome. However, it is a common feature of all three PPIA substrates investigated here. To our knowledge, our preprinted manuscript describes the first identification of a chaperone that prefers to engage with intrinsically disordered proteins (IDPs) [23, 24], many of which partake in LLPS. All three PPIA substrates are vital and have well-documented functions in mRNA translation, transcript turnover, ribosome biogenesis, and nuclear compartmentalization. Furthermore, the ability of these proteins to phase separate is intimately linked to their cellular function as evident from the current literature.

For the purpose of elucidation, it should be noted that the phase separation of stress granules, P-bodies, and nucleoli in Fig. 5 was analyzed at endogenous levels without overexpression of any of the proteins involved (for NPM1, endogenous tagging was utilized; DDX6, and G3BP were monitored using antibodies). According to consensus in the field, phase separation is heavily contingent on the concentration of the implicated proteins [15]. It is critical to point out that both expression levels and structural conformation of PPIA substrates are likely integral to their demixing properties, which we elaborated upon in the revised discussion. Only in the supplementary videos did we employ an overexpression system, by introducing GFP-tagged PABPC1.

Overall, the authors didn’t make sufficient effort to connect pieces of observations for a coherent model explaining the role of PPIA in ageing of haematopoietic stem cells. I can’t recommend its further consideration for pulication in Nature Cell Biology.

We respectfully disagree with reviewer #3’s assessment, which is not shared by the other two reviewers, who considered our work “*novel*” and “*well-presented*” (reviewer #1) and an “*interesting topic*” (reviewer #2). We wish to reiterate that our work makes a significant contribution to the field in two ways: First, we identified PPIA as a principal chaperone of disordered proteins. Second, we detected a shift in proteome composition during

haematopoietic aging, reducing the ratio of disordered proteins. The latter implies that proteome alterations arise not only from functional changes in gene transcription, but also from transitions in the structural composition of polypeptides that disfavor intrinsically disordered proteins.

We have performed additional experiments to more robustly link PPIA to changes in the proteome composition:

1. Loss of intrinsically disordered proteins (IDPs) increases with age and with PPIA deficiency in the haematopoietic compartment (new Extended Data Figs. 4b-e and 5a-c).
2. Increased expression of PPIA in aged haematopoietic stem cells partially rescues the aging phenotype, arguing for a causal role of the PPIA chaperone in this tissue compartment (new Fig. 2).
3. Loss of PPIA does not trigger a protein stress response, but instead activates the cytoplasmic translation machinery (new Extended Data Fig. 3d,g), indicating a function of this chaperone in protein synthesis. Consistently, acute inhibition of PPIA impairs translation in primary haematopoietic stem cells (new Extended Data Fig. 3a).
4. Proteomic studies were validated with detailed quantitative and functional investigations of three PPIA substrates (new Figs. 4f,g, 5b,c, and new Extended Data Fig. 3b).

To help them improving their study for future submission, I listed some suggestions below:

Major issues

1) When doing differential analysis of proteome to assess the effects of PPIA on IDR-enriched clients, the authors shall divide PPIA clients and nonclients into IDR-containing, IDR-noncontaining, four categories.

We appreciate the reviewer's suggestion and utilized two orthogonal clustering methods when comparing the proteome in dependence of PPIA expression. We analyzed the proteome of both *Ppia*-expressing or knockout haematopoietic cells and considered both their genotypes as well as the expression levels of intrinsically disordered proteins. Both methodologies yielded congruent results, indicating that IDR-rich proteins are particularly sensitive to PPIA loss and that PPIA disproportionately affects IDR-rich proteins (new Extended Data Fig. 5a-c). In contrast to the SILAC experiments that employ acute *PPIA* knockdown (Fig. 4c), we did not observe a comparable loss of PPIA target proteins in the constitutive knockout model. This is presumably due to transcriptional compensation (new Extended Data Fig. 3f) and the upregulation of genes encoding PPIA substrates (see our response to Reviewer 2's query #8).

2) Genetic manipulation or pharmacological perturbation for days are rather slow to assess acute and direct functions. I suggest the authors to use an AID system or PROTAC system (if available) to chemically KD PPIA and assess whether the reported effects are direct or indirect.

Beyond genetic perturbation, we utilized pharmacological intervention with the PPIA-specific inhibitor TMN355 [25]. While degron-based systems are an option, they can be leaky and require high doses of auxin [26]. Prior studies have demonstrated that PPIA inhibitors obstruct stress granule formation [13]. Using TMN355, we find that acute PPIA inhibition (24-48 h) disrupts nucleoli formation and protein translation (new Extended Data Fig. 3a and data not shown). However, we did not observe a substantial alteration in the number and morphology of nucleoli or stress granules during shorter treatment. This is consistent with previously published work [13] and may be explained by the slow synthesis rates of PPIA substrates [12]. These findings suggest that PPIA influences the protein function of newly synthesized polypeptides, which drive phase separation, rather than affecting the functionality of pre-existing membraneless organelles.

Minor issues

1) There are no legends for SI-Videos.

Legends for SI-Videos are now included in the revised manuscript and can be found with the Extended Data.

REFERENCES:

- 1: Cardenas ME, Lim E, Heitman J. Mutations that perturb cyclophilin A ligand binding pocket confer cyclosporin A resistance in *Saccharomyces cerevisiae*. *J Biol Chem*. 1995 Sep 8;270(36):20997-1002. doi: 10.1074/jbc.270.36.20997. PMID: 7673124.
- 2: Favretto F, Flores D, Baker JD, Strohäker T, Andreas LB, Blair LJ, Becker S, Zweckstetter M. Catalysis of proline isomerization and molecular chaperone activity in a tug-of-war. *Nat Commun*. 2020 Nov 27;11(1):6046. doi: 10.1038/s41467-020-19844-0. PMID: 33247146; PMCID: PMC7695863.
- 3: Zaro BW, Noh JJ, Mascetti VL, Demeter J, George B, Zukowska M, Gulati GS, Sinha R, Flynn RA, Banuelos A, Zhang A, Wilkinson AC, Jackson P, Weissman IL. Proteomic analysis of young and old mouse hematopoietic stem cells and their progenitors reveals post-transcriptional regulation in stem cells. *Elife*. 2020 Nov 25;9:e62210. doi: 10.7554/eLife.62210. PMID: 33236985; PMCID: PMC7688314.
- 4: Mészáros B, Erdos G, Dosztányi Z. IUPred2A: context-dependent prediction of protein disorder as a function of redox state and protein binding. *Nucleic Acids Res*. 2018 Jul 2;46(W1):W329-W337. doi: 10.1093/nar/gky384. PMID: 29860432; PMCID: PMC6030935.
- 5: Tsherniak A, Vazquez F, Montgomery PG, Weir BA, Kryukov G, Cowley GS, Gill S, Harrington WF, Pantel S, Krill-Burger JM, Meyers RM, Ali L, Goodale A, Lee Y, Jiang G, Hsiao J, Gerath WFJ, Howell S, Merkel E, Ghandi M, Garraway LA, Root DE, Golub TR, Boehm JS, Hahn WC. Defining a Cancer Dependency Map. *Cell*. 2017 Jul 27;170(3):564-576.e16. doi: 10.1016/j.cell.2017.06.010. PMID: 28753430; PMCID: PMC5667678.
- 6: Colgan J, Asmal M, Luban J. Isolation, characterization and targeted disruption of mouse ppia: cyclophilin A is not essential for mammalian cell viability. *Genomics*. 2000 Sep 1;68(2):167-78. doi: 10.1006/geno.2000.6295. PMID: 10964515.
- 7: Tabula Muris Consortium. A single-cell transcriptomic atlas characterizes ageing tissues in the mouse. *Nature*. 2020 Jul;583(7817):590-595. doi: 10.1038/s41586-020-2496-1. Epub 2020 Jul 15. PMID: 32669714; PMCID: PMC8240505.
- 8: Chambers SM, Shaw CA, Gatz C, Fisk CJ, Donehower LA, Goodell MA. Aging hematopoietic stem cells decline in function and exhibit epigenetic dysregulation. *PLoS Biol*. 2007 Aug;5(8):e201. doi: 10.1371/journal.pbio.0050201. PMID: 17676974; PMCID: PMC1925137.
- 9: Flohr Svendsen A, Yang D, Kim K, Lazare S, Skinder N, Zwart E, Mura-Meszaros A, Ausema A, von Eyss B, de Haan G, Bystrykh L. A comprehensive transcriptome signature of murine hematopoietic stem cell aging. *Blood*. 2021 Aug 12;138(6):439-451. doi: 10.1182/blood.2020009729. PMID: 33876187.
- 10: López-Otín C, Blasco MA, Partridge L, Serrano M, Kroemer G. Hallmarks of aging: An expanding universe. *Cell*. 2023 Jan 19;186(2):243-278. doi: 10.1016/j.cell.2022.11.001. Epub 2023 Jan 3. PMID: 36599349.
- 11: Liang H, Dong S, Fu W, Zhang S, Yu W, Dong F, He B, Wang J, Gao Y, Zhou Y, Ru Y. Deciphering the Heterogeneity of Mitochondrial Functions During Hematopoietic Lineage Differentiation. *Stem Cell Rev Rep*. 2022 Aug;18(6):2179-2194. doi: 10.1007/s12015-022-10354-8. Epub 2022 Feb 21. PMID: 35188601.
- 12: Schwanhäusser B, Busse D, Li N, Dittmar G, Schuchhardt J, Wolf J, Chen W, Selbach M. Global quantification of mammalian gene expression control. *Nature*. 2011 May 19;473(7347):337-42. doi: 10.1038/nature10098. Erratum in: *Nature*. 2013 Mar 7;495(7439):126-7. PMID: 21593866.
- 13: Daito T, Watashi K, Sluder A, Ohashi H, Nakajima S, Borroto-Esoda K, Fujita T, Wakita T. Cyclophilin inhibitors reduce phosphorylation of RNA-dependent protein kinase to restore expression of IFN-stimulated genes in HCV-infected cells. *Gastroenterology*. 2014 Aug;147(2):463-72. doi: 10.1053/j.gastro.2014.04.035. Epub 2014 Apr 29. PMID: 24786893.

- 14: Xiang S, Kato M, Wu LC, Lin Y, Ding M, Zhang Y, Yu Y, McKnight SL. The LC Domain of hnRNPA2 Adopts Similar Conformations in Hydrogel Polymers, Liquid-like Droplets, and Nuclei. *Cell*. 2015 Nov 5;163(4):829-39. doi: 10.1016/j.cell.2015.10.040. PMID: 26544936; PMCID: PMC4879888.
- 15: Riback JA, Zhu L, Ferrolino MC, Tolbert M, Mitrea DM, Sanders DW, Wei MT, Kriwacki RW, Brangwynne CP. Composition-dependent thermodynamics of intracellular phase separation. *Nature*. 2020 May;581(7807):209-214. doi: 10.1038/s41586-020-2256-2. Epub 2020 May 6. PMID: 32405004; PMCID: PMC7733533.
- 16: Babu M, Favretto F, Rankovic M, Zweckstetter M. Peptidyl Prolyl Isomerase A Modulates the Liquid-Liquid Phase Separation of Proline-Rich IDPs. *J Am Chem Soc*. 2022 Sep 7;144(35):16157-16163. doi: 10.1021/jacs.2c07149. Epub 2022 Aug 26. PMID: 36018855; PMCID: PMC9460772.
- 17: Lang K, Schmid FX, Fischer G. Catalysis of protein folding by prolyl isomerase. *Nature*. 1987 Sep 17-23;329(6136):268-70. doi: 10.1038/329268a0. PMID: 3306408.
- 18: Kiefhaber T, Quaas R, Hahn U, Schmid FX. Folding of ribonuclease T1. 2. Kinetic models for the folding and unfolding reactions. *Biochemistry*. 1990 Mar 27;29(12):3061-70. doi: 10.1021/bi00464a024. PMID: 2110824.
- 19: Hartl FU, Hayer-Hartl M. Molecular chaperones in the cytosol: from nascent chain to folded protein. *Science*. 2002 Mar 8;295(5561):1852-8. doi: 10.1126/science.1068408. PMID: 11884745.
- 20: Buss WC, Stepanek J, Bennett WM. Proposed mechanism of cyclosporine toxicity: inhibition of protein synthesis. *Transplant Proc*. 1988 Jun;20(3 Suppl 3):863-7. PMID: 3388521.
- 21: Buss WC, Stepanek J. Characterization of the inhibition of renal translation in the Sprague-Dawley rat following in vivo cyclosporin A. *Int J Immunopharmacol*. 1993 Jan;15(1):63-76. doi: 10.1016/0192-0561(93)90032-t. PMID: 8432624.
- 22: Kandala DT, Del Piano A, Minati L, Clamer M. Targeting Translation Activity at the Ribosome Interface with UV-Active Small Molecules. *ACS Omega*. 2019 Jun 13;4(6):10336-10345. doi: 10.1021/acsomega.9b00366. PMID: 31460127; PMCID: PMC6648492.
- 23: Maneix L, Iakova P, Moree SE, King JC, Sykes DB, Hill CT, Saez B, Spooner E, Krause DS, Sahin E, Berk BC, Scadden DT, Catic A. Cyclophilin A regulates protein phase separation and mitigates haematopoietic stem cell aging. 2021 February. *BioRxiv*. doi: 10.1101/2021.02.24.432737.
- 24: Hegyi H, Tompa P. Intrinsically disordered proteins display no preference for chaperone binding in vivo. *PLoS Comput Biol*. 2008 Mar 7;4(3):e1000017. doi: 10.1371/journal.pcbi.1000017. PMID: 18369417; PMCID: PMC2265518.
- 25: Ni S, Yuan Y, Huang J, Mao X, Lv M, Zhu J, Shen X, Pei J, Lai L, Jiang H, Li J. Discovering potent small molecule inhibitors of cyclophilin A using de novo drug design approach. *J Med Chem*. 2009 Sep 10;52(17):5295-8. doi: 10.1021/jm9008295. PMID: 19691347.
- 26: Yesbolatova A, Saito Y, Kitamoto N, Makino-Itou H, Ajima R, Nakano R, Nakaoka H, Fukui K, Gamo K, Tominari Y, Takeuchi H, Saga Y, Hayashi KI, Kanemaki MT. The auxin-inducible degron 2 technology provides sharp degradation control in yeast, mammalian cells, and mice. *Nat Commun*. 2020 Nov 11;11(1):5701. doi: 10.1038/s41467-020-19532-z. PMID: 33177522; PMCID: PMC7659001.
- 27: Wilkinson AC, Ishida R, Nakauchi H, Yamazaki S. Long-term ex vivo expansion of mouse hematopoietic stem cells. *Nat Protoc*. 2020 Feb;15(2):628-648. doi: 10.1038/s41596-019-0263-2. Epub 2020 Jan 8. PMID: 31915389; PMCID: PMC7206416.

Decision Letter, first revision:

*Please delete the link to your author homepage if you wish to forward this email to co-authors.

Dear Dr Catic,

I apologize once again for this very long delay. Furthermore, we had also gone back to Reviewer #1 for further clarification on their report. Your manuscript, "Cyclophilin A supports translation of intrinsically disordered proteins and mitigates haematopoietic stem cell aging", has now been seen by our original referees, who are experts in stem cell senescence (referee 1); proteostasis and proteomics (referee 2); and biomolecular condensation (referee 3). As you will see from their comments (attached below) they find this work of interest, but have raised some important points. Although we are also very interested in this study, we believe that their concerns should be addressed before we can consider publication in Nature Cell Biology.

Nature Cell Biology editors discuss the referee reports in detail within the editorial team, including the chief editor, to identify key referee points that should be addressed with priority, and requests that are overruled as being beyond the scope of the current study. To guide the scope of the revisions, I have listed these points below. We are committed to providing a fair and constructive peer-review process, so please feel free to contact me if you would like to discuss any of the referee comments further.

In particular, it would be essential to:

- A) If the data can be obtained within this revision's timeframe (1 month), then we'd ask for providing this data regarding potential effects of PPI2 overexpression on stem cell compartments with immunostaining of cell surface markers of HSCs/MPPs (Reviewer #1). We hope this data can be obtained within a reasonable timeframe and please get in touch with us if you would like to discuss.
- B) Provide clear methodology for the proteomics analysis (Reviewer #2) and please ensure that this is now available on a recognized repository with at least reviewer access credentials in your revised Data Availability Statement and Reporting Summary.
- C) Tone down as necessary any claims of biomolecular condensation as the sole or major driver of aging, highlighting these as potentials and avenues for future research (reviewer #3).
- D) Please note that while $n=3$ or greater will be acceptable for statistical analysis (in contrast to Reviewer #1's report), please ensure that all n values are described clearly in your text. Please also note that statistical analysis of $n<3$ are not appropriate (Reviewer #2).
- E) All other referee concerns pertaining to strengthening existing data, providing controls, methodological details, clarifications and textual changes, should also be addressed.
- F) Finally please pay close attention to our guidelines on statistical and methodological reporting

(listed below) as failure to do so may delay the reconsideration of the revised manuscript. In particular please provide:

- a Supplementary Figure including unprocessed images of all gels/blots in the form of a multi-page pdf file. Please ensure that blots/gels are labeled and the sections presented in the figures are clearly indicated.
- a Supplementary Table including all numerical source data in Excel format, with data for different figures provided as different sheets within a single Excel file. The file should include source data giving rise to graphical representations and statistical descriptions in the paper and for all instances where the figures present representative experiments of multiple independent repeats, the source data of all repeats should be provided.

We therefore invite you to take these points into account when revising the manuscript. In addition, when preparing the revision please:

- ensure that it conforms to our format instructions and publication policies (see below and <https://www.nature.com/nature/for-authors>).
- provide a point-by-point rebuttal to the full referee reports verbatim, as provided at the end of this letter.
- provide the completed Reporting Summary (found here <https://www.nature.com/documents/nr-reporting-summary.pdf>). This is essential for reconsideration of the manuscript and will be available to editors and referees in the event of peer review. For more information see <http://www.nature.com/authors/policies/availability.html> or contact me.

When submitting the revised version of your manuscript, please pay close attention to our [href="https://www.nature.com/nature-portfolio/editorial-policies/image-integrity">Digital Image Integrity Guidelines](https://www.nature.com/nature-portfolio/editorial-policies/image-integrity). and to the following points below:

Nature Cell Biology is committed to improving transparency in authorship. As part of our efforts in this direction, we are now requesting that all authors identified as 'corresponding author' on published papers create and link their Open Researcher and Contributor Identifier (ORCID) with their account on the Manuscript Tracking System (MTS), prior to acceptance. ORCID helps the scientific community achieve unambiguous attribution of all scholarly contributions. You can create and link your ORCID

from the home page of the MTS by clicking on 'Modify my Springer Nature account'. For more information please visit www.springernature.com/orcid.

This journal strongly supports public availability of data. Please place the data used in your paper into a public data repository, or alternatively, present the data as Supplementary Information. If data can only be shared on request, please explain why in your Data Availability Statement, and also in the correspondence with your editor. Please note that for some data types, deposition in a public repository is mandatory - more information on our data deposition policies and available repositories appears below.

[Redacted]

We would like to receive the revision within four weeks. If submitted within this time period, reconsideration of the revised manuscript will not be affected by related studies published elsewhere, or accepted for publication in Nature Cell Biology in the meantime. We would be happy to consider a revision even after this timeframe, but in that case we will consider the published literature at the time of resubmission when assessing the file.

We hope that you will find our referees' comments, and editorial guidance helpful. Please do not hesitate to contact me if there is anything you would like to discuss.

Best wishes,

Daryl

Daryl Jason Verzosa David, PhD

Senior Editor, Nature Cell Biology
Nature Portfolio

Heidelberger Platz 3, 14197 Berlin, Germany
Email: daryl.david@nature.com
ORCID: <https://orcid.org/0000-0002-9253-4805>

Reviewers' Comments:

Reviewer #1:

Remarks to the Author:

This Reviewer recognizes that the authors have done substantial work to address the raised concerns and discussed potential limitations of their study, which is good. This includes the challenges of performing more mechanistic studies and applying mass spec analysis in the relevant cell type (e.g. HSC) under study. Some questions remain on new data included, for example, the data presented in Figure 2 are important, but it would be great if the authors could better characterize the impact of PPI2 overexpression on the stem cell compartment as well, together with the lineage output presented. As a general consideration, statistical analysis with $n=3$ should not be presented and instead, descriptive statistics should be applied.

Reviewer #2:

Remarks to the Author:

The authors have improved the manuscript and addressed most concerns. A couple of points remain:

There are still a number of panels that provide statistics without indicating the number of replicates. These need to be added. Please remove statistics for all panels with less than 3 replicates.

I cannot find the data (raw files deposited and processed data in table format) for pulsed SILAC. Please make this data available.

A description of how interactors were identified based on the proteomics data appears to still be missing. Please provide a clear indication and the relevant data.

For the supplementary tables, please add a tab for every table to provide a description of the data shown.

Reviewer #3:

Remarks to the Author:

The authors have made significant efforts to address my concerns. They have incorporated a substantial amount of crucial new data into the original version of the manuscript. As a result, I am now more convinced that PPIA plays a significant role in protein translation and that its levels are closely related to the aging phenotype. However, I still suggest that the authors moderate their emphasis on perturbed phase separation as the primary mechanism driving aging. Phase separation, ultimately, is influenced by and closely intertwined with protein-protein and protein-RNA interactions. There is a vast array of activities associated with the 400+ substrates that likely contribute to the aging process. Therefore, singling out phase separation as the "major" mechanism underlying aging may not be entirely appropriate.

Minor comments:

- 1) Do PPIA's substrates happen to contain higher abundance of proline residues?
- 2) Supplemental videos are not available to reviewers.

GUIDELINES FOR SUBMISSION OF NATURE CELL BIOLOGY ARTICLES

ARTICLE FORMAT

ABSTRACT – should not exceed 150 words and should be unreferenced. This paragraph is the most visible part of the paper and should briefly outline the background and rationale for the work, and accurately summarize the main results and conclusions. Key genes, proteins and organisms should be specified to ensure discoverability of the paper in online searches.

TEXT – the main text consists of the Introduction, Results, and Discussion sections and must not exceed 3500 words including the abstract. The Introduction should expand on the background relating to the work. The Results should be divided in subsections with subheadings, and should provide a concise and accurate description of the experimental findings. The Discussion should expand on the findings and their implications. All relevant primary literature should be cited, in particular when discussing the background and specific findings.

REFERENCES – are limited to a total of 70 in the main text and Methods combined,. They must be numbered sequentially as they appear in the main text, tables and figure legends and Methods and must follow the precise style of Nature Cell Biology references. References only cited in the Methods should be numbered consecutively following the last reference cited in the main text. References only associated with Supplementary Information (e.g. in supplementary legends) do not count toward the total reference limit and do not need to be cited in numerical continuity with references in the main text. Only published papers can be cited, and each publication cited should be included in the numbered reference list, which should include the manuscript titles. Footnotes are not permitted.

Methods should be written concisely, but should contain all elements necessary to allow interpretation and replication of the results. As a guideline, Methods sections typically do not exceed 3,000 words. The Methods should be divided into subsections listing reagents and techniques. When citing previous methods, accurate references should be provided and any alterations should be noted. Information must be provided about: antibody dilutions, company names, catalogue numbers and clone numbers for monoclonal antibodies; sequences of RNAi and cDNA probes/primers or company names and catalogue numbers if reagents are commercial; cell line names, sources and information on cell line identity and authentication. Animal studies and experiments involving human subjects must be reported in detail, identifying the committees approving the protocols. For studies involving human subjects/samples, a statement must be included confirming that informed consent was obtained. Statistical analyses and information on the reproducibility of experimental results should be provided in a section titled "Statistics and Reproducibility".

All Nature Cell Biology manuscripts submitted on or after March 21 2016, must include a Data availability statement as a separate section after Methods but before references, under the heading "Data Availability". For Springer Nature policies on data availability see <http://www.nature.com/authors/policies/availability.html>; for more information on this particular policy see <http://www.nature.com/authors/policies/data/data-availability-statements-data-citations.pdf>. The Data availability statement should include:

- Accession codes for primary datasets (generated during the study under consideration and designated as "primary accessions") and secondary datasets (published datasets reanalysed during the study under consideration, designated as "referenced accessions"). For primary accessions data should be made public to coincide with publication of the manuscript. A list of data types for which submission to community-endorsed public repositories is mandated (including sequence, structure, microarray, deep sequencing data) can be found here

<http://www.nature.com/authors/policies/availability.html#data>.

- Unique identifiers (accession codes, DOIs or other unique persistent identifier) and hyperlinks for datasets deposited in an approved repository, but for which data deposition is not mandated (see here for details <http://www.nature.com/sdata/data-policies/repositories>).
- At a minimum, please include a statement confirming that all relevant data are available from the authors, and/or are included with the manuscript (e.g. as source data or supplementary information), listing which data are included (e.g. by figure panels and data types) and mentioning any restrictions on availability.
- If a dataset has a Digital Object Identifier (DOI) as its unique identifier, we strongly encourage including this in the Reference list and citing the dataset in the Methods.

We recommend that you upload the step-by-step protocols used in this manuscript to the Protocol Exchange. More details can found at www.nature.com/protocolexchange/about.

DISPLAY ITEMS – main display items are limited to 6-8 main figures and/or main tables. For Supplementary Information see below.

FIGURES – Colour figure publication costs \$395 per colour figure. All panels of a multi-panel figure must be logically connected and arranged as they would appear in the final version. Unnecessary figures and figure panels should be avoided (e.g. data presented in small tables could be stated briefly in the text instead).

All imaging data should be accompanied by scale bars, which should be defined in the legend. Cropped images of gels/blots are acceptable, but need to be accompanied by size markers, and to retain visible background signal within the linear range (i.e. should not be saturated). The boundaries of panels with low background have to be demarked with black lines. Splicing of panels should only be considered if unavoidable, and must be clearly marked on the figure, and noted in the legend with a statement on whether the samples were obtained and processed simultaneously. Quantitative comparisons between samples on different gels/blots are discouraged; if this is unavoidable, it has to be performed for samples derived from the same experiment with gels/blots were processed in parallel, which needs to be stated in the legend.

- For line art, graphs, charts and schematics we prefer Adobe Illustrator (.AI), Encapsulated PostScript (.EPS) or Portable Document Format (.PDF). Files should be saved or exported as such directly from the application in which they were made, to allow us to restyle them according to our journal house style.
- We accept PowerPoint (.PPT) files if they are fully editable. However, please refrain from adding PowerPoint graphical effects to objects, as this results in them outputting poor quality raster art. Text used for PowerPoint figures should be Helvetica (preferred) or Arial.
- We do not recommend using Adobe Photoshop for designing figures, but we can accept Photoshop generated (.PSD or .TIFF) files only if each element included in the figure (text, labels, pictures, graphs, arrows and scale bars) are on separate layers. All text should be editable in 'type layers' and line-art such as graphs and other simple schematics should be preserved and embedded within 'vector smart objects' - not flattened raster/bitmap graphics.
- Some programs can generate Postscript by 'printing to file' (found in the Print dialogue). If using an application not listed above, save the file in PostScript format or email our Art Editor, Allen Beattie for advice (a.beattie@nature.com).

Regardless of format, all figures must be vector graphic compatible files, not supplied in a flattened raster/bitmap graphics format, but should be fully editable, allowing us to highlight/copy/paste all text and move individual parts of the figures (i.e. arrows, lines, x and y axes, graphs, tick marks, scale bars etc). The only parts of the figure that should be in pixel raster/bitmap format are photographic images or 3D rendered graphics/complex technical illustrations.

Unprocessed scans of all key data generated through electrophoretic separation techniques need to be presented in a supplementary figure that should be labeled and numbered as the final supplementary figure, and should be mentioned in every relevant figure legend. This figure does not count towards the total number of figures and is the only figure that can be displayed over multiple pages, but should be provided as a single file, in PDF or TIFF format. Data in this figure can be displayed in a relatively informal style, but size markers and the figures panels corresponding to the presented data must be indicated.

The total number of Supplementary Figures (not including the “unprocessed scans” Supplementary Figure) should not exceed the number of main display items (figures and/or tables (see our Guide to Authors and March 2012 editorial <http://www.nature.com/ncb/authors/submit/index.html#suppinfo>; <http://www.nature.com/ncb/journal/v14/n3/index.html#ed>). No restrictions apply to Supplementary Tables or Videos, but we advise authors to be selective in including supplemental data.

GUIDELINES FOR EXPERIMENTAL AND STATISTICAL REPORTING

REPORTING REQUIREMENTS – We ask authors to complete a Reporting Summary that collects information on experimental design and reagents. We hope this will aid in your evaluation of the paper. The Reporting Summary can be found here <https://www.nature.com/documents/nr-reporting-summary.pdf>) Please note that these forms are dynamic ‘smart pdfs’ and must therefore be downloaded and completed in Adobe Reader. We will then flatten them for ease of use. If you would like to reference the guidance text as you complete the template, please access these flattened versions at <http://www.nature.com/authors/policies/availability.html>.

Author Rebuttal, first revision:

We would like to express our sincere gratitude to the reviewers for their continued dedication in evaluating our manuscript. We value their insightful feedback and are eager to address the outstanding concerns in the revised manuscript and in our point-by-point response to their queries. For clarity and to facilitate a seamless review process, we have highlighted our responses below and the relevant modifications to the manuscript in blue font.

Reviewer #1

Remarks to the Author:

This Reviewer recognizes that the authors have done substantial work to address the raised concerns and discussed potential limitations of their study, which is good. This includes the challenges of performing more mechanistic studies and applying mass spec analysis in the relevant cell type (e.g. HSC) under study. Some questions remain on new data included, for example, the data presented in Figure 2 are important, but it would be great if the authors could better characterize the impact of PPI2 overexpression on the stem cell compartment as well, together with the lineage output presented. As a general consideration, statistical analysis with $n=3$ should not be presented and instead, descriptive statistics should be applied.

We thank this reviewer for the thorough assessment of our work. We concur that several questions warrant further exploration, such as the changes in stem cells (alterations in proliferation versus apoptosis), variations in progenitor cell pools, and whether blood cell functions are altered beyond their output numbers. Our experimental design aimed to ensure the most rigorous phenotypic outcomes. In our study, modified haematopoietic stem cells were transplanted into lethally irradiated recipient mice, and the outcomes were monitored in the peripheral blood over a six-month period. This duration is commonly recognized in the field to capture phenotypic differences attributable to the long-term stem cell pool, thus enabling us to attribute the observed results to this cell type.

Regrettably, due to natural attrition of the transplanted animals following submission of the revised manuscript in July, we are unable to further explore how PPIA overexpression impacted the progenitor pool within the context of this specific experiment. We appreciate the reviewer's comment and plan to bridge this knowledge gap in future work. However, experiments to comprehensively address these queries would require a substantial amount of time and effort, which are beyond the scope of the current manuscript. Importantly, we note that these results, although desirable, would not fundamentally alter our conclusions derived from the experiment presented in Fig. 2. The updated discussion addresses our study's limitations with regard to the progenitor and immune cell pool.

Following conversations with the Nature Cell Biology editorial team, we have taken additional steps to clarify the distinction between "sample size" and "independent biological replicate" as stated in our figure legends. In our previous manuscript, the term 'n' denoted the number of times the experiment was conducted independently, rather than representing sample size or technical replicates. The revised manuscript does not feature experiments with sample sizes of less than three, and we agree with the reviewer that such a sample size would preclude the use of statistical significance testing.

Reviewer #2

Remarks to the Author:

The authors have improved the manuscript and addressed most concerns. A couple of points remain:

There are still a number of panels that provide statistics without indicating the number of replicates. These need to be added. Please remove statistics for all panels with less than 3 replicates.

We appreciate the reviewer's constructive feedback. We have clarified the distinction between independent experiments and sample size in our revised manuscript (see our response to reviewer #1). Furthermore, we agree that sample sizes less than three, which are not present in our manuscript, are not suitable for significance testing.

I cannot find the data (raw files deposited and processed data in table format) for pulsed SILAC. Please make this data available.

We apologize for this oversight. The SILAC and isobaric proteomics data have now been deposited with the MassIVE and ProteomeXchange public repositories. The data are accessible without any restrictions. They can be accessed using the following identifiers with the MassIVE Repository and the ProteomeXchange Consortium:

MSV000093125 / PXD046245

MSV000093126 / PXD046246

MSV000093127 / PXD046247

Additionally, the "Source Data" Excel file includes all numerical values for the graphs presented in this manuscript. Separate Excel files for the proteomics and transcriptomics data are also included as Supplementary Information (SI) files. Further, the transcriptomic data within the GEO dataset (GSE151125) has been released for public access.

A description of how interactors were identified based on the proteomics data appears to still be missing. Please provide a clear indication and the relevant data.

In the methods section, we have provided additional details regarding the comparative analysis of the pulldown experiments using the wild-type PPIA and the G104A mutant, respectively. The numerical values of peptide spectral matches (PSM) are presented in the updated Supplementary Information Excel file "SI-PPIA client proteins".

For the supplementary tables, please add a tab for every table to provide a description of the data shown.

We have added a brief text box in each Supplementary Information (SI) Excel file, offering a succinct description of the data. More detailed explanations can be found in the methods section of the manuscript and the figure legends.

Reviewer #3

Remarks to the Author:

The authors have made significant efforts to address my concerns. They have incorporated a substantial amount of crucial new data into the original version of the manuscript. As a result, I am now more convinced that PPIA plays a significant role in protein translation and that its levels are closely related to the aging phenotype.

We appreciate the reviewer's constructive criticism and suggestions, and for pointing out needed improvements, which strengthened our paper.

However, I still suggest that the authors moderate their emphasis on perturbed phase separation as the primary mechanism driving aging. Phase separation, ultimately, is influenced by and closely intertwined with protein-protein and protein-RNA interactions. There is a vast array of activities associated with the 400+ substrates that likely contribute to the aging process. Therefore, singling out phase separation as the "major" mechanism underlying aging may not be entirely appropriate.

We concur. We have revised our manuscript, in particular the discussion section, to reflect a more nuanced understanding of the role of phase separation in the aging process. We now emphasize that this process may contribute to aging without necessarily being the primary or predominant factor.

Minor comments:

1) *Do PPIA's substrates happen to contain higher abundance of proline residues?*

We assessed the proline ratio in PPIA substrates vs non-substrates, which we identified by co-IP/MS. Our findings indicate that PPIA-targeted proteins do not exhibit elevated proline ratios on average (see Figure left panel). However, this does not discount the presence of low-complexity regions within these proteins where proline may be locally highly enriched. Prolines induce disorder and are disproportionately represented in intrinsically disordered regions [Theillet *et al.*, 2013]. Indeed, previous studies have demonstrated that PPIA can catalyze isomerization within protein segments that are proline-rich [Favretto *et al.*, 2020].

A limitation of our analysis is that it does not take into account protein abundance. To address this, we normalized the proline ratio based on the abundance of each protein. As a result, the relative proline ratio in more abundant proteins outweighs that in less abundant ones. Following this normalization, we observed that PPIA substrates show a significantly higher overall proline abundance (see Figure right panel).

Figure: Proline ratios in the proteome. Left Panel: Depicts the median proline ratio in the human proteome, excluding PPIA substrates, at 0.057, in contrast to a median ratio of 0.047 in PPIA substrates ($p < 0.0001$). Right Panel: Demonstrates the abundance-normalized proline ratio in young mouse haematopoietic stem and progenitor cells, as referenced from our supplementary data (SI-365 Proteomics Profiling.xls). This panel highlights an enhanced proline ratio within PPIA substrates, attributable to the relatively high expression of proline-rich proteins. This results in an overall higher proline abundance of PPIA substrates ($p < 0.0079$). Two-sided Wilcoxon rank-sum test used.

In response to earlier critiques, we have omitted results related to the potential effects of primary or secondary structures on Cyclophilin A substrate affinity, and as such, this data does not appear in the final manuscript.

2) *Supplemental videos are not available to reviewers.*

We apologize and have informed the editorial team to release the supplementary videos.

References:

- Theillet FX, Kalmar L, Tompa P, Han KH, Selenko P, Dunker AK, Daughdrill GW, Uversky VN. The alphabet of intrinsic disorder: I. Act like a Pro: On the abundance and roles of proline residues in intrinsically disordered proteins. *Intrinsically Disord Proteins*. 2013 Apr 1;1(1):e24360. doi: 10.4161/idp.24360. PMID: 28516008; PMCID: PMC5424786.
- Favretto F, Flores D, Baker JD, Strohäker T, Andreas LB, Blair LJ, Becker S, Zweckstetter M. Catalysis of proline isomerization and molecular chaperone activity in a tug-of-war. *Nat Commun*. 2020 Nov 27;11(1):6046. doi: 10.1038/s41467-020-19844-0. PMID: 33247146; PMCID: PMC7695863.

Decision Letter, second revision:

Our ref: NCB-A48335B

13th December 2023

Dear Dr. Catic,

Thank you for submitting your revised manuscript "Cyclophilin A supports translation of intrinsically disordered proteins and mitigates haematopoietic stem cell aging" (NCB-A48335B). It has now been seen by the original referees and their comments are below. The reviewers find that the paper has improved in revision, and therefore we'll be happy in principle to publish it in Nature Cell Biology, pending minor revisions to satisfy the referees' final requests and to comply with our editorial and formatting guidelines.

The current version of your manuscript is in a PDF format, so please email us a copy of the file in an editable format (Microsoft Word or LaTeX)-- we can not proceed with PDFs at this stage.

Thank you again for your interest in Nature Cell Biology Please do not hesitate to contact me if you have any questions.

Sincerely,
Daryl

Daryl Jason Verzosa David, PhD

Senior Editor, Nature Cell Biology
Nature Portfolio

Heidelberger Platz 3, 14197 Berlin, Germany
Email: daryl.david@nature.com
ORCID: <https://orcid.org/0000-0002-9253-4805>

Reviewer #1 (Remarks to the Author):

The authors have addressed my concerns even if not experimentally.
I am now in favor of publication.

Reviewer #2 (Remarks to the Author):

The authors have addressed my concerns sufficiently. Congratulations on an interesting story!

Reviewer #3 (Remarks to the Author):

The authors addressed my concerns. I don't have further comments.

Decision Letter, final checks:

Our ref: NCB-A48335B

11th January 2024

Dear Dr. Catic,

Thank you for your patience as we've prepared the guidelines for final submission of your Nature Cell Biology manuscript, "Cyclophilin A supports translation of intrinsically disordered proteins and mitigates haematopoietic stem cell aging" (NCB-A48335B). Please carefully follow the step-by-step instructions provided in the attached file, and add a response in each row of the table to indicate the changes that you have made. Please also check and comment on any additional marked-up edits we have proposed within the text. Ensuring that each point is addressed will help to ensure that your revised manuscript can be swiftly handed over to our production team.

In recognition of the time and expertise our reviewers provide to Nature Cell Biology's editorial process, we would like to formally acknowledge their contribution to the external peer review of your manuscript entitled "Cyclophilin A supports translation of intrinsically disordered proteins and mitigates haematopoietic stem cell aging". For those reviewers who give their assent, we will be publishing their names alongside the published article.

Nature Cell Biology offers a Transparent Peer Review option for new original research manuscripts submitted after December 1st, 2019. As part of this initiative, we encourage our authors to support increased transparency into the peer review process by agreeing to have the reviewer comments, author rebuttal letters, and editorial decision letters published as a Supplementary item. When you submit your final files please clearly state in your cover letter whether or not you would like to

participate in this initiative. Please note that failure to state your preference will result in delays in accepting your manuscript for publication.

Cover suggestions

COVER ARTWORK: We welcome submissions of artwork for consideration for our cover. For more information, please see our guide for cover artwork.

Nature Cell Biology has now transitioned to a unified Rights Collection system which will allow our Author Services team to quickly and easily collect the rights and permissions required to publish your work. Approximately 10 days after your paper is formally accepted, you will receive an email in providing you with a link to complete the grant of rights. If your paper is eligible for Open Access, our Author Services team will also be in touch regarding any additional information that may be required to arrange payment for your article.

Please note that *Nature Cell Biology* is a Transformative Journal (TJ). Authors may publish their research with us through the traditional subscription access route or make their paper immediately open access through payment of an article-processing charge (APC). Authors will not be required to make a final decision about access to their article until it has been accepted. Find out more about Transformative Journals

Please use the following link for uploading these materials:
[Redacted]

Best regards,

Kendra Donahue

Staff
Nature Cell Biology

On behalf of

Daryl Jason Verzosa David, PhD

Senior Editor, Nature Cell Biology
Nature Portfolio
Advisory Editor, npj Biological Physics and Mechanics

Heidelberger Platz 3, 14197 Berlin, Germany
Email: daryl.david@nature.com
ORCID: <https://orcid.org/0000-0002-9253-4805>

Reviewer #1:

Remarks to the Author:

The authors have addressed my concerns even if not experimentally.
I am now in favor of publication.

Reviewer #2:

Remarks to the Author:

The authors have addressed my concerns sufficiently. Congratulations on an interesting story!

Author Rebuttal, second revision:

We are grateful to the reviewers for their dedication and commitment to our shared research.
We appreciate their positive comments and acceptance of our manuscript.

Reviewer #1:

Remarks to the Author:

The authors have addressed my concerns even if not experimentally.
I am now in favor of publication.

Thank you very much - we look forward to performing the outstanding experiments in the near future.

Reviewer #2:

Remarks to the Author:

The authors have addressed my concerns sufficiently. Congratulations on an interesting story!

Thank you very much for your kind words and support.

Reviewer #3:

Remarks to the Author:

The authors addressed my concerns. I don't have further comments.

Thank you very much for supporting our research.

Final Decision Letter:

Dear Dr Catic,

I am pleased to inform you that your manuscript, "Cyclophilin A supports translation of intrinsically disordered proteins and affects haematopoietic stem cell aging", has now been accepted for publication in Nature Cell Biology.

Please note that *Nature Cell Biology* is a Transformative Journal (TJ). Authors may publish their research with us through the traditional subscription access route or make their paper immediately open access through payment of an article-processing charge (APC). Authors will not be required to make a final decision about access to their article until it has been accepted. Find out more about Transformative Journals

If you have not already done so, we strongly recommend that you upload the step-by-step protocols used in this manuscript to the Protocol Exchange (www.nature.com/protocolexchange), an open online resource established by Nature Protocols that allows researchers to share their detailed experimental know-how. All uploaded protocols are made freely available, assigned DOIs for ease of citation and are

fully searchable through nature.com. Protocols and Nature Portfolio journal papers in which they are used can be linked to one another, and this link is clearly and prominently visible in the online versions of both papers. Authors who performed the specific experiments can act as primary authors for the Protocol as they will be best placed to share the methodology details, but the Corresponding Author of the present research paper should be included as one of the authors. By uploading your Protocols to Protocol Exchange, you are enabling researchers to more readily reproduce or adapt the methodology you use, as well as increasing the visibility of your protocols and papers. You can also establish a dedicated page to collect your lab Protocols. Further information can be found at www.nature.com/protocolexchange/about

With kind regards,
Daryl

Daryl Jason Verzosa David, PhD

Senior Editor, Nature Cell Biology
Nature Portfolio
Advisory Editor, npj Biological Physics and Mechanics

Heidelberger Platz 3, 14197 Berlin, Germany
Email: daryl.david@nature.com
ORCID: <https://orcid.org/0000-0002-9253-4805>

** Visit the Springer Nature Editorial and Publishing website at www.springernature.com/editorial-and-publishing-jobs for more information about our career opportunities. If you have any questions please click here.**